# Cold Start Streaming Learning for Deep Networks

## Abstract

The ability to dynamically adapt neural networks to newly-available data without performance deterioration would revolutionize deep learning applications. Streaming learning (i.e., learning from one data example at a time) has the potential to enable such real-time adaptation, but current approaches $i$) freeze a majority of network parameters during streaming and $ii$) are dependent upon offline, base initialization procedures over large subsets of data, which damages performance and limits applicability. To mitigate these shortcomings, we propose Cold Start Streaming Learning (CSSL), a simple, end-to-end approach for streaming learning with deep networks that uses a combination of replay and data augmentation to avoid catastrophic forgetting.

Because CSSL updates all model parameters during streaming, the algorithm is capable of beginning streaming from a random initialization, making base initialization optional. Going further, the algorithm's simplicity allows theoretical convergence guarantees to be derived using analysis of the Neural Tangent Random Feature (NTRF). In experiments, we find that CSSL outperforms existing baselines for streaming learning in experiments on CIFAR100, ImageNet, and Core50 datasets. Additionally, we propose a novel multi-task streaming learning setting and show that CSSL performs favorably in this domain. Put simply, CSSL performs well and demonstrates that the complicated, multi-step training pipelines adopted by most streaming methodologies can be replaced with a simple, end-to-end learning approach without sacrificing performance.

## 1 Introduction

**Background.** Many autonomous applications would benefit from real-time, dynamic adaption of models to new data. As such, online learning[1] has become a popular topic in deep learning; e.g., continual (Lopez-Paz & Ranzato, 2017; Zenke et al., 2017), lifelong (Aljundi et al., 2017; Chaudhry et al., 2018b), incremental (Rebuffi et al., 2017; Castro et al., 2018), and streaming (Hayes et al., 2018; Hayes et al., 2020) learning. However, the (potentially) non-i.i.d. nature of incoming data causes catastrophic forgetting (McCloskey & Cohen, 1989; Kirkpatrick et al., 2017), thus complicating the learning process.

Batch-incremental learning (Rebuffi et al., 2017; Castro et al., 2018), where batches of data—typically sampled from disjoint sets of classes or tasks within a dataset—become available to the model sequentially, is a widely-studied form of online learning. Within this setup, however, one must wait for a sizeable batch of data[2] to accumulate before the model is updated with an expensive, offline training procedure over new data, thus introducing latency that prevents real-time model updates.

To avoid such latency, we adopt the streaming learning setting where $i$) each data example is seen once and $ii$) the dataset is learned in a single pass (Hayes et al., 2018). Streaming learning performs brief, online updates (i.e., one or a few forward/backward passes) for each new data example, *forcing learning of new data to occur in real-time.* Additionally, streaming learning techniques can be adapted to cope with batches of data instead of single examples (Wang et al., 2018), while batch-incremental learning techniques tend to

---

[1]We use "online learning" to generically describe methodologies that perform training in a sequential manner.
[2]These "batches" are large (e.g., a 100-class subset of ImageNet with >100,000 data examples (Rebuffi et al., 2017)) and using smaller batches damages performance (Belouadah et al., 2020; Hayes et al., 2020).

deteriorate drastically given smaller batch sizes (Hayes et al., 2020). As such, the streaming learning setting, which has recently been explored for applications with deep neural networks (Hayes et al., 2018; Hayes et al., 2020; Hayes & Kanan, 2020), is generic and has the potential to enable low-cost updates of deep networks to incoming data.

Current streaming methodologies for deep neural networks learn in two-phases: base initialization and streaming. During base initialization, network parameters (and other modules, if needed) are pre-trained over a subset of data. Then, a majority of network parameters are frozen (i.e., not updated) throughout the learning process. During streaming, most approaches maintain a replay buffer—though other techniques exist (Hayes & Kanan, 2020)—to avoid catastrophic forgetting by allowing prior data to be sampled and included in each online update.

The current, multi-stage streaming learning pipeline suffers a few notable drawbacks. Namely, the learning process is dependent upon a latency-inducing, offline pre-training procedure and a majority of network parameters are not updated during the streaming process. As a result, the underlying network $i$) has reduced representational capacity, $ii$) cannot adapt a large portion of its parameters to new data, and $iii$) is dependent upon high-quality pre-training (during base initialization) to perform well. Given these considerations, one may begin to wonder whether a simpler streaming procedure could be derived to realize the goal of adapting deep networks to new data in real time.

**This Work.** Inspired by the simplicity of offline training, we propose a novel method for streaming learning with deep networks, called Cold Start Streaming Learning (CSSL), that updates all network parameters in an end-to-end fashion throughout the learning process. Because no parameters are fixed by CSSL, base initialization and pre-training procedures are optional—streaming can begin from a completely random initialization (hence, "cold start" streaming learning). By leveraging a basic replay mechanism coupled with sophisticated data augmentation techniques, CSSL outperforms existing streaming techniques in a variety of domains. Furthermore, the simplicity of the approach makes our algorithm more apt to theoretical analysis, as well as easier to implement and deploy in practice. A summary of our contributions is as follows:

- We propose CSSL, a simple streaming methodology that combines replay buffers with sophisticated data augmentation, and provide extensive comparison to existing techniques on common class-incremental streaming problems. We show that CSSL often outperforms baseline methodologies by a large margin.

- We leverage techniques related to neural tangent random feature (NTRF) analysis (Cao & Gu, 2019) to prove a theoretical bound on the generalization loss of CSSL over streaming iterations.

- We propose a multi-task streaming learning benchmark, where multiple, disjoint classification datasets are presented to the model sequentially and in a streaming fashion. We show that CSSL enables significant performance improvements over baseline methodologies in this new domain.

- We extensively analyze models trained via CSSL, showing that resulting models $i$) achieve impressive streaming performance even when beginning from a random initialization (i.e., a cold start); $ii$) are robust to the compression of examples in the replay buffer for reduced memory overhead; and $iii$) provide highly-calibrated confidence scores due to our proposed data augmentation policy.

## 2 Related Work

**Online Learning.** Numerous experimental setups have been considered for online learning, but they all share two properties: $i$) the sequential nature of the training process and $ii$) performance deterioration due to catastrophic forgetting when incoming data is non-i.i.d. (McCloskey & Cohen, 1989; Kemker et al., 2018). Replay mechanisms, which maintain a buffer of previously-observed data (or a generative model to produce such data (Rannen et al., 2017; Shin et al., 2017)) to include in online updates, are highly-effective at preventing catastrophic forgetting at scale (Douillard et al., 2020; Chaudhry et al., 2019; Hayes et al., 2020), leading us to base our proposed methodology upon replay. Similarly, knowledge distillation (Hinton et al., 2015) can prevent performance deterioration by stabilizing feature representations throughout the online learning process (Hou et al., 2019; Wu et al., 2019), even while training the network end-to-end (e.g., end-to-end incremental learning (Castro et al., 2018) and iCarl (Rebuffi et al., 2017)). Though distillation

and replay are widely-used, numerous other approaches to online learning also exist (e.g., architectural modification (Rusu et al., 2016; Draelos et al., 2017), regularization (Dhar et al., 2019; Li & Hoiem, 2017), and dual memory (Kemker & Kanan, 2017; Belouadah & Popescu, 2019)).

**Streaming Learning.** Streaming, which we study in this work, performs a single pass over the dataset, observing each sample once (Hayes et al., 2018). Having recently become popular in deep learning, streaming learning (Hayes et al., 2020; Acharya et al., 2020; Hayes & Kanan, 2020; Gallardo et al., 2021) trains the model in two-phases: base initialization and streaming. Base initialization uses a subset of data to pre-train the model and initialize relevant network modules. Then, the streaming phase learns the dataset in a single-pass with a majority of network parameters fixed. Within this training paradigm, replay-based techniques perform well at scale (Hayes et al., 2020), though other approaches may also be effective (Hayes & Kanan, 2020).

**Data Augmentation.** The success of the proposed methodology is enabled by our data augmentation policy. Though data augmentation for computer vision is well-studied (Shorten & Khoshgoftaar, 2019), we focus upon interpolation methods and learned augmentation policies. Interpolation methods (e.g., Mixup (Zhang et al., 2017; Wolfe & Lundgaard, 2019; Inoue, 2018) and CutMix (Yun et al., 2019)) take stochastically-weighted combinations of images and label pairs during training, which provides regularization benefits. Learned augmentation policies (e.g., AutoAugment (Cubuk et al., 2018; Lim et al., 2019; Hataya et al., 2020) and RandAugment (Cubuk et al., 2020)), on the other hand, consider a wide scope of augmentation techniques and adopt a data-centric approach to learn an optimal augmentation policy (i.e., using reinforcement learning or gradient-based techniques).

**Confidence Calibration.** Beyond the core methodology of CSSL, we extensively explore the confidence calibration properties of resulting models. Put simply, confidence calibration is the ability of a model to accurately predict the probability of its own correctness—poor predictions should be made with low confidence and vice versa. Numerous methodologies have been proposed for producing calibrated models, including post-hoc softmax temperature scaling (Guo et al., 2017), ensemble-based techniques (Lakshminarayanan et al., 2017), Mixup (Zhang et al., 2017; Thulasidasan et al., 2019), Monte-Carlo dropout, and uncertainty estimates in Bayesian networks (Neal, 2012; Gal & Ghahramani, 2016). Confidence calibration has also been applied to problems including domain shift and out-of-distribution detection due to its ability to filter incorrect, or low confidence predictions (Hendrycks & Gimpel, 2016; Hendrycks & Dietterich, 2019).

**Multi-task learning.** *Our work is the first to consider multi-task streaming learning*, though multi-task learning has been previously explored for both offline and batch-incremental settings. (Hou et al., 2018) studies the sequential learning of several datasets via knowledge distillation and replay, and several other works study the related problem of domain shift (Jung et al., 2016; Furlanello et al., 2016), where two datasets are learned in a sequential fashion. For the offline setting, multi-task learning has been explored extensively, leading to a variety of successful learning methodologies for computer vision (Lu et al., 2020), natural language processing (Stickland & Murray, 2019), multi-modal learning (Wolfe & Lundgaard, 2021; Nguyen & Okatani, 2019), and more. The scope of work in offline multi-task learning is too broad to provide a comprehensive summary here, though numerous surveys on the topic are available (Ruder, 2017; Worsham & Kalita, 2020).

## 3   Methodology

The proposed streaming methodology, formulated in Algorithm 1, begins either from pre-trained parameters or a random-initialization (`Initialize` in Algorithm 1) and utilizes a replay buffer $\mathcal{R}$ to prevent catastrophic forgetting. At each iteration, CSSL receives a new data example $x_t, y_t := \mathcal{D}_t$, combines this data with random samples from the replay buffer, performs a stochastic gradient descent (SGD) update with the combined data, and stores the new example within the buffer for later replay. Within this section, we will first provide relevant preliminaries and definitions, then each component of CSSL will be detailed and contrasted with prior work.

---

**Algorithm 1:** Cold Start Streaming Learning

---

**Parameters: W** model parameters, $\mathcal{D}$ data stream, $\mathcal{R}$ replay buffer, $C$ maximum replay buffer size,
  $B$ number of replay samples per iteration

\# **Initialize** model parameters randomly or via pre-training (if possible)
**W** := `Initialize()`
$\mathcal{R} := \emptyset$
**for** $t = 1, \ldots , |\mathcal{D}|$ **do**

  \# **Sample** data from the replay buffer to combine with new data from $\mathcal{D}$
  $x_{\text{new}}, \ y_{\text{new}} := \mathcal{D}_t$
  $\mathcal{X}_{\text{replay}}, \ \mathcal{Y}_{\text{replay}} := \texttt{ReplaySample}(\mathcal{R}, \ B)$
  $\mathcal{X}, \ \mathcal{Y} := \{x_{\text{new}}\} \cup \mathcal{X}_{\text{replay}}, \ \{y_{\text{new}}\} \cup \mathcal{Y}_{\text{replay}}$

  \# **Update** all model parameters over augmented new and replayed data
  `StreamingUpdate(`**W**`, Augment(`$\mathcal{X}, \ \mathcal{Y}$`))`

  \# **Compress** and **Store** the new data example for replay
  `ReplayStore(`$\mathcal{R}, (\texttt{Compress}(x_{\text{new}}), y_{\text{new}})$`)`

  \# **Evict** data from the replay buffer to maintain capacity
  **if** $|\mathcal{R}| > C$ **then**
  | `ReplayEvict(`$\mathcal{R}$`)`
  **end**
**end**

---

### 3.1 Preliminaries

**Problem Setting.** Following prior work (Hayes et al., 2020), we consider the problem of streaming for image classification.[3] In most cases, we perform class-incremental streaming experiments, in which examples from each distinct semantic class within the dataset are presented to the learner sequentially. However, experiments are also performed using other data orderings; see Section 6.2. Within our theoretical analysis only, we consider a binary classification problem that maps vector inputs to binary labels; see Section 5 for further details. Such a problem setup is inspired by prior work (Cao & Gu, 2019; Allen-Zhu et al., 2019).

**Streaming Learning Definition.** Streaming learning considers an incoming data stream $\mathcal{D} = \{(x_t, y_t)\}_{t=1}^{n}$[4] (i.e., $t$ denotes temporal ordering) and adopts the following rule set during training:

  1. Each unique data example appears once in $\mathcal{D}$.
  2. The order of data within $\mathcal{D}$ is arbitrary and possibly non-iid.
  3. Model evaluation may occur at any time $t$.

Notably, these requirements make no assumptions about model state prior to the commencement of the streaming process. Previous methodologies perform offline, base initialization procedures before streaming begins, while CSSL may either begin streaming from randomly-initialized or pre-trained parameters. As such, CSSL is capable of performing streaming even without first observing examples from $\mathcal{D}$, while other methodologies are reliant upon data-dependent base initialization.

**Evaluation Metric.** In most experiments, performance is measured using $\Omega_{\text{all}}$ (Hayes et al., 2018), defined as $\Omega_{\text{all}} = \frac{1}{T} \sum_{t=1}^{T} \frac{\alpha_t}{\alpha_{\text{offline},t}}$, where $\alpha_t$ is streaming performance at testing event $t$, $\alpha_{\text{offline},t}$ is offline performance at testing event $t$, and $T$ is the number of total testing events. $\Omega_{\text{all}}$ captures aggregate streaming performance (relative to offline training) throughout streaming. A higher score indicates better performance.

As an example, consider a class-incremental streaming setup with two testing events: one after ½ of the classes have been observed and one at the end of streaming. The streaming learner is evaluated at each testing

---

[3]Streaming learning has been considered in the object detection domain (Acharya et al., 2020), but we consider this application beyond the scope of our work.
[4]E.g., $x_t \in \mathbb{R}^{3 \times H \times W}$ (RGB image) and $y_t \in \mathbb{Z}^{\geq 0}$ (class label) for computer vision applications.

event, yielding accuracy $\alpha_1$ and $\alpha_2$. Two models are trained offline over ½ of the classes and the full dataset, respectively, then evaluated to yield $\alpha_{\text{offline},1}$ and $\alpha_{\text{offline},2}$. From here, $\Omega_{\text{all}}$ is given by $\frac{1}{2}\left(\frac{\alpha_1}{\alpha_{\text{offline},1}} + \frac{\alpha_2}{\alpha_{\text{offline},2}}\right)$.

### 3.2 Cold Start Streaming Learning

**Replay Buffer.** As the learner progresses through the data stream, full images and their associated labels are stored (`ReplayStore`) within a fixed-size replay buffer $\mathcal{R}$. The replay buffer $\mathcal{R}$ has a fixed capacity $C$. Data can be freely added to $\mathcal{R}$ if $|\mathcal{R}| < C$, but eviction must occur (`ReplayEvict` in Algorithm 1) to make space for new examples when the buffer is at capacity. One of the simplest eviction policies—which is adopted in prior work (Hayes et al., 2020)—is to $i$) identify the class containing the most examples in the replay buffer and $ii$) remove a random example from this class. This policy, which we adopt in CSSL, is computationally efficient and performs similarly to more sophisticated techniques; see Appendix B.3.

**Data Compression.** Data is compressed (`Compress` in Algorithm 1) before addition to $\mathcal{R}$. By freezing a majority of network layers during streaming, previous methods can leverage learned quantization modules and pre-trained feature representations to drastically reduce the size of replay data (Hayes & Kanan, 2020; Hayes et al., 2020). For CSSL, full images, which induce a larger memory overhead, are stored in $\mathcal{R}$ and no network parameters are fixed, meaning that `Compress` cannot rely on pre-trained, fixed network layers.

We explore several, data-independent `Compress` operations, such as resizing images, quantizing the integer representations of image pixels, and storing images on disk with JPEG compression; see Appendix B.2. Such compression methodologies significantly reduce memory overhead without impacting performance. Because storing the replay buffer on disk is not always possible (e.g., edge devices may lack disk-based storage), we present this as a supplemental approach and always present additional results without JPEG compression.

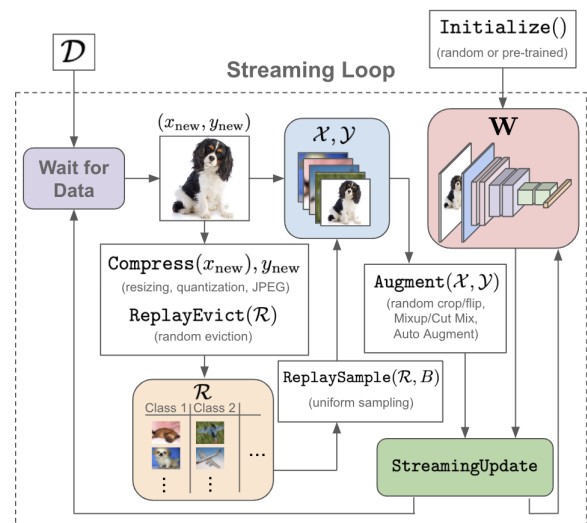

Figure 1: Illustration of CSSL following the notation of Algorithm 1.

Due to storing full images in the replay buffer, the proposed methodology has more memory overhead relative to prior techniques, making it less appropriate for memory-limited scenarios. However, many streaming applications (e.g., e-commerce recommendation systems, pre-labeling for data annotation, etc.) store and update models on cloud servers, making memory efficiency less of a concern. *We recommend CSSL for such scenarios, as it outperforms prior methods given sufficient memory for replay.*

**Model Updates.** For each new example within $\mathcal{D}$, CSSL updates learner parameters $\Theta$ (`StreamingUpdate` in Algorithm 1) using the new example and $N$ replay buffer samples obtained via `ReplaySample`. Similar to prior work, `ReplaySample` uniformly samples data from $\mathcal{R}$—alternative sampling strategies provide little benefit at scale (Aljundi et al., 2019b;a; Hayes et al., 2020); see Appendix B.4. `StreamingUpdate` is implemented as a simple SGD update of all network parameters over new and replayed data.

Streaming updates pass the mixture of new and replayed data through a data augmentation pipeline (`Augment` in Algorithm 1). Though prior work uses simple augmentations (Castro et al., 2018; Verma et al., 2019; Hayes et al., 2020), we explore data interpolation (Zhang et al., 2017; Yun et al., 2019) and learned augmentation policies (Cubuk et al., 2018). In particular, CSSL combines random crops and flips, Mixup, Cutmix, and

Autoaugment into a single, sequential augmentation policy[5]; see Appendix B.1 for further details. *Curating a high quality data augmentation pipeline is the key CSSL's impressive performance.*

**Our Methodology.** In summary, CSSL, illustrated in Figure 1, initializes the network either randomly or with pre-trained parameters (`Initialize`). Then, for each new data example, $N$ examples are sampled uniformly from the replay buffer (`ReplaySample`), combined with the new data, passed through a data augmentation pipeline (`Augment`), and then used to perform a training iteration (`StreamingUpdate`). Each new data example is added to the replay buffer (`ReplayStore`), and an example may be randomly removed from the replay buffer via random, uniform selection (`ReplayEvict`) if the capacity $C$ is exceeded.

## 4    Why is this useful?

The proposed methodology has two major benefits:

- **Full Plasticity** (i.e., no fixed parameters)

- **Pre-Training is Optional**

Such benefits are unique to CSSL; see Appendix C. However, one may question the validity of these "benefits"—*do they provide any tangible value?*

**Full Plasticity.** CSSL updates all model parameters throughout the streaming process.[6] Some incremental learning methodologies train networks end-to-end(Castro et al., 2018; Hou et al., 2019; Wu et al., 2019), but these approaches either $i$) cannot be applied or $ii$) perform poorly when adapted to the streaming domain(Hayes et al., 2020); see Appendix B.6. Fixing network parameters is detrimental to the learning process. To show this, we pre-train a ResNet18 on ImageNet and fix different ratios of

| No. Base Init.Classes | Top-1 $\Omega_{\text{all}}$ | Top-5 $\Omega_{\text{all}}$ |
|:---:|:---:|:---:|
| 10 | 0.478 | 0.651 |
| 50 | 0.727 | 0.848 |
| 100 | 0.835 | 0.856 |

Table 1: $\Omega_{\text{all}}$ of REMIND on ImageNet with different amounts of base initialization data. *REMIND performance deteriorates with less data.*

network parameters during fine-tuning on CIFAR10/100, finding that final accuracy deteriorates monotonically with respect to the ratio of frozen parameters; see Figure 2. Even given high-quality pre-training, fixing network parameters $i$) limits representational capacity and $ii$) prevents network representations from being adapted to new data, making end-to-end training a more favorable approach.

**Pre-Training is Optional.** Prior streaming methods perform base initialization (i.e., fitting of network parameters and other modules over a subset of data) prior to streaming. Not only does base initialization introduce an expensive, offline training procedure, but it makes model performance dependent upon the availability of sufficient (possibly unlabeled (Gallardo et al., 2021)) pre-training data prior to streaming. Streaming performance degrades with less base initialization data; see Table 1. Thus, if little (or worse, no) data is available when streaming begins, such methods are doomed to poor performance.

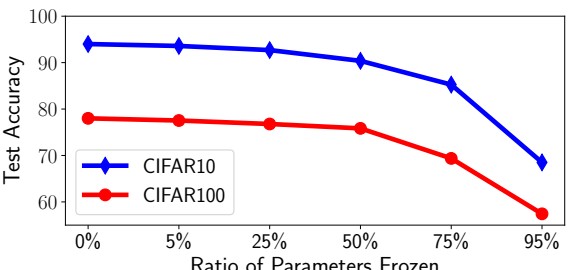

Figure 2: Test acccuracy of ResNet18 models that are pre-trained on ImageNet and fine-tuned on CIFAR10/100 with different ratios of frozen parameters.

CSSL has no dependency upon pre-training because the network is trained end-to-end during streaming. Thus, the CSSL training pipeline is quite simple—*just initialize, then train.* Such simplicity makes CSSL

---

[5]Mixup and Cutmix are not performed simultaneously. Our policy randomly chooses between Mixup or Cutmix for each data example with equal probability.

[6]Training the network end-to-end within CSSL does make model inference and updates more computationally expensive, which adds latency to streaming updates; see Appendix B.7 for further analysis.

easy to implement and deploy in practice. We emphasize that CSSL *can* begin streaming from a pre-trained set of model parameters, which often leads to better performance. The benefit of CSSL, however, is that such a pre-training step is *completely optional* and network parameters are still updated during streaming.

**Implications and Potential Applications.** The proposed methodology yields improvements in *performance*, *simplicity*, and *applicability* relative to prior work. Fixing no network parameters enables better *performance* by increasing network representational capacity and adaptability, while the ability to stream from a cold start *simplifies* the streaming pipeline and makes streaming *applicable* to scenarios that lack sufficient data for base initialization.

Consider, for example, pre-labeling for data annotation, where a streaming learner works with a human annotator to improve labeling efficiency. In this case, the annotated dataset may be built from scratch, meaning that no data is yet available when streaming begins. Similarly, cold-start scenarios for recommendation or classification systems often lack prior data from which to learn. Though pre-trained models may sometimes be downloaded online, this is not possible for domains that align poorly with public models and datasets (e.g., medical imaging). While prior approaches fail in these scenarios due to a dependence upon high-quality base initialization, CSSL can simply begin streaming from a random initialization and still perform well.

## 5 Theoretical Results

CSSL's simple training pipeline enables the algorithm to be analyzed theoretically. To do this, we extend analysis of the neural tangent random feature (NTRF) (Cao & Gu, 2019) to encompass multi-layer networks trained via streaming to perform binary classification using a replay mechanism and data augmentation. The analytical setup is similar to Algorithm 1 with a few minor differences. We begin by providing relevant notation and definitions, then present the main theoretical result.

### 5.1 Preliminaries and Definitions

**Notation.** We denote scalars and vectors with lower case letters, distinguished by context, and matrices with upper case letters. We define $[l] = \{1, 2, \ldots, l\}$ and $\|v\|_p = (\sum_{i=1}^d |v_i|^p)^{\frac{1}{p}}$ for $v = (v_1, v_2, \ldots, v_d)^\top \in \mathbb{R}^d$ and $1 \le p < \infty$. We also define $\|v\|_\infty = \max_i |v_i|$. For matrix $A \in \mathbb{R}^{m \times n}$, we use $\|A\|_0$ to denote the number of non-zero entries in $A$. We use $i \sim \mathcal{N}$ to denote a random sample $i$ from distribution $\mathcal{N}$, and $\mathcal{N}(\mu, \sigma)$ to denote the normal distribution with mean $\mu$ and standard deviation $\sigma$. $\mathbb{1}\{\cdot\}$ is the indicator function. We then define $\|A\|_F = \sqrt{\sum_{i,j} A_{i,j}^2}$ and $\|A\|_p = \max_{\|v\|_p=1} \|Av\|_p$ for $p \ge 1$ and $v \in \mathbb{R}^n$. For two matrices $A, B \in \mathbb{R}^{m \times n}$, we have $\langle A, B \rangle = \text{Tr}(A^\top B)$. For vector $v \in \mathbb{R}^d$, $\texttt{diag}(v) \in \mathbb{R}^{d \times d}$ is a diagonal matrix with the entries of $v$ on the diagonal. We also define the asymptotic notations $\mathcal{O}(\cdot)$ and $\Omega(\cdot)$ as follows. If $a_n$ and $b_n$ are two sequences, $a_n = \mathcal{O}(b_n)$ if $\limsup_{n \to \infty} |\frac{a_n}{b_n}| < \infty$ and $a_n = \Omega(b_n)$ if $\limsup_{n \to \infty} |\frac{a_n}{b_n}| > 0$.

**Network Formulation.** We consider fully forward neural networks with width $m$, depth $L$, input dimension $d$, and an output dimension of one. The weight matrices of the network are $W_1 \in \mathbb{R}^{m \times d}$, $W_\ell \in \mathbb{R}^{m \times m}$ for $\ell = 2, \ldots, L-1$, and $W_L \in \mathbb{R}^{1 \times m}$.[7] We denote $\mathbf{W} = \{W_1, \ldots, W_L\}$ and define $\langle \mathbf{W}, \mathbf{W}' \rangle = \sum_{i=1}^L \langle W_i, W_i' \rangle$ when $\mathbf{W}, \mathbf{W}'$ are two sets containing corresponding weight matrices with equal dimension. All of such sets with corresponding weight matrices of the same dimension form the set $\mathcal{W}$. Then, a forward pass with weights $\mathbf{W} = \{W_1, \ldots, W_L\} \in \mathcal{W}$ for input $x \in \mathbb{R}^d$ is given by:

$$f_{\mathbf{W}}(x) = \sqrt{m} \cdot W_L \sigma(W_{L-1} \sigma(W_{L-2} \ldots \sigma(W_1(x)) \ldots)), \tag{1}$$

where $\sigma(\cdot)$ is the ReLU activation function. Following (Cao & Gu, 2019), the first and last weight matrices are not updated during training.

**Objective Function.** For data stream $\mathcal{D} = \{(x_t, y_t)\}_{t=1}^n$, we define $L_{\mathcal{D}}(\mathbf{W}) \triangleq \mathbb{E}_{(x,y) \sim \mathcal{D}} L_{(x,y,\xi)}(\mathbf{W})$, where $L_{(x,y,\xi)}(\mathbf{W}) = \ell[y \cdot f_{\mathbf{W}}(x + \xi)]$ is the loss over $(x, y) \in \mathcal{D}$ with arbitrary perturbation vector $\xi$—representing data augmentation—and $\ell(z) = \log(1 + e^{-z})$ is the cross-entropy loss. We use the notation

---

[7] Assuming the widths of each hidden layer are the same is a commonly-used simplification. See (Allen-Zhu et al., 2019; Cao & Gu, 2019) for papers that have adopted the same assumption.

$L_{(t,\xi)}(\cdot) = L_{(x_t,y_t,\xi)}(\cdot)$ for convenience, and denote the 0-1 generalization error over the entire data stream $\mathcal{D}$ as $L_{\mathcal{D}}^{0-1}(\mathbf{W}) \triangleq \mathbb{E}_{(x,y)\sim\mathcal{D}}[\mathbb{1}\{y \cdot f_{\mathbf{W}}(x + \xi) < 0\}]$ with arbitrary perturbation vector $\xi$.

**CSSL Formulation.** Following Algorithm 1, our analysis considers a random initialization (`Initialize`) of model parameters $\mathbf{W}^{(0)} = \{W_1^{(0)}, \ldots, W_L^{(0)}\}$ as shown below:

$$W_j^{(0)} \sim \mathcal{N}\left(0, \frac{2}{m}\right), \ \forall \, j \in [L-1],$$
$$W_L^{(0)} \sim \mathcal{N}\left(0, \frac{1}{m}\right). \tag{2}$$

A buffer $\mathcal{R}$ of size $C$ is used to maintain data for replay. All incoming data is added to $\mathcal{R}$ (`ReplayStore`), and an entry is randomly (uniformly) evicted from $\mathcal{R}$ (`ReplayEvict`) if $|\mathcal{R}| > C$.

Each online update takes $B$ uniform samples from $\mathcal{R}$ (`ReplaySample`), forming a set $\mathcal{S}_t$ of replay examples at iteration $t$. The $t$-th update of model parameters $\mathbf{W}$ (`StreamingUpdate`) over new and replayed data is given by the following expression:

$$\mathbf{W}^{(t+1)} = \mathbf{W}^{(t)} - \eta\left(\nabla_{\mathbf{W}^{(t)}} L_{(x_t,y_t,\xi_t)}(\mathbf{W}^{(t)}) + \sum_{(x_{\text{rep}},y_{\text{rep}})\in\mathcal{S}_t} \nabla_{\mathbf{W}^{(t)}} L_{(x_{\text{rep}},y_{\text{rep}},\xi_{\text{rep}})}(\mathbf{W}^{(t)})\right), \tag{3}$$

where $\eta$ is the learning rate, $\xi_t$ is an arbitrary perturbation vector for the $t$-th data stream example, and $\xi_{\text{rep}}$ is an arbitrary perturbation vector that is separately generated for each individual replay example. Our theoretical analysis does not consider the compression of replay examples (`Compress`$(x) = x$).

**Data Augmentation.** The $t$-th example in the data stream $(x_t, y_t) \in \mathcal{D}$ is augmented (`Augment`) using the perturbation vector $\xi_t$; i.e., the network always observes an augmented version of the data, given by $(x_t + \xi_t, y_t)$. No assumptions are made about vectors $\xi_t$, though our final rate depends on their magnitude. Similarly, all replay samples are augmented via $\xi_{\text{rep}}$, such that the network observes $(x_{\text{rep}} + \xi_{\text{rep}}, y_{\text{rep}})$ for each $(x_{\text{rep}}, y_{\text{rep}}) \in \mathcal{S}_t$. All replay samples have unique perturbation vectors $\xi_{\text{rep}}$, but we use $\xi_{\text{rep}}$ to denote all replay perturbation vectors for simplicity. We denote the set of all perturbation vectors used for new and replayed data throughout streaming as $\mathcal{X}$ and define $\Xi = \sup_{\xi\in\mathcal{X}} \|\xi\|_2^2$.

**Neural Tangent Random Feature.** For $\mathbf{W} \in \mathcal{W}$, we define the $\omega$-neighborhood as follows:

$$\mathcal{B}(\mathbf{W}, \omega) \triangleq \{\mathbf{W}' \in \mathcal{W} : \|W_l' - W_l\|_F \leq \omega, \ \forall l \in [L]\}.$$

Given a set of all-zero weight matrices $\mathbf{0} \in \mathcal{W}$ and randomly initialized (according to equation 2) $\mathbf{W}^{(0)}$, the Neural Tangent Random Feature (NTRF) (Cao & Gu, 2019) is then defined as the following function set

$$\mathcal{F}(\mathbf{W}^{(0)}, R) = \{f(\cdot) = f_{\mathbf{W}^{(0)}}(\cdot) + \langle\nabla_{\mathbf{W}} f_{\mathbf{W}^{(0)}}(\cdot), \mathbf{W}\rangle : \mathbf{W} \in \mathcal{B}(\mathbf{0}, R)\},$$

where $R > 0$ controls the size of the NTRF. We use the following shorthand to denote the first order Taylor approximation of network output, given weights $\mathbf{W}, \mathbf{W}' \in \mathcal{W}$ and input $x \in \mathbb{R}^d$:

$$F_{\mathbf{W},\mathbf{W}'}(x) \triangleq f_{\mathbf{W}}(x) + \langle\nabla_{\mathbf{W}} f_{\mathbf{W}}(x), \mathbf{W}' - \mathbf{W}\rangle.$$

Using this shorthand, the NTRF can be alternatively formulated as:

$$\mathcal{F}(\mathbf{W}^{(0)}, R) = \{f(\cdot) = F_{\mathbf{W}^{(0)},\mathbf{W}'}(\cdot) : \mathbf{W}' - \mathbf{W}^{(0)} \in \mathcal{B}(\mathbf{0}, R)\}.$$

## 5.2 Convergence Guarantees

Here, we present the main result of our theoretical analysis for CSSL—an upper bound on the zero-one generalization error of networks trained via Algorithm 1, as described in Section 5.1. Proofs are deferred to Appendix E, but we provide a sketch in this section. Our analysis adopts the following set of assumptions.

*Assumption* 1. For all data $(x_t, y_t) \in \mathcal{D}$, we assume $\|x_t\|_2 = 1$. Given two data examples $(x_i, y_i), (x_j, y_j) \in \mathcal{D}$, we assume $\|x_i - x_j\|_2 \geq \lambda$.

The conditions in Assumption 1 are widely-used in analysis of overparameterized neural network generalization (Cao & Gu, 2019; Allen-Zhu et al., 2019; Oymak & Soltanolkotabi, 2020; Zou et al., 2018; Li & Liang, 2018). The unit norm assumption on the data is adopted for simplicity but can be relaxed to $c_1 \leq \|x_i\| \leq c_2$ for all $(x_i, y_i) \in \mathcal{D}$ and absolute constants $c_1, c_2 > 0$. Given these assumptions, we derive the following:

**Theorem 1.** *Assume Assumption 1 holds, $\omega \leq \mathcal{O}\left(L^{-6} \log^{-3}(m)\right)$, and $m \geq \mathcal{O}\left(nB\sqrt{1+\Xi}L^6 \log^3(m)\right)$. Then, defining $\hat{\mathbf{W}}$ as a uniformly-random sample from iterates $\{\mathbf{W}^{(0)}, \ldots, \mathbf{W}^{(n)}\}$ obtained via Algorithm 1 with $\eta = \mathcal{O}\left(m^{-\frac{3}{2}}\right)$ and $B$ replay samples chosen at every iteration from buffer $\mathcal{R}$, we have the following with probability at least $1 - \delta$:*

$$\mathbb{E}\left[L_{\mathcal{D}}^{0-1}\left(\hat{\mathbf{W}}\right)\right] \leq \inf_{f \in \mathcal{F}(\mathbf{W}^{(0)}, R/m)} \left(\frac{4}{n} \sum_{t=1}^{n} \ell(y_t \cdot f(x_t + \xi_t))\right) + \sqrt{\frac{2\log(\frac{1}{\delta})}{n}} + \mathcal{O}\left(\frac{(1+\Xi)^{\frac{3}{4}} B^{\frac{1}{2}} n + R^2}{L^2 \log^{\frac{3}{2}}(m)}\right)$$

*where $\xi_t$ is an arbitrary perturbation vector representing data augmentation applied at streaming iteration $t$, $R > 0$ is a constant controlling the size of the NTRF function class, and $\delta \in [0, 1)$.*

**Discussion.** Theorem 1 provides an upper bound on the 0-1 generalization error over $\mathcal{D}$ for a network trained via Algorithm 1. This bound has three components. The first component—shaded in red—captures the 0-1 error achieved by the best function within the NTRF $\mathcal{F}(\mathbf{W}^{(0)}, R/m)$. Intuitively, this term captures the "classifiability" of the data—it has a large value when no function in the $\omega$-neighborhood of $\mathbf{W}^{(0)}$ can fit the data well and vice versa. Given fixed $m$, however, this term can be made arbitrarily small by increasing $R$ to enlarge the NTRF and, in turn, increase the size of the function space considered in the infimum.

The second term—shaded in green—is a probabilistic term that arises after invoking an online-to-batch conversion argument (Cesa-Bianchi et al., 2004) to relate training error to 0-1 generalization error. This term contains $\log(1/\delta)$ in the numerator and $n$ in the denominator. Smaller values of $\delta$ will make the bound looser but allow it to hold with higher probability. On the other hand, larger values of $n$ make this term arbitrarily small, meaning that the bound can be improved by observing more data.

The final term—shaded in blue—considers all other additive terms that arise from technical portions of the proof. This asymptotic expression grows with $\Xi$, $R$, $B$, and $n$. However, the term $L^2 \log^{\frac{3}{2}}(m)$ occurs in the denominator, revealing that this final term will shrink as the model is made wider and deeper.[8] Thus, Theorem 1, as a whole, shows that the network generalizes well given sufficiently large $m$, $L$, and $n$—*meaning that the network is sufficiently overparameterized and enough data has been observed.*

**Sketch.** The proof of Theorem 1—provided in Appendix E—proceeds as follows:

- We first show that the average 0-1 loss of iterates $\{\mathbf{W}^{(0)}, \ldots, \mathbf{W}^{(n)}\}$ obtained during streaming can be upper bounded with a constant multiple of the average cross-entropy loss of the same iterates.

- Invoking Lemma 1, we convert this average loss of iterates $\{\mathbf{W}^{(0)}, \ldots, \mathbf{W}^{(n)}\}$ to an average loss of weights within the set $\mathbf{W}^\star \in \mathcal{B}(\mathbf{W}^{(0)}, R/m)$, where $R > 0$.

- Using an online-to-batch conversion argument (see Proposition 1 in (Cesa-Bianchi et al., 2004)), we lower bound average 0-1 loss with expected 0-1 loss over $\mathcal{D}$ with probability $1 - \delta$.

- From here, we leverage Lemma 3 to relate average loss of $\mathbf{W}^\star$ over streaming iterations to the NTRF, taking an infimum over all functions $f \in \mathcal{F}(\mathbf{W}^{(0)}, R/m)$ to yield the final bound.

---

[8]Observe that $n$ appears in the denominator of the green term and the numerator of the blue term of Theorem 1. However, because the model is assumed to be overparameterized (i.e., $m \gg n$), the blue term is still negligible even given values of $n$ that are sufficiently large to make the green term small.

| Method | Base Init. Required | Ratio of Params Frozen | Uses Replay |
|---|---|---|---|
| ExStream (Hayes et al., 2018) | ✓ | 95% | ✓ |
| DeepSLDA (Hayes & Kanan, 2020) | ✓ | 95% | ✗ |
| REMIND (Hayes et al., 2020) | ✓ | 75% | ✓ |
| CSSL | ✗ | 0% | ✓ |

Table 2: A basic outline of properties for each of the considered streaming algorithms.

## 6 Experiments

Following prior work, we use ResNet18 to perform class-incremental streaming experiments on CIFAR100 and ImageNet (Krizhevsky et al., 2009; Deng et al., 2009), experiments with various different data orderings on Core50 (Lomonaco & Maltoni, 2017), and experiments using a novel, multi-task streaming learning setting. As baselines, we adopt the widely-used streaming algorithms ExStream (Hayes et al., 2018), Deep SLDA (Hayes & Kanan, 2020), and REMIND (Hayes et al., 2020); see Table 2 for details. Because REMIND freezes most network layers after base initialization, we also study a REMIND variant ("Remind + Extra Params") with added, trainable layers such that the number of trainable parameters is equal to CSSL.[9]

All experiments split the dataset(s) into several batches and data ordering is fixed between comparable experiments. Evaluation occurs after each batch and baselines use the first batch for base initialization—the first batch is seen both during base initialization and streaming. CSSL performs no base initialization, but may begin the streaming process with pre-trained parameters—such experiments are identified accordingly. We first present class-incremental streaming experiments, followed by the analysis of different data orderings and multi-task streaming. The section concludes with supplemental analysis that considers training other network architectures and studies behavioral properties of networks trained via CSSL.

### 6.1 Class-Incremental Learning

We perform class-incremental streaming learning experiments using a ResNet18 architecture on CIFAR100 and ImageNet. The results of these experiments are summarized below, and a discussion of the results follows.

**CIFAR100.** The dataset is divided into batches containing 20 classes and results are averaged over three different data orderings. Experiments are conducted using replay buffer memory capacities ranging from 30Mb (i.e., 10K CIFAR100 images) to 150Mb. These capacities are not relevant to Deep SLDA or REMIND, as Deep SLDA does not use a replay buffer and REMIND can store the full dataset using $< 30$Mb of memory. CSSL compresses replay examples by quantizing pixels to 6 bits and resizing images to 66% of their original area. In some cases, CSSL is initialized with parameters that are pre-trained over the first 20 classes of CIFAR100; see Appendix A.1 for details. The results of these experiments are illustrated in Figure 3.

**ImageNet.** The dataset is divided into batches of 100 classes. Following (Hayes et al., 2020), we per-

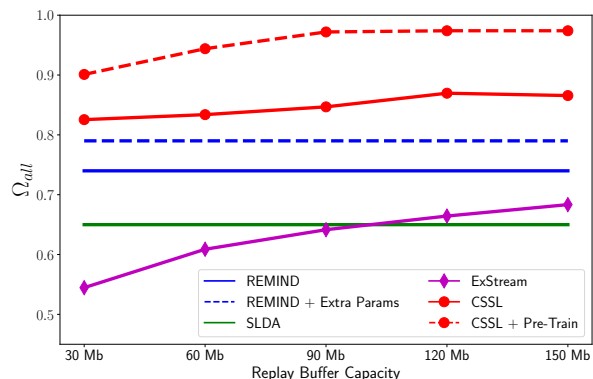

Figure 3: Streaming performance on CIFAR100 using different replay buffer capacities. *CSSL outperforms all existing approaches, no matter the buffer capacity, and performs even better with pre-training.*

---

[9]This modified version of REMIND has the same number of trainable parameters as a normal ResNet18 model and is included as a baseline to ensure the benefit of CSSL is not simply due to the presence of more trainable parameters within the underlying network.

form experiments with replay buffer memory capacities of 1.5Gb and 8Gb. For reference, 1.5Gb and 8Gb experiments with (without) JPEG compression store 100,000 (10,000) and 500,000 (50,000) images for replay, respectively, given standard pre-processing. Such capacities are not relevant to Deep SLDA.

Several CSSL variants are tested that *i*) optionally pre-train over the base initialization set and *ii*) may store the replay buffer on disk. For in-memory replay, data is compressed by quantizing each pixel to four bits and resizing images to 75% of their original area. No quantization or resizing is employed when the replay buffer is stored on disk; see Appendix A.1 for details. ImageNet results are provided in Table 3.

**Discussion.** CSSL far outperforms baselines on CI-FAR100. Even at the lowest buffer capacity with a random parameter initialization, our methodology exceeds the performance of REMIND by 3.6% absolute $\Omega_{\text{all}}$ given an equal number of trainable parameters, revealing that *i*) the performance improvement of CSSL is not solely due to having more trainable parameters and *ii*) the proposed approach is capable of achieving impressive performance even with limited memory overhead. When a pre-trained parameter initialization is used, the performance improvement of CSSL is even more pronounced, reaching an absolute $\Omega_{\text{all}}$ of 0.974 that nearly matches offline performance. For comparison, REMIND achieves $\Omega_{\text{all}}$ of 0.790 with an equal number of trainable parameters and unlimited replay capacity.

On ImageNet, the proposed methodology without JPEG compression performs comparably to Deep SLDA at a memory capacity of 1.5Gb. Performance

| Method | Replay Buffer Size | |
|---|---|---|
| | 1.5Gb | 8Gb |
| ExStream | 0.569 | 0.594 |
| Deep SLDA | 0.752 | 0.752 |
| REMIND | 0.855 | 0.856 |
| REMIND + Extra Params | **0.869** | 0.873 |
| CSSL | 0.740 | 0.873 |
| CSSL + Pre-Train | 0.750 | **0.903** |
| CSSL + JPEG | 0.899 | 0.951 |
| CSSL + Pre-Train + JPEG | **0.909** | **0.964** |

Table 3: Top-5 $\Omega_{\text{all}}$ on ImageNet with 1.5Gb and 8Gb buffer capacities. Methods that store the replay buffer on main memory and disk are separated by a horizontal line. REMIND performs well under memory constraints, *but CSSL surpasses REMIND performance as the replay buffer grows in size.*

improves significantly—and surpasses baseline performance—as the replay buffer grows. For example, CSSL without JPEG compression matches or exceeds the performance of all baselines given a replay buffer capacity of 8Gb, and performance continues to improve when a pre-trained parameter initialization is used and as more images are retained for replay. In fact, using pre-trained parameters and 8Gb of memory on disk for replay, the proposed methodology can achieve a Top-5 $\Omega_{\text{all}}$ of 0.964, nearly matching offline training performance.

**What's possible with more memory?** REMIND is capable of storing the replay buffer with limited memory overhead and is the highest-performing method under strict memory limitations; e.g., REMIND achieves the best performance given a 1.5Gb buffer capacity on ImageNet assuming no access to disk storage. Despite the applicability of streaming learning to edge-device scenarios (Hayes et al., 2018; Hayes & Kanan, 2022), many streaming applications exist that do not induce strict memory requirements due to ease of access to high-end hardware on the cloud. Thus, one may begin to wonder whether better performance is possible when memory constraints upon the replay buffer are relaxed.

As outlined in Table 3, REMIND achieves an $\Omega_{\text{all}}$ of 0.856 (0.873 with equal trainable parameters to CSSL) when the entire dataset is stored for replay. Such performance is $> 9\%$ absolute $\Omega_{\text{all}}$ below the best-performing CSSL experiment (i.e., 8Gb capacity with JPEG storage and pre-training), which, despite increased memory overhead relative to REMIND, does not store the whole dataset in the replay buffer. The proposed approach continues improving as the replay buffer becomes larger, reaching a Top-5 $\Omega_{\text{all}}$ of 0.975 given unlimited replay capacity and a pre-trained parameter initialization. Although REMIND may be preferable in memory-constrained scenarios, *the proposed approach continues improving as more memory is allocated for replay, reaching performance levels much closer to that of offline training.*

## 6.2 Different Data Orderings

We perform streaming experiments on the Core50 dataset using i.i.d. (ID), class i.i.d. (CID), instance (I), and class-instance (CI) orderings (Lomonaco & Maltoni, 2017). For each ordering, ten different permutations

| Method | Top-1 $\Omega_{\text{all}}$ without Pre-Training | | | | Top-1 $\Omega_{\text{all}}$ with Pre-Training | | | |
|---|---|---|---|---|---|---|---|---|
| | ID | CID | I | CI | ID | CID | I | CI |
| ExStream | 0.719 | 0.635 | 0.734 | 0.627 | 0.959 | 0.936 | 0.954 | 0.935 |
| Deep SLDA | 0.721 | 0.660 | 0.730 | 0.626 | 0.971 | 0.952 | 0.962 | 0.957 |
| REMIND | 0.719 | 0.628 | 0.741 | 0.600 | 0.985 | 0.971 | 0.986 | 0.973 |
| REMIND + Extra Params | 0.636 | 0.586 | 0.693 | 0.594 | **0.994** | 0.961 | **0.996** | 0.966 |
| CSSL (100Mb Buffer) | 1.106 | 0.977 | 1.071 | 0.990 | 0.963 | 0.979 | 0.962 | 0.976 |
| CSSL (200 Mb Buffer) | 1.107 | 1.005 | 1.104 | 0.995 | 0.968 | 0.981 | 0.977 | 0.976 |
| CSSL (300 Mb Buffer) | **1.153** | **1.009** | **1.112** | **1.024** | 0.976 | **0.986** | 0.974 | **0.979** |

Table 4: Core50 streaming performance for different data orderings, averaged across ten permutations.

are generated, and performance is reported as an average across permutations. The dataset is split into batches of 1200 samples. Experiments are conducted with replay buffer memory capacities of 100Mb (i.e., 2,000 images from Core50, or ⅓ of the full dataset), 200Mb, and 300Mb. These memory capacities only apply to CSSL, while baselines are given unlimited replay capacity. The proposed methodology compresses images using 4-bit integer quantization and resizing to 75% of original area; see Appendix A.2 for details.

Experiments are performed both with and without ImageNet pre-trained weights, allowing performance to be observed in scenarios without a high-quality parameter initialization. In experiments without ImageNet pre-training, baseline methodologies perform base initialization over the first 1200 dataset examples, while CSSL begins streaming from a random initialization. When ImageNet pre-training is used, CSSL begins the streaming process with these ImageNet pre-trained parameters, while baseline methodologies perform base initialization beginning from the pre-trained weights. *CSSL observes no data from the target domain prior to the commencement of streaming.* Core50 results are provided in Table 4.

**Discussion.** Without ImageNet pre-training, our methodology surpasses baseline performance at all memory capacities and often exceeds offline training performance. In this case, $\Omega_{\text{all}} > 1$ indicates that models trained via CSSL outperform offline-trained models during evaluation. This result is likely possible due to the use of simple augmentations (i.e., random crops and flips) for training offline models used to compute $\Omega_{\text{all}}$; see Appendix A.2. Nonetheless, these findings highlight the broad applicability of CSSL. The proposed approach flourishes in this setting because it *i*) has no dependence upon base initialization and *ii*) is capable of learning all model parameters via streaming. In contrast, baseline methodologies struggle with the low-quality base initialization provided by only 1200 data examples.

When initialized with pre-trained weights, baseline methodologies perform much better, though the proposed methodology still matches or exceeds this performance. We emphasize that such pre-trained parameter initializations are not always possible; see Section 4. In numerous practical scenarios (e.g., surgical video, medical imaging, satellite or drone imaging, etc.), well-aligned, annotated public datasets may not exist. *The proposed methodology performs well with or without pre-training due to its end-to-end approach to streaming learning and even surpasses offline training performance when beginning streaming from a cold start.*

### 6.3 Multi-Task Streaming Learning

**The Setting.** In addition to our proposal of CSSL, we propose and explore a new, multi-task streaming learning domain. Although similar setups have been explored for batch incremental learning (Hou et al., 2018), no work has attempted multi-task learning in a streaming fashion, where multiple datasets with disjoint output spaces are learned sequentially, one example at a time. Such a setting is particularly interesting because the introduction of new datasets causes large shifts in properties of the data stream. In such scenarios, the learner must adapt to significant changes in incoming data, making the multi-task streaming learning domain an interesting test case for CSSL.

We consider multi-task streaming learning over several image classification datasets. In particular, we perform multi-task streaming learning over the CUB-200 (Wah et al., 2011), Oxford Flowers (Nilsback & Zisser-

| ImageNet Pre-Training | Method | Top-1 Accuracy | | | |
|---|---|---|---|---|---|
| | | MIT Scenes | CUB-200 | Flowers | FGVC |
| ✓ | Offline | $71.42 \pm 0.60$ | $73.70 \pm 0.01$ | $91.98 \pm 0.02$ | $76.05 \pm 0.17$ |
| | ExStream | $58.96 \pm 0.70$ | $26.04 \pm 0.46$ | $69.16 \pm 0.27$ | $33.33 \pm 0.56$ |
| | Deep SLDA | $\mathbf{60.05 \pm 0.35}$ | $29.85 \pm 0.04$ | $72.56 \pm 0.13$ | $34.16 \pm 0.20$ |
| | REMIND | $54.23 \pm 0.58$ | $46.27 \pm 1.01$ | $78.06 \pm 1.47$ | $36.56 \pm 3.24$ |
| | CSSL | $54.30 \pm 1.95$ | $\mathbf{59.94 \pm 0.97}$ | $\mathbf{88.36 \pm 0.98}$ | $\mathbf{45.29 \pm 2.11}$ |
| ✗ | Offline | $51.60 \pm 0.56$ | $46.95 \pm 0.25$ | $83.10 \pm 0.57$ | $60.12 \pm 0.44$ |
| | ExStream | $31.84 \pm 0.34$ | $9.37 \pm 0.08$ | $40.31 \pm 0.15$ | $16.58 \pm 0.20$ |
| | Deep SLDA | $35.05 \pm 0.19$ | $11.30 \pm 0.10$ | $44.04 \pm 0.09$ | $16.89 \pm 0.06$ |
| | REMIND | $28.56 \pm 0.71$ | $17.97 \pm 0.41$ | $43.97 \pm 1.29$ | $16.53 \pm 0.68$ |
| | CSSL | $\mathbf{42.49 \pm 0.73}$ | $\mathbf{40.08 \pm 0.45}$ | $\mathbf{78.70 \pm 0.45}$ | $\mathbf{25.98 \pm 2.30}$ |

Table 5: Top-1 test accuracy of models trained with different methodologies for multi-task streaming learning. *CSSL is able to better adapt its representations to each of the new datasets over time, resulting in significant performance improvements relative to baselines.*

man, 2008), MIT Scenes (Quattoni & Torralba, 2009), and FGVC-Aircrafts datasets (Maji et al., 2013). The datasets are learned in that order and are introduced to the data stream one example at a time in a class-incremental fashion; see Appendix A.3 for further details. Though the ordering of datasets is never changed, the ordering of data within each dataset may be changed. Comparable experiments use the same ordering of data. To handle the disjoint output spaces of each dataset, models for both the proposed methodology and baselines are modified to have a separate output layer for each dataset.

**Methodology.** For simplicity, all methodologies are given unlimited replay capacity for multi-task streaming learning experiments. Baseline methodologies perform base initialization over the first half of the MIT indoor dataset, optionally beginning with ImageNet pre-trained parameters. CSSL begins the streaming process with either random or ImageNet pre-trained parameters, such that no data from the target domain is observed prior to the commencement of streaming. All tests are performed over three distinct permutations of the data, and performance is averaged across permutations.

During streaming learning, each update performs *i*) an update with replayed and new data for the dataset currently being learned and *ii*) a separate, full update with replayed data for each of the datasets that have been learned previously. We find that such a strategy for multi-task replay performs better than other strategies, such as performing a single update that encompasses all datasets; see Appendix B.5. Deep SLDA and ExStream do not adopt this replay strategy, as only the final classification layers of the model are updated during streaming. Thus, multi-task streaming learning is identical to performing separate streaming experiments on each dataset. Performance in terms of Top-1 test accuracy is recorded separately over each dataset at the end of the streaming process, as shown in Table 5.[10]

**Discussion** As shown in Table 5, CSSL, both with and without pre-triaining, far outperforms all baseline methodologies on nearly all datasets in the multi-task streaming domain. For example, when ImageNet pre-training is used, CSSL outperforms REMIND by 10% absolute test accuracy (or more) on CUB-200, Oxford Flowers, and FGVC-Aircrafts datasets. When no ImageNet pre-training is used, the performance improvement is even more drastic. In particular, CSSL provides a 22% and 35% absolute boost in Top-1 accuracy on the CUB-200 and Oxford Flowers datasets, respectively.

Interestingly, CSSL achieves lower performance than ExStream and Deep SLDA on the MIT Scenes dataset when ImageNet pre-training is used. The impressive performance of these baselines on MIT Scenes is caused by the fact that *i*) ExStream and Deep SLDA only update the model's classification layer during streaming

---

[10]We choose to avoid reporting performance in terms of $\Omega_{\text{all}}$ and only report performance at the end of streaming to match the evaluation strategy of (Hou et al., 2018) and make results less difficult to interpret.

| Dataset | Method | Average ECE |
|---------|--------|-------------|
| CIFAR100 | ExStream | $0.301 \pm 0.008$ |
| | Deep SLDA | $0.296 \pm 0.012$ |
| | REMIND | $\mathbf{0.023 \pm 0.000}$ |
| | CSSL + Pre-Train (30Mb Buffer) | $0.031 \pm 0.005$ |
| | CSSL + Pre-Train (150Mb Buffer) | $\mathbf{0.023 \pm 0.003}$ |
| | CSSL + Pre-Train + No Aug. (150Mb Buffer) | $0.161 \pm 0.031$ |
| | CSSL (30Mb Buffer) | $0.040 \pm 0.001$ |
| | CSSL (150Mb Buffer) | $0.036 \pm 0.002$ |
| | CSSL + No Aug. (150Mb Buffer) | $0.149 \pm 0.015$ |
| ImageNet | REMIND | $\mathbf{0.021 \pm 0.005}$ |
| | CSSL | $0.043 \pm 0.003$ |
| | CSSL + No Aug. | $0.266 \pm 0.009$ |

Table 6: Average ECE of models trained via diferent streaming methodologies on class incremental streaming experiments with CIFAR100 and ImageNet datasets. *Both REMIND and CSSL are found to facilitate favorable calibration properties within the underlying model.*

and *ii*) the fixed feature extractor being used is pre-trained on ImageNet and fine-tuned on over half of the MIT Scenes dataset. Thus, ExStream and Deep SLDA perform well on this dataset only because their feature extractor has been extensively pre-trained and exposed to a large portion of the dataset during base initialization. When ImageNet pre-training is not included, the fixed feature extractor is of lower quality, leading ExStream and Deep SLDA to be outperformed by CSSL even on the MIT Scenes dataset.

Adapting the model's representations to new datasets is pivotal to achieving competitive performance in the multi-task streaming learning domain. Baseline methodologies—due to the fixing of parameters after base initialization—cannot adapt a majority of the model's parameters, leading their representations to be biased towards data observed during base initialization. Such a dilemma emphasizes the importance of developing streaming algorithms, such as CSSL, that learn end-to-end. By using CSSL for multi-task streaming learning, model representations are not fixed based on a subset of data that does not reflect future, incoming datasets, and model performance can benefit from the positive inductive transfer that occurs by updating all model parameters over multiple tasks or datasets simultaneously.

### 6.4 Supplemental Experiments and Analysis

We now present supplemental results that explore the properties of models trained with CSSL, as well as other model architectures.

**Confidence Calibration Analysis.** Confidence calibration characterizes the ability of a model to provide an accurate probability of correctness along with any of its predictions. In many cases, model softmax scores are erroneously interpreted as valid confidence estimates, though it is widely-known that deep networks make incorrect or poor predictions with high confidence (Guo et al., 2017).

Ideally, deep models should be highly-calibrated, as such a property would allow incorrect model predictions to be identified and discarded. For streaming learning, confidence calibration is especially useful, as one must decide during the streaming process whether the model's predictions should start to be

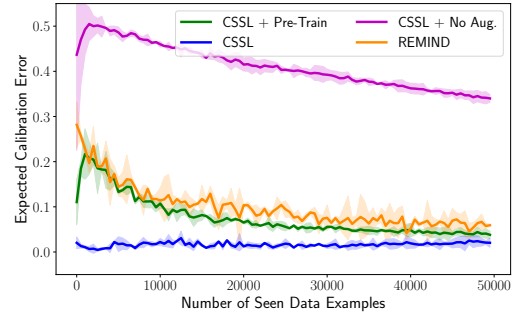

Figure 4: ECE of models trained via REMIND and CSSL on an i.i.d. ordering of CIFAR100. *CSSL maintains better anytime calibration compared to REMIND.*

relied upon. If the underlying model is highly-calibrated throughout streaming, this problem solves itself—just use predictions that are made with high-confidence, indicating a high probability of correctness.

Previous work indicates that using Mixup encourage good calibration properties (Thulasidasan et al., 2019), indicating that models obtained from CSSL and REMIND—both of which use some form of Mixup—should be highly-calibrated. In this section, we measure confidence calibration, in terms of average expected calibration error (ECE) across testing events, of models during class-incremental streaming experiments on CIFAR100 and ImageNet with different streaming methodologies; see Appendix A.4 for further details.

As shown in Table 6, CSSL and REMIND both aid model calibration. Interestingly, the confidence calibration of CSSL degrades significantly if only random crops and flips are used for augmentation (i.e., "CSSL + No Aug."), *highlighting the positive impact of a proper data augmentation policy on calibration.* Using an i.i.d. data ordering, we measure calibration more frequently throughout the streaming process in Figure 4 and find that CSSL is highly-calibrated throughout streaming when beginning from a random initialization and improves upon the calibration of REMIND when beginning from pre-trained parameters.

**Performance Impact of a Cold Start.** To better understand how model performance evolves throughout the streaming process, we perform class-incremental learning experiments on CIFAR100 and measure model accuracy after every new class that is introduced during streaming; see Appendix A.5 for more details.

As shown in Figure 5, baseline methodologies begin with a high accuracy but sharply decline in performance when the model encounters data not included in base initialization and continue to slowly degrade in performance throughout the remainder of the learning process. In contrast, randomly-initialized CSSL begins with relatively poor performance that improves gradually as more data is observed, even-

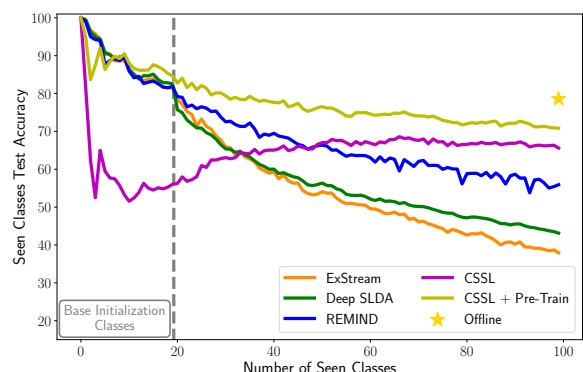

Figure 5: CIFAR100 class-incremental streaming performance measured after each new class is learned.

tually reaching a stable plateau in performance that surpasses all baseline methodologies. When beginning streaming from a pre-trained parameter initialization, this initial period of poor performance is eliminated, and the model still reaches a relatively stable plateau of higher performance.

Baseline methodologies perform well initially because of the data that was observed during base initialization, which is made evident by the sharp drop in baseline performance upon encountering new data; see Figure 5. Notably, CSSL does not experience this drop in accuracy even when beginning from pre-trained parameters, revealing that end-to-end training mitigates bias towards data used for base initialization. Furthermore, the ability of CSSL find a stable plateau in performance reveals that streaming learners need not always decay in performance as the data stream moves further away from data seen during base initialization. With CSSL, model representations can be adapted to changes in data over time to facilitate stable performance.

**Different Network Architectures.** Because CSSL trains the underlying network in an end-to-end fashion and is not dependent upon base initialization, its implementation is simple and agnostic to the particular choice of model architecture (i.e., substituting different architectures requires no implementation changes). Because a majority of prior work only studies ResNet architectures (Hayes et al., 2018; Hayes et al., 2020), we use the proposed methodology to study the behavior of ResNet101

| Model | Top-5 $\Omega_{\text{all}}$ |
|---|---|
| ResNet101 | 0.872 |
| MobileNetV2 | 0.904 |
| DenseNet121 | 0.898 |
| Wide ResNet50-2 | 0.880 |

Table 7: Class-incremental streaming performance with CSSL and various model architectures. *CSSL performs competitively with all architectures.*

(He et al., 2016), MobileNetV2 (Sandler et al., 2018), DenseNet121 (Huang et al., 2017), and Wide ResNet50-2 (Zagoruyko & Komodakis, 2016) architectures in class-incremental streaming experiments on ImageNet; see Appendix A.6 for further details. The replay buffer is stored on disk using JPEG compression with a

memory capacity of 1.5Gb. The results of these experiments, recorded in terms of Top-5 $\Omega_{\text{all}}$, are provided in Table 7, where it can be seen that CSSL achieves competitive performance with each of the different model architectures.

## 7 Conclusion

We present CSSL, a new approach to streaming learning with deep neural networks that trains the network end-to-end and uses a combination of replay mechanisms with sophisticated data augmentation to prevent catastrophic forgetting. Because the underlying network is learned end-to-end with CSSL, no base initialization is required prior to streaming, yielding a single-stage learning approach that is both simple and performant. In experiments, CSSL is found to surpass baseline performance—and even match or exceed offline training performance—on numerous established experimental setups, as well as on a multi-task streaming learning setup proposed in this work. We hope that CSSL inspires further developments in deep streaming learning by demonstrating the surprising effectiveness of simple learning techniques.

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

# A    Experimental Details

## A.1    Class-Incremental Learning

Here, we provide all details for the class-incremental streaming learning experiments presented in Section 6.1. For both CIFAR100 and ImageNet, the ordering of data (i.e., both by class and by example) is fixed for all methodologies, such that data is always encountered in the same order.

**Offline Training Details.** For CIFAR100, Top-1 $\Omega_{\text{all}}$ is computed using an offline-trained ResNet18 model that achieves a Top-1 accuracy of 78.61%. This model is trained using standard data augmentation for

CIFAR100 (i.e., random crops and flips) using the SGD optimizer with momentum of 0.9 and an initial learning rate of 0.1. The learning rate is decayed throughout training using a cosine learning rate decay schedule over 200 total epochs. A weight decay of $5 \times 10^{-4}$ is used. These offline training settings are adopted from a widely-used repository for achieving state-of-the-art performance on CIFAR10 and CIFAR100 (Liu, 2017).

Similarly, Top-5 $\Omega_{\text{all}}$ on ImageNet is computed using an offline-trained ResNet18 model that achieves a Top-5 accuracy of 89.09%. This pre-trained model is made publicly available via torchvision (Paszke et al., 2017) and matches the offline normalization model used to evaluate class-incremental streaming learning on ImageNet in previous work (Hayes et al., 2020).

**Data Details.** On CIFAR100, the dataset is divided into batches of 20 classes, where the first 20 classes are reserved for base initialization. Similarly, ImageNet is divided into batches of 100 classes, and the first 100 classes are used for base initialization. Evaluation occurs after each batch of data in the streaming process, and all testing events are aggregated within the $\Omega_{\text{all}}$ score. For all baselines, standard test augmentations for both CIFAR100 and ImageNet are adopted. Namely, because the feature extractors of ExStream, Deep SLDA, and REMIND are fixed, we perform appropriate resizing, center cropping, and normalization (i.e., following standard test settings of CIFAR100 and ImageNet) prior to passing each image into the fixed feature extractor. REMIND also leverages random resized crops and manifold Mixup (Verma et al., 2019) on the feature representations stored within the replay buffer throughout streaming. The proposed methodology leverages the data augmentation policy described in Section 3, and images are cropped and normalized following standard practice prior to augmentation.

**Baseline Details.** For ExStream, Deep SLDA, and REMIND, we utilize official, public implementations within all experiments (Hayes, 2019; 2020b;a). ExStream is optimized using Adam with a learning rate of $2 \times 10^{-3}$ and no weight decay, as in the official implementation. For CIFAR100, we modify the number of class exemplars (i.e., 10K, 20K, ..., 50K exemplars) to study ExStream's behavior with different replay buffer sizes. For ImageNet, we follow the settings of (Hayes et al., 2020) and train ExStream using 20 and 100 exemplars per class for buffer capacities of 1,5Gb and 8Gb, respectively. REMIND uses the SGD with momentum optimizer with a learning rate that decays from 0.1 to 0.001 from the beginning to the end of each class. Momentum is set to 0.9 and weight decay to $1 \times 10^{-5}$. REMIND samples 100 and 50 replay samples during each online update for CIFAR100 and ImageNet, respectively. For Deep SLDA, we adopt the variant that does not fix the covariance matrix during the streaming process for all experiments. For REMIND, we perform experiments with a version that exactly matches the settings of the original papers, as well as with a version that adds two extra layers to the trainable portion of the ResNet18 model such that the number of trainable parameters between REMIND and CSSL is identical. All other experimental settings for each of the baseline methodologies exactly match the settings within each of the respective papers and/or public implementations (Hayes, 2019; 2020b;a).

All baseline methodologies perform base initialization over the first batch of data within the dataset. This base initialization includes 50 epochs of tine-tuning, followed by the initialization of any algorithm-specific modules to be used during the streaming process (e.g., the PQ module for REMIND). For fine-tuning, we adopt the same hyperparameter settings used for training the offline baseline models. After base initialization, streaming begins at the first class within the base initialization batch, meaning that data used within base initialization is re-visited during streaming. For ExStream and Deep SLDA, models are initialized with pre-trained ImageNet weights during base initialization on CIFAR100 experiments to enable more comparable experimental results on the smaller dataset.

**CSSL Details.** The proposed methodology is trained with a SGD optimizer with momentum of 0.9 and a learning rate that decays linearly from 0.1 to 0.001 from the first to last example of each class (i.e., the same learning rate decay strategy as (Hayes et al., 2020) is adopted). We use weight decay of $1 \times 10^{-5}$. 100 replay samples are used within each online update for both CIFAR100 and ImageNet. We utilize the augmentation strategy described in Section 3 in all experiments. For Mixup and Cutmix, we utilize $\alpha$ values of 0.1 and 0.8, respectively. For AutoAugment, we adopt the CIFAR and ImageNet learned augmentation policies for each of the respective datasets. See Appendix B.1 for details regarding the derivation of the optimal augmentation policy and associated hyperparameters. For experiments that begin with a pre-trained parameter setting,

we use identical training settings as used in base initialization for baseline methodologies to train model parameters over the first 20 classes of CIFAR100 or 100 classes of ImageNet, then use the resulting model parameters to initialize CSSL for streaming.

## A.2   Different Data Orderings

Here, we overview the experimental details of the experiments performed using the Core50 dataset (Lomonaco & Maltoni, 2017) in Section 6.2.

**Data Orderings.** Experiments are performed using four different data orderings: i.i.d. (ID), class i.i.d. (CID), instance (I), and class-instance (CI). Such data orderings are described in detail in (Lomonaco & Maltoni, 2017), but we also provide a brief description of each ordering scheme below:

- `i.i.d.`: batches contain a uniformly random selection of images from the dataset.
- `class i.i.d.`: batches contain all images from two classes (i.e., out of ten total classes) in the dataset.
- `instance`: batches contain all images, in temporal order, corresponding to 80 unique object instances within the dataset.
- `class-instance`: batches contain images, in temporal order, corresponding to two classes in the dataset.

For each of the four possible ordering, ten unique data permutations are generated. Then, all experiments are repeated over each of these permutations, and performance is recorded for each ordering scheme as an average over all possible permutations.

**Offline Training Details.** Top-1 $\Omega_{\text{all}}$ on Core50 is computed using an offline-trained ResNet18 model that achieves a Top-1 accuracy of 43.91%. This model is trained for 40 epochs from a random intialization using SGD with momentum of 0.9 and an initial learning rate of 0.01. We use a weight decay of $1 \times 10^{-4}$ and decay the learning rate by $10\times$ at epochs 15 and 30 during training. The settings of training our offline model for Core50 exactly match those provided in (Hayes et al., 2020), aside from not initializing the model with pre-trained ImageNet weights. We attempted to improve the performance of the offline model by utilizing a greater number of epochs and modifying hyperparameter settings, but such tuning resulted in only negligible performance improvements.

**Data Details.** We sample the Core50 dataset at 1 frame per second, resulting in 600 and 225 training and testing images for each class (Hayes et al., 2018). The Core50 dataset has 10 total classes, resulting in 6000 total training images and 2250 total testing images. We adopt the same bounding box crops and splits from (Lomonaco & Maltoni, 2017). The dataset is split into batches of 1200 examples, where the content of each batch is determined by the ordering scheme and particular data permutation chosen for a particular experiment. Comparable experiments utilize both the same ordering scheme and the same permutation, where 10 unique permutations exist for each ordering scheme. The first batch is utilized for base initialization, and evaluation occurs after each batch to compute the final $\Omega_{\text{all}}$ score. To match the settings of (Hayes et al., 2020), we do not utilize data augmentation within any of the baseline methodologies (i.e., adding data augmentation was shown to degrade performance on Core50). The proposed methodology utilizes the same data augmentation policy that is described in Section 3.

**Baseline Details.** Again, we utilize official, public implementations of ExStream, Deep SLDA, and RE-MIND (Hayes et al., 2018; Hayes & Kanan, 2020; Hayes et al., 2020). ExStream is optimized using Adam with a learning rate of $2 \times 10^{-3}$ and no weight decay, and the memory buffer is allowed to store the full dataset (i.e., this can be done with $< 100$Mb of memory). REMIND is trained using SGD with momentum of 0.9 and a fixed learning rate of 0.01. A weight decay of $1 \times 10^{-4}$ is used and 20 samples are taken during each online update, which matches settings of (Hayes et al., 2020). For Deep SLDA, we again adopt the variant that does not fix the covariance matrix during the streaming process. All other experimental settings match the hyperparameters provided in the respective papers and/or public implementations (Hayes, 2019; 2020b;a). No data augmentation is used during baseline streaming experiments, as outline in (Hayes et al., 2020).

The first batch of data is used for base initialization within all streaming baseline methodologies. During base initialization, the fine-tuning procedure again adopts the same hyperparameters as the offline training

| Dataset | Classes | Training Examples | Testing Examples |
|---------|---------|-------------------|------------------|
| CUB-200 | 200 | 5994 | 5794 |
| Oxford Flowers | 102 | 2040 | 6149 |
| MIT Scenes | 67 | 5360 | 1340 |
| FGVC-Aircrafts | 100 | 6667 | 3333 |

Table 8: Details of datasets used for multi-task streaming learning experiments.

procedure described above. After the network has been fine-tuned over base intialization data, all algorithm-specific modules are initialized using the same data, and streaming begins from the first class of the base initialization step. We do not utilize pre-trained ImageNet weights to initialize baseline model parameters within Core50 experiments, as we are attempting to accurately assess model performance in low-resource scenarios on Core50 experiments.

**CSSL Details.** The proposed methodology is trained with SGD with momentum of 0.9 and a fixed learning rate of 0.01. We use a weight decay of $1 \times 10^{-4}$. 100 replay samples are observed during each online update, and we adopt the data augmentation policy described in Section 3. For AutoAugment, we utilize the ImageNet augmentation policy, and $\alpha$ values of 0.1 and 0.8 are again adopted for Mixup and CutMix, respectively.

The proposed methodology is tested with replay buffer capacities of 100, 200, and 300Mb. For the 200Mb experiment, image pixels are quantized to four bits, while for the 100Mb experiment we both quantize pixels and resize images to 75% of their original area. The 300Mb experiment performs no quantization or resizing, as the full dataset can be stored as raw images with memory overhead slightly below 300Mb.

### A.3 Multi-Task Streaming Learning

Here, we overview the details of the multi-task streaming learning experiments in Section 6.3.

**Datasets.** We perform multi-task streaming learning with the CUB-200 (Wah et al., 2011), Oxford Flowers (Nilsback & Zisserman, 2008), MIT Scenes (Quattoni & Torralba, 2009), and FGVC-Aircrafts datasets (Maji et al., 2013). The details of each of these datasets are provided in Table 8. Following the settings of (Hou et al., 2018), the datasets are learned in the following order: MIT Scenes, CUB-200, Oxford Flowers, then FGVC-Aircrafts.

**Offline Training Details.** The offline trained models for multi-task streaming experiments are trained separately for each of the datasets listed in Table 8. For each dataset, offline models are trained for 50 total epochs with cosine learning rate decay. Initial learning rates of 0.1, 0.01, and 0.001 are tested and results of the best-performing model are reported in Section 6.3. We also test baseline models with step learning rate schedules but find that cosine learning rate decay tends to produce models with better performance.

**Data Details.** For multi-task streaming learning performance, testing is performed after each dataset has been observed. We choose to not perform more regular evaluation (e.g., after subsets of each individual dataset have completed streaming) to make performance simple to interpret. All results are reported in terms of Top-1 accuracy computed separately on each dataset at the end of the streaming process. During streaming, each dataset is observed one example at a time in a class-incremental order. Three separate class-incremental data permutations are generated, and results are averaged across these three different permutations. The order of the datasets—presented above—is kept fixed in the three different permutations, as we only shuffle class and example orderings.

Because the feature extractors of ExStream, Deep SLDA, and REMIND are fixed, we perform appropriate resizing, center cropping, and normalization prior to passing each image into the fixed feature extractor. REMIND also leverages random resized crops and manifold mixup (Verma et al., 2019) on the feature representations stored within the replay buffer throughout streaming. The proposed methodology leverages the data augmentation policy described in Section 3, and images are cropped and normalized following

standard practice prior to augmentation. Baselines utilize standard training and testing augmentations as used on ImageNet, which matches the settings of (Hou et al., 2018).

**Baseline Details.** Again, official, public implementations of baseline methodologies with the ResNet18 architecture are used in all experiments. For baseline methodologies, base initialization is performed over 50% of the MIT scenes dataset (i.e., the first dataset to be observed during streaming), encompassing 34 of the 67 total classes. All baseline methodologies are allowed unlimited memory capacity for replay. ExStream is optimized using Adam with a learning rate of $2 \times 10^{-3}$ and no weight decay. REMIND uses the stochastic gradient descent (SGD) optimizer with a learning rate that decays from 0.1 to 0.001 from the beginning to the end of each class. Momentum is set to 0.9 and weight decay to $1 \times 10^{-5}$. REMIND samples 50 examples from the replay buffer for each online update. Furthermore, we modify REMIND's replay strategy to sample a batch of replayed data from each of the seen tasks when performing each online update. Such a change is necessary to not forget previous tasks when learning each new dataset, and we do not enforce any memory capacity on the replay buffer (i.e., all data can be kept for replay). For Deep SLDA, we adopt the variant that does not fix the covariance matrix during the streaming process for all experiments.

All baseline methodologies perform base initialization over the first 34 classes of the MIT scenes dataset. This base initialization includes 50 epochs of tine-tuning, followed by the initialization of any algorithm-specific modules to be used during the streaming process (e.g., the PQ module for REMIND). For fine-tuning, we adopt the same hyperparameter settings used for training the offline baseline models. After base initialization, streaming begins at the first class within the base initialization batch, meaning that data used within base initialization is re-visited during streaming. For each of the baseline methodologies, we perform experiments both with and without pre-training on ImageNet. Experiments with pre-training simply initialize the underlying ResNet18 architecture with ImageNet pre-trained weights prior to performing base initialization.

**CSSL Details.** The proposed methodology is trained with an SGD optimizer with momentum of 0.9 and a learning rate that decays linearly from 0.1 to 0.001 from the beginning to the end of each class. Weight decay of $1 \times 10^{-5}$ is used. For all datasets, 100 replay samples are taken at each online update, and we modify the replay strategy to sample a batch of replayed data from each of the seen tasks when performing an online update. The augmentation strategy described in 3 is used in all experiments. Mixup and Cutmix use $\alpha$ values of 0.1 and 0.8, respectively, and AutoAugment adopts the Imagenet learned augmentation policy. See Appendix B.1 for details regarding the derivation of the optimal augmentation strategy. For all experiments, CSSL is fine-tuned on the first 34 classes of the MIT scenes dataset with identical hyperparameters as are used for base initialization with other methodologies. Some experiments utilize ImageNet pre-training and are identified as such.

## A.4 Confidence Calibration Analysis

Here, we present experimental details for the confidence calibration analysis performed in Section 6.4.

**Expected Calibration Error.** Confidence calibration is measured in terms of expected calibration error (ECE). Assume that the $i$-th example within the testing set is associated with a true label, a model prediction, and a confidence value. Additionally, assume $N$ total examples exist within the test set. Given $N_{\text{bin}}$ bins, ECE groups data into uniformly-spaced bins based on confidence values. For example, if $N_{\text{bin}} = 2$, all predictions are separated into two groups, one group within confidence values in the range $[0.0, 0.5)$ and another with confidence values in the range $[0.5, 1.0]$.

Denote the $i$-th bin as $\text{bin}_i$. Once model predictions are separated into such bins, ECE computes the average accuracy and confidence of predictions within each bin. Then, using $a_i$ and $c_i$ to denote the accuracy and average confidence within the $i$-th bin, ECE can be computed as shown in the equation below.

$$\text{ECE} = \sum_{i=1}^{N_{\text{bin}}} \frac{|\text{bin}_i|}{N} \cdot |a_i - c_i|$$

| Model | Top-5 Acc. |
|---|---|
| ResNet101 | 93.54% |
| MobileNetV2 | 90.29% |
| DenseNet121 | 91.98% |
| Wide ResNet50-2 | 94.09% |

Table 9: Top-5 accuracies achieved by offline models of different architectures on ImageNet.

For all experiments within Section 6.4, we follow the settings of (Guo et al., 2017) and set $N_{bin} = 15$. Performance is sometimes reported in terms of average ECE. In such cases, average ECE is computed by measuring ECE separately at each testing event during the streaming process, then taking a uniform average of ECE values across all testing events.

**CIFAR100 Experiments.** For the CIFAR100 experiments reported in Table 6, we perform class-incremental streaming using the same data ordering described in Appendix A.1. CSSL experiments use two different memory capacities: 30Mb (10K CIFAR100 images) and 150Mb (full CIFAR100 dataset). The version of CSSL denoted as "No Aug." replaces the CSSL augmentation strategy outlined in Section 3 with a simple augmentation policy that performs only random crops and flips. Furthermore, the pre-trained variant of CSSL simply initializes model parameters with pre-trained ImageNet weights at the beginning of the streaming process. All other experimental settings are identical to that of the class incremental learning experiments outlined in Appendix A.1.

For the streaming experiment depicted in Figure 4, we perform streaming with an i.i.d. ordering of the CIFAR100 dataset. Because the data ordering is i.i.d., all testing events perform evaluation over the entire CIFAR100 testing dataset. For CSSL, a memory capacity of 150Mb is used for the replay buffer. All experiments with CSSL utilize the augmentation strategy described in Section 3, except for the "No Aug." experiment that utilizes only random crops and flips for data augmentation. REMIND is allowed to store the full dataset for replay. All other experimental settings match those of the class-incremental learning experiments for CIFAR100 described in Appendix A.1.

**ImageNet.** For the confidence calibration experiments on ImageNet presented in Table 6, we utilize a class-incremental data ordering for streaming. For CSSL, we use a 1.5Gb replay buffer capacity and store all images with JPEG compression on disk. For the "No Aug." variant of CSSL, we again use random crops and flips for data augmentation instead of the augmentation strategy described in Section 3. All other settings for these experiments are identical to those of the class incremental learning experiments on ImageNet described in Appendix A.1.

## A.5 Performance Impact of a Cold Start

The experiments in Section 6.4 utilize the same experimental setup for CIFAR100 as described in Section 6.1 and Appendix A.1. All results are recorded in terms of Top-1 $\Omega_{all}$ using the same offline normalization model from 6.1 that achieves a Top-1 accuracy of 78.61%. The only difference between the two experimental setups is that model accuracy is recorded after every new class is observed during the streaming process, whereas experiments in Section 6.1 only record model accuracy after every 20-class batch. Within Section 6.4, both the proposed methodology and all baselines are allowed to retain the full dataset within the replay buffer. The proposed methodology does not perform and quantization or resizing on the images. All hyperparameter settings are the same as Appendix A.1 for all methodologies.

## A.6 Different Network Architectures

The experiments with different network architectures in Section 6.4 follow the same settings as the class-incremental streaming experiments on ImageNet provided in Section 6.4; see Appendix A.1 for further details. However, the ResNet18 architecture used in previous experiments is substituted for several different model architectures. The experimental results presented in Section 6.4 store the replay buffer on disk using JPEG

| MU | CM | AA | RA | Top-1 $\Omega_{\text{all}}$ |
|----|----|----|----|-----------------|
|    |    |    |    | $0.689 \pm 0.001$ |
| ✓  |    |    |    | $0.807 \pm 0.012$ |
|    | ✓  |    |    | $0.863 \pm 0.020$ |
| ✓  | ✓  |    |    | $0.877 \pm 0.004$ |
|    |    | ✓  |    | $0.822 \pm 0.037$ |
| ✓  |    | ✓  |    | $0.886 \pm 0.018$ |
|    | ✓  | ✓  |    | $0.852 \pm 0.019$ |
| ✓  | ✓  | ✓  |    | $\mathbf{0.886 \pm 0.002}$ |
| ✓  |    |    | ✓  | $0.856 \pm 0.025$ |
|    | ✓  |    | ✓  | $0.857 \pm 0.008$ |
| ✓  | ✓  |    | ✓  | $0.875 \pm 0.005$ |

Table 10: Performance of the proposed methodology for CSSL with numerous augmentation policies that combine Mixup (MU), CutMix (CM), AutoAugment (AA), and RandAugment (RA) for class-incremental streaming experiments on CIFAR100. All tests use random clops and flips, and ✓ indicates the use of a particular augmentation technique. *The best performance is achieved with a combination of Mixup, Cutmix, and AutoAugment.*

compression, and a replay buffer capacity of 1.5Gb is adopted. All results are reported in terms of Top-5 $\Omega_{\text{all}}$, and the offline models used for normalization (i.e., pre-trained models taken from the torchvision package (Paszke et al., 2017)) achieve Top-5 accuracies listed in Table 9

### A.7   Closing the Performance Gap with Offline Learning

Here, we present the details of experiments performed in Appendix B.6. Separate comparison are performed between incremental learning methodologies (i.e., iCarl and E2EIL) and streaming learning methodologies (i.e., REMIND) for class-incremental learning on the CIFAR100 dataset. In all cases, we use a ResNet18 model, but the architecture is slightly modified to yield best-possible performance on the CIFAR100 dataset, as demonstrated in (Liu, 2017). Within these experiments, a class-specific ordering of the CIFAR100 dataset is induced that is used to ensure all experiments receive data in the same order. In all cases, the first 20 classes of CIFAR100 are used for base initialization.

For REMIND, we use the public implementation, but adapt it to utilize the modified ResNet18 architecture and perform training over CIFAR100. The learning rate is decayed from 0.1 to 0.001 for each class during streaming (i.e., we perform tests with different learning rates and fine that 0.1 still performs best), and we leverage random crops and manifold mixup within REMIND's feature representations. We allow all dataset examples to be stored within the replay buffer, due to the memory-efficiency of REMIND's memory indexing approach. Settings for PQ are kept identical to ImageNet experiments. We find that weight decay of $5 \times 10^{-4}$ yields the best performance when evaluated using grid search over a validation set.

For iCarl and E2EIL, we adopt the official, public implementations of both approaches. Because the CIFAR100 dataset is small, we allow both iCarl and E2EIL store the full dataset within the replay buffer (i.e., we do not enforce a fixed buffer size) during the online learning process. To tune hyperparameters in the streaming setting, we perform a grid search over a validation set of CIFAR100 to obtain the optimal settings. We train iCarl and E2EIL in the typical, incremental fashion so that the impact of class imbalance can be observed. Within the incremental learning process, we use 20-class subsets of CIFAR100 as each sequential batch during the learning process. From here, we adopt identical hyperparameter settings as the original publication to replicate the original experimental settings.

| Quant. | Resize Ratio | Top-1 $\Omega_{\text{all}}$ |
|--------|--------------|-----------------------------|
| 8 bit | 100% | $0.810 \pm 0.009$ |
| 6 bit | 100% | $0.822 \pm 0.014$ |
| 4 bit | 100% | $0.813 \pm 0.025$ |
| 8 bit | 66% | $0.819 \pm 0.001$ |
| 8 bit | 50% | $0.764 \pm 0.029$ |
| 6 bit | 66% | $\mathbf{0.826 \pm 0.002}$ |
| 4 bit | 66% | $0.715 \pm 0.038$ |

Table 11: Performance of the proposed methodology on class-incremental streaming experiments with CI-FAR100, where different levels of quantization and resizing are used when adding examples into the replay buffer.

# B  Further Experimental Results

## B.1  Data Augmentation Policies

We test all combinations of the Mixup, Cutmix, AutoAugment, and RandAugment data augmentation strategies to arrive at the final, optimal augmentation policy used within the proposed methodology for CSSL. All experiments perform random crops and flips in addition to other augmentation strategies. The comparison of different data augmentation policies is performed with class-incremental learning experiments on CIFAR100. These experiments adopt identical experimental settings as CIFAR100 experiments in Section 6.1; see Appendix A.1 for further details.

For these experiments, the entire dataset is allowed to be stored in the replay buffer, and the replay buffer is kept in main memory without the use of any compression schemes (i.e., no integer quantization or resizing). All reported results represent average performance recorded over three independent trials. For Mixup and Cutmix, we perform experiments using all combinations of $\alpha$ values in the set $\{0.1, 0.4, 0.8, 1.0, 1.2\}$. We leverage the CIFAR100 AutoAugment policy, and experiments are performed with RandAugment using 1-2 augmentations and all magnitudes in the set $\{4, 9, 14\}$. All combinations of considered data augmentation methodologies are tested, and results with best-performing hyperparameters are presented. These results inform the data augmentation policy that is adopted within the proposed methodology for CSSL used in later experiments.

The results of these experiments, reported in terms of Top-1 $\Omega_{\text{all}}$, are provided in Table 10, where it can be seen that the best results are achieved by combining Mixup, Cutmix, and AutoAugment techniques into a single augmentation policy. Interestingly, this augmentation policy provides a nearly 20% improvement in absolute $\Omega_{\text{all}}$ compared to a simple crop and flip augmentation strategy, *thus demonstrating the massive impact of proper data augmentation on streaming performance.*

## B.2  Memory Efficient Replay.

The proposed methodology for CSSL compresses replay examples using integer quantization (i.e., on the `uint8` pixels of each image) and image resizing. Within this section, we analyze the performance impact of this strategy in class-incremental streaming experiments on CIFAR100 using the same experimental setup as outlined in Section 6.1. All combinations of quantizing `uint8` pixels to six or four bits[11] and resizing images to 66% or 50% of their original area are tested. For each possible augmentation strategy, the corresponding experiment stores the maximum number of images within the replay buffer without exceeding 30Mb of storage. Three separate trails are performed for each possible compression strategy, and average results across the three trials are reported. These results inform the optimal strategy of reducing memory overhead for the replay mechanism used within the proposed methodology for CSSL.

---

[11]The integer quantization procedure is simulated by integer dividing each pixel value by $2^{8-b}$, where $b$ is the number of bits present in the quantized pixel values, then re-scaling the result into the range $[0, 256)$.

| Eviction Strategy | Top-1 $\Omega_{\text{all}}$ |
|---|---|
| Mixup (Zhang et al., 2017) | $0.746 \pm 0.019$ |
| CutMix (Yun et al., 2019) | $0.778 \pm 0.014$ |
| SamplePairing (Inoue, 2018) | $0.735 \pm 0.023$ |

Table 12: Performance of CSSL when various interpolation strategies are used to combine replay examples instead of evicting them. *These eviction policies are found to perform worse than the quantization and resizing strategy used within the main experiments.*

| Buffer Capacity | Random | Reservoir |
|---|---|---|
| 30Mb | $0.810 \pm 0.019$ | $0.709 \pm 0.099$ |
| 60Mb | $0.855 \pm 0.011$ | $0.764 \pm 0.010$ |
| 90Mb | $0.889 \pm 0.024$ | $0.795 \pm 0.004$ |
| 120Mb | $0.874 \pm 0.009$ | $0.830 \pm 0.001$ |

Table 13: Comparison of different strategies for maintaining the replay buffer for class-incremental CSSL experiments on CIFAR100. *A random eviction strategy is shown to outperform reservoir sampling for all different replay buffer capacities that were considered.*

Adopting a fixed buffer capacity of 30Mb, we explore different levels of quantization and resizing within the replay buffer.[12] The results are reported in terms of Top-1 $\Omega_{\text{all}}$ in Table 11, where it can be seen that the best performance is achieved by quantizing pixel values to 6 bits and resizing images to 66% of their original area (i.e., the exact strategy adopted for CIFAR100 experiments in Section 6.1).

Interestingly, the proposed methodology is robust to high levels of quantization; e.g., using 4 bit quantization slightly exceeds the performance of storing full images for replay. Similar behavior is observed on ImageNet, where 4-bit quantization is found to outperform 6-bit quantization given a fixed buffer capacity. Moreover, resizing images beyond 66% of their original area noticeably degrades performance; e.g., 50% resizing without quantization yields noticeably degraded performance in Table 11. For datasets with higher-resolution images, we observe that performance is even more sensitive to the resizing ratio, leading us to adopt a ratio of 75% within ImageNet and Core50 experiments in Sections 6.1 and 6.2.

### B.3 Consolidating Replay Examples

In addition to the approaches for memory efficient replay described in Section 3, we attempt to use data interpolation methods to consolidate examples within the replay buffer. In particular, when a replay example is about to be evicted from the replay before, interpolation is performed between two replay examples, thus consolidating two entrties in the replay buffer into one. Such an approach avoids evicting examples within the replay buffer without increasing memory overhead. In particular, we perform experiments with Mixup, CutMix, and SamplePairing interpolation strategies[13] for class-incremental learning on the CIFAR100 dataset, adopting the same experimental setting as described in Section 6.1. We use a fixed replay buffer capacity of 30Mb and report results of these experiments in terms of Top-1 $\Omega_{\text{all}}$ within Table 12. As can be seen, these eviction strategies are outperformed by the quantization and resizing strategies utilized in Section 6.4. We also consider setting a fixed number of times a certain example can be interpolated, such that a replay examples is evicted after it has been interpolated more than two or three times, but such an approach still does not meaningfully improve performance and introduces an extra hyperparameter.

---

[12]When images are compressed, more images are added into the replay buffer to reach the 30Mb capacity.

[13]The $\alpha$ values used for Mixup and Cutmix are tuned by searching over the set $\{0.1, 0.4, 0.8, 1.0, 1.2\}$ using a hold-out validation set, and we present the results of the best-performing $\alpha$ for each method. By definition, Sample Pairing always takes an average between two images (i.e., there are no hyperparameters to tune).

| Full | Split | Sep | Sum | MIT Scenes | Top-1 Accuracy | | | Average Top-1 |
| :---: | :---: | :---: | :---: | :---: | :---: | :---: | :---: | :---: |
| | | | | | CUB-200 | Flowers | FGVC | Accuracy |
| ✓ | | ✓ | | **56.11** | 60.79 | **88.08** | 45.22 | **62.55** |
| | ✓ | ✓ | | 53.13 | 57.232 | 84.50 | 36.78 | 57.91 |
| ✓ | | | ✓ | 54.40 | **62.12** | 84.39 | **49.09** | 62.50 |
| | ✓ | | ✓ | 54.48 | 60.32 | 86.57 | 43.05 | 61.11 |

Table 14: Multi-task streaming learning performance of CSSL with different replay strategies.

### B.4 Alternative Sampling Approaches

The methodology described in Section 3 assumes that, when a new example is added into an already-full replay buffer, an example must be removed via `ReplayEvict` in Algorithm 1. Within the experiments in Section 6, a random eviction procedure is adopted as described in Section 3. However, such random maintenance of the replay buffer is not the only option. Namely, different approaches such as herding selection (Rebuffi et al., 2017; Castro et al., 2018; Hou et al., 2019) or reservoir sampling (Chaudhry et al., 2019; Aljundi et al., 2019a) could be adopted instead to add and remove elements from the replay buffer in a more sophisticated manner.

Although herding selection is a popular method in incremental learning literature (Rebuffi et al., 2017; Castro et al., 2018; Hou et al., 2019) for determining which examples within each incremental batch should be added into the replay buffer, it is not a viable approach given the restrictions of CSSL. Namely, herding selection requires that, for each class, a mean feature vector should be computed across all available data. Then, a representative subset of exemplars is selected within that class, such that the mean feature vector of that subset closely matches the overall mean feature vector. Such an approach is easily adoptable within incremental learning, as one can compute mean feature vectors and relevant exemplars for the disjoint classes that appear within each incremental batch. Further, herding selection can be modified to the streaming setting (i.e., based on a greedy approach) if fixed feature representations are available for each input image throughout the streaming process. However, within CSSL, the entire network is updated during streaming, meaning that the feature respresentations of data change over time, making the application of herding selection difficult.

Despite the difficulty of adopting herding selection within CSSL, we compare the performance of our random eviction policy with reservoir sampling with class-incremental learning experiments on the CIFAR100 dataset, where we adopt the same setup as described in Section 6.1. Reservoir sampling adds items into the replay buffer until the buffer is full. Assuming the capacity of the replay buffer is $C$ examples, once the $k$-th example is received (i.e., $k > C$), we sample a random integer $i$ in the set $\{0, 1, \ldots, k-1\}$. If $i \in \{0, 1, \ldots, k-1\}$, then the $i$-th example in the replay buffer is replaced with the $k$-th example (i.e., the new data). Otherwise, the $k$-th example is discarded. The performance of reservoir sampling in comparison to the random eviction strategy described in Section 3 is provided in Table 13, where we measure performance across multiple different buffer capacities (i.e., 150Mb is full capacity for CIFAR100). As can be seen, the more sophisticated reservoir sampling approach is outperformed by the random eviction strategy outlined in Section 3.

### B.5 Different Strategies for Multi-Task Streaming Learning

Within Section 6.3, we generalize the replay strategy of CSSL and REMIND to the multi-task streaming learning domain. In this domain, each model has multiple classification layers and replay buffers (i.e., one for each dataset).[14] Thus, the previously-used strategy for replay—sampling a fixed number of replay samples to be included with each new data example during an online update—is no longer valid, as one must incorporate replay examples from previous tasks to avoid catastrophic forgetting.

---

[14]The separate replay buffers for each dataset could be considered as one replay buffer than encompasses all datasets and the resulting algorithm and discussin would be the same.

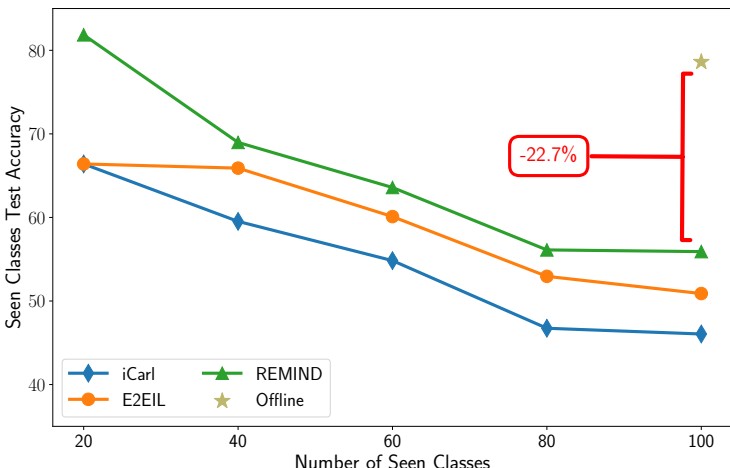

Figure 6: Top-1 test accuracy for ResNet18 trained on CIFAR100 with both online and offline approaches. Accuracy is measured over "seen" classes at different points during online learning.

The main strategies that were explored for multi-task replay were differentiated by $i$) whether each task is updated with $B$ replay examples or if $B$ total replay examples are split between each task (i.e., Full or Split) and $ii$) whether each task performs a separate update or if gradients are summed over all tasks to perform a single update (i.e., Sep or Sum). Considering these options forms four different strategies for multi-task replay. Using a fixed replay size of $B = 100$ and adopting the experimental settings explained in Appendix A.3 for multi-task streaming learning, these different strategies for multi-task streaming learning are explored for CSSL in Table 14. Models are initialized with ImageNet pre-trained parameters, but we do not perform any fine-tuning on the MIT Indoor dataset prior to streaming.

As can be seen in Table B.5, the best performing strategy utilizes the full replay size for each task during the online update and performs a separate model update for each task. A strategy of using the full replay size for each task but performing a single update that sums the gradients of each task together performs similarly. However, this strategy is slightly worse than performing a separate update for each task. Thus, within Section 6.3, we adopt the strategy of, at each online update, sampling $B$ replay examples—or $B - 1$ replay examples for the current task, plus the new example—for each of the tasks that have been seen so far and performing a separate model update for each task. The added computation of performing a separate update for each task is minimal relative to performing a single update across all tasks.

### B.6 Closing the Performance Gap with Offline Learning

We claim that streaming learning methodologies have the potential for drastically-improved performance, but cannot be derived via simple modifications to existing techniques. To substantiate this claim, we perform class-incremental learning experiments on CIFAR100 (Krizhevsky et al., 2009) with two batch-incremental methods—iCarl (Rebuffi et al., 2017) and End-to-End Incremental Learning (E2EIL) (Castro et al., 2018)— and a state-of-the-art streaming approach—REMIND (Hayes et al., 2020); see Appendix A.7 for details. Performance is measured using $\Omega_{\text{all}}$ (Hayes et al., 2018) and displayed in Figure 6.

As can be seen, a performance gap exists between online and offline-trained models; e.g., in Figure 6, the final Top-1 accuracy of REMIND is $> 20\%$ below that of offline training. To be of use to most practitioners, this performance gap between online and offline training must be reduced—*no viable, real-time alternatives to offline training yet exist.* Considering possible sources of such a performance gap, one may look to the freezing of network layers within existing approaches to online learning. Such an approach is shown to significantly degrade network performance in Section 4. Thus, an approach with the potential to perform more comparably to offline training likely should not freeze large portions of network parameters.

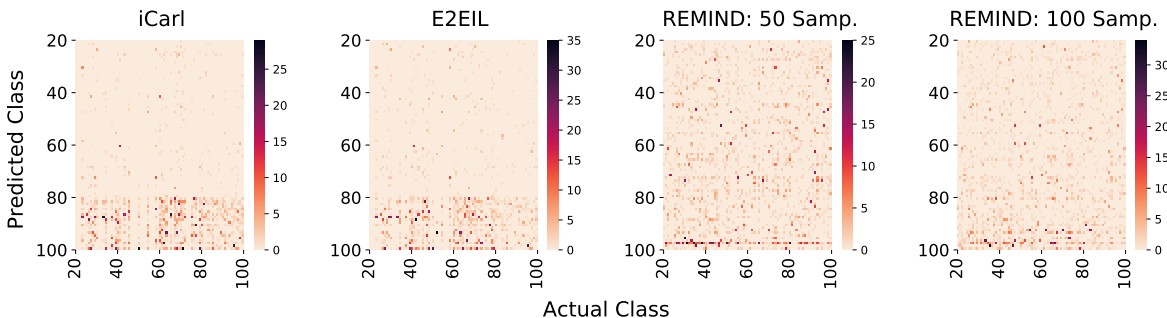

Figure 7: Confusion matrices (excluding base initialization data) for incremental and streaming learning models after class-incremental CIFAR100 training. Incremental learning techniques (iCarl and E2EIL) are biased towards recently-observed classes (see the many incorrect predictions for classes 80-100 in the two leftmost plots). In contrast, incorrect predictions for REMIND are concentrated on the last class observed during streaming (see the line of incorrect predictions at the bottom of third plot). This bias starts getting corrected when 100 replay samples—instead of 50—are used during each online update (see the rightmost plot).

| Method | PQ Encoding Time | Forward Pass Time | Backward Pass Time | Parameter Update Time |
|---|---|---|---|---|
| CSSL | - | $3.67 \pm 0.45$ | $5.41 \pm 0.97$ | $1.53 \pm 0.06$ |
| REMIND | $23.37 \pm 1.13$ | $2.65 \pm 0.02$ | $1.62 \pm 0.03$ | $0.52 \pm 0.01$ |
| REMIND + Extra Params | $23.37 \pm 1.13$ | $2.95 \pm 0.12$ | $1.84 \pm 0.09$ | $0.69 \pm 0.04$ |

Table 15: Timing metrics (in milliseconds) for performing model inference and updates with different streaming methodologies on ImageNet. *Because CSSL is fully-plastic, backward passes and parameter updates take longer to complete, but REMIND's use of product quantization to encode model hidden representations make inference time an order of magnitude slower in comparison to CSSL.*

Interestingly, many incremental learning approaches exist that perform end-to-end training (Castro et al., 2018; Hou et al., 2019; Wu et al., 2019). Thus, one may wonder whether a high-quality streaming approach— one that comes closer to matching offline learning performance—could be developed by simple modifications to such techniques. Unfortunately, these techniques, which employ replay, bias correction, and knowledge distillation to prevent catastrophic forgetting, are known to perform poorly in the streaming domain (Hayes et al., 2020). To understand why this is the case, it should be noted that knowledge distillation is known to provide minimal added benefit when combined with replay mechanisms (Belouadah & Popescu, 2020; 2019). Furthermore, we find that correcting bias in the network's classification layer is unnecessary for streaming learning. Though batch-incremental techniques are biased towards recently-observed classes, REMIND is only biased towards a single class (i.e., the last streamed class), which is easily corrected by increasing the number of replay samples observed during each online update; see Figure 7 and Appendix A.7 for experimental details.[15]

Such observations reveal that findings relevant to batch-incremental learning may not translate to the streaming setting. Additionally, a better streaming methodology cannot be derived by simply adopting and modifying established techniques for batch incremental learning. To close the performance gap with offline learning, streaming techniques likely need to avoid freezing network parameters and leverage more effective forms of replay, as other techniques like knowledge distillation and bias correction provide minimal benefit.

---

[15]Many incremental learning works have been proposed to correct such bias towards recently-observed classes(Hou et al., 2019; Wu et al., 2019). We emphasize that our focus here is not to compare streaming learning to the most recent approaches for incremental learning, but rather to demonstrate that bias towards recent classes is not a consideration for the streaming domain that we consider.

### B.7 Computational Impact of Full Plasticity

Given that CSSL makes all model parameters trainable, one may wonder about the impact of such an approach on the computational complexity of performing updates and inferences with the underlying model throughout streaming. To better understand the complexity impact of full plasticity, we measure the time taken to *i*) perform a forward pass, *ii*) perform a backward pass, and *iii*) update model parameters with CSSL over several streaming iterations. For comparison, we perform the same test with REMIND, which freezes a majority of the underlying network's early layers.

These timing tests use a ResNet18 model and are performed on the ImageNet dataset using the same procedure described in Appendix A.1. At each iteration, 100 replay examples are sampled for both CSSL and REMIND. For REMIND, we also perform supplemental test in which the underlying model has added layers to make the number of trainable parameters equal to that of CSSL. In all cases, we measure timing across five streaming iterations and report the average time; see Table 15.

CSSL has a more expensive forward pass, backward pass, and parameter update in comparison to REMIND, due to the fact that all network layers are updated by CSSL. However, REMIND encodes the output of the network's frozen feature extractor with a product quantization (PQ) (Jegou et al., 2010) module, which is quite slow in comparison to the other operations. As such, when data is passed through the REMIND model—aside from replay examples for which the model's encoded hidden representations can be cached—it must be passed through the frozen portion of the network, encoded via PQ, then passed through the remaining, trainable layers, making the total inference time of REMIND an order of magnitude slower than that of CSSL.

## C   Adapting Baselines to CSSL.

CSSL is capable of randomly initializing the full network—and all network modules—immediately prior to the commencement of streaming; see Section 3. The baseline methodologies used within Section 6.1 (i.e., ExStream, Deep SLDA, and REMIND), however, all utilize pre-trained networks within their respective streaming algorithms. In particular, ExStream (Hayes et al., 2018) and Deep SLDA (Hayes & Kanan, 2020) train only the final, fully-connected layer of the neural network, while REMIND trains the last few convolutional layers and the final fully-connected layer during streaming—all other network layers are fixed either by using pre-trained network parameters or performing offline training over a subset of data during base initialization.

To be capable of starting streaming from a random initialization, baseline methodologies would either have to *i*) allow nearly all (or a majority of) network layers to remain random throughout streaming or *ii*) be adapted to train the network end-to-end. Forcing network layers to remain random throughout streaming catastrophically degrades performance, and adapting such methodologies to perform end-to-end training is highly non-trivial, as a majority of techniques within each methodology would break down (e.g., compressing the feature representations with PQ, clustering feature vectors, computing feature covariance, etc.). As such, we simply compare CSSL to baseline approaches in their original form, without forcing baselines to conform a cold-start setting (i.e., baselines do not satisfy rule four from Section 3).

## D   Technical Results

Within this section, we outline the technical theoretical results (i.e., Lemmas and Corollaries) that are established in the process of proving Theorem 1. Again, full proofs of each result are deferred to Section E.

*Lemma* 1. Assume Assumption 1 holds and $\omega \leq \mathcal{O}\left(L^{-6}\log^{-3}(m)\right)$. Consider iterates $\mathbf{W}^{(0)}, \ldots, \mathbf{W}^{(n)} \in \mathcal{B}(\mathbf{W}^{(0)}, \omega)$ obtained via Algorithm 1 with $B$ replay examples taken from buffer $\mathcal{R}$ per iteration with $\eta = \mathcal{O}(m^{-\frac{3}{2}})$. Given sufficient overparamterization characterized by

$$m \geq \mathcal{O}\left(nB\sqrt{1+\Xi}L^6\log^3(m)\right)$$

We have

$$\sum_{t=1}^{n} L_{(t,\xi_t)}(\mathbf{W}^{(i)}) \leq \sum_{t=1}^{n} L_{(t,\xi_t)}(\mathbf{W}^{\star}) + \mathcal{O}\left(LR^2 m^{-\frac{1}{2}}\right) + \mathcal{O}\left(nm^{-\frac{1}{2}}BL(1+\Xi)\right)$$

for $\mathbf{W}^{\star} \in \mathcal{B}(\mathbf{W}^{(0)}, R/m)$ with $R > 0$ and probability at least $1 - \mathcal{O}(nBL)e^{-\Omega\left(m\omega^{\frac{2}{3}}L\right)}$.

*Lemma* 2. Given Assumption 1 and taking $\omega \leq \mathcal{O}\left(L^{-6}\log^{-3}(m)\right)$, we have that

$$\mathbf{W}^{(0)}, \ldots, \mathbf{W}^{(n)} \in \mathcal{B}(\mathbf{W}^{(0)}, \omega)$$

over $n$ successive iterations of Algorithm 1 with $B$ replay samples taken from buffer $\mathcal{R}$ per iteration, arbitrary perturbation vectors applied to both new and replayed data, $\eta = \mathcal{O}(m^{-\frac{3}{2}})$, and sufficient overparameterization given by

$$m \geq \mathcal{O}\left(nB\sqrt{1+\Xi}L^6\log^3(m)\right)$$

with probability at least $1 - \mathcal{O}(nLB)e^{-\Omega\left(m\omega^{\frac{2}{3}}L\right)}$.

*Lemma* 3. Assume Assumption 1 holds and $\omega \leq \mathcal{O}\left(L^{-6}\log^{-3}(m)\right)$. Then, when $\mathbf{W}, \mathbf{W}' \in \mathcal{B}(\mathbf{W}^{(0)}, \omega)$, we have

$$|f_{\mathbf{W}'}(x_t + \xi) - f_{\mathbf{W}}(x_t + \xi) - \langle \nabla_{\mathbf{W}} f_{\mathbf{W}}(x_t + \xi), \mathbf{W}' - \mathbf{W}\rangle|$$
$$\leq \mathcal{O}\left(\sqrt{m\left(1 + \|\xi\|_2^2\right)}\omega^{\frac{4}{3}}L^3\log(m)\right)$$

for arbitrary perturbation vector $\xi$[16] and all $t \in [n]$ with probability at least $1 - \mathcal{O}(nL)e^{-\Omega\left(m\omega^{\frac{2}{3}}L\right)}$.

*Corollary* 1. Assume Assumption 1 holds and $\omega \leq \mathcal{O}\left(L^{-6}\log^{-3}(m)\right)$, then when $\mathbf{W}, \mathbf{W}' \in \mathcal{B}(\mathbf{W}^{(0)}, \omega)$, we can extend Lemma 3 to show

$$L_{(t,\xi)}(\mathbf{W}') - L_{(t,\xi)}(\mathbf{W}) \geq \sum_{l=1}^{L} \langle \nabla_{W_l} L_{(t,\xi)}(\mathbf{W}), W_l' - W_l\rangle - \mathcal{O}\left(\sqrt{m(1 + \|\xi\|_2^2)}\omega^{\frac{4}{3}}L^3\log(m)\right)$$

for all $t \in [n]$ and arbitrary perturbation vector $\xi$ with probability at least $1 - \mathcal{O}(nL)e^{-\Omega\left(m\omega^{\frac{2}{3}}L\right)}$.

*Lemma* 4. Assume Assumption 1 holds and $\omega \leq \mathcal{O}\left(L^{-\frac{9}{2}}\log^{-3}(m)\right)$. Then, for $\mathbf{W} \in \mathcal{B}(\mathbf{W}^{(0)}, \omega)$ we have

$$\|g_{(t,l,\xi)} - g_{(t,l,\xi)}^{(0)}\|_2, \|h_{(t,l,\xi)} - h_{(t,l,\xi)}^{(0)}\|_2 \leq \mathcal{O}\left(\omega L^{\frac{5}{2}}\sqrt{(1 + \|\xi\|_2^2)\log(m)}\right)$$

for all $t \in [n]$, $l \in [L-1]$, and arbitrary perturbation vector $\xi$ with probability at least $1 - \mathcal{O}(nL)e^{-\Omega\left(m\omega^{\frac{2}{3}}L\right)}$.

*Lemma* 5. Assume Assumption 1 holds and $\omega \leq \mathcal{O}\left(L^{-6}\log^{-3}(m)\right)$. Then, with probability at least $1 - \mathcal{O}(nL)e^{-\Omega\left(m\omega^{\frac{2}{3}}L\right)}$ we have for all $l$ such that $l \in [L-1]$ and $t \in [n]$

$$\left\| W_L' \prod_{r=l}^{L-1}\left(D_{(t,r,\xi)}' + D_{(t,r)}''\right)W_r' - W_L \prod_{r=l}^{L-1} D_{(t,r,\xi)}W_r\right\|_2 \leq \mathcal{O}\left(\omega^{\frac{1}{3}}L^2\sqrt{\log(m)}\right)$$

where $\mathbf{W}, \mathbf{W}' \in \mathcal{B}(\mathbf{W}^{(0)}, \omega)$, $D_{i,r}'' \in [-1,1]^{m\times m}$ is any random diagonal matrix with at most $\mathcal{O}\left(m\omega^{\frac{2}{3}}L\right)$ non-zero entries, and $\xi$ is an arbitrary perturbation vector.

---

[16]Here, we do not assign a subscript $\xi_t$ to the perturbation vector intentionally. This is because the perturbation vector used for this result is arbitrary. Namely, different perturbation vectors can be used for any $t \in [n]$ and both when the data is encountered newly or sampled for replay. The result holds for any perturbation vector chosen for each of these scenarios, where the final bound then depends on the factor $\|\xi\|_2^2$.

*Lemma* 6. Given Assumption 1, randomly initialized weights $\mathbf{W}^{(0)}$, and arbitrary perturbation vector $\xi$ we have

$$\left\| h_{(t,j,\xi)}^{(0)} \right\|_2^2 \in \left[ 1 - \|\xi\|_2^2, 1 + \|\xi\|_2^2 \right]$$

with probability at least $1 - \mathcal{O}(nL) e^{-\Omega(m/L)}$ for all $t \in [n]$ and $j \in [L-1]$.

*Corollary* 2. Given Assumption 1 and assuming $\omega \leq \mathcal{O}(L^{-\frac{9}{2}} \log^{-3}(m))$, we have for $\mathbf{W} \in \mathcal{B}(\mathbf{W}^{(0)}, \omega)$ and arbitrary perturbation vector $\xi$

$$\|h_{(t,j,\xi)}\|_2 \leq \mathcal{O}\left( \sqrt{1 + \|\xi\|_2^2} \left( \omega L^{\frac{5}{2}} \sqrt{\log(m)} + 1 \right) \right)$$

for all $t \in [n]$ and $j \in [L-1]$ with probability at least $1 - \mathcal{O}(nL)e^{-\Omega\left( m\omega^{\frac{2}{3}} L \right)}$.

*Lemma* 7. Given Assumption 1 and $\omega \leq \mathcal{O}\left( L^{-6} \log^{-3}(m) \right)$, the following can be shown when $\mathbf{W} \in \mathcal{B}(\mathbf{W}^{(0)}, \omega)$ for arbitrary perturbation vector $\xi$

$$\|\nabla_{W_l} f_{\mathbf{W}}(x_t + \xi)\|_F, \left\| \nabla_{W_l} L_{(t,\xi)}(\mathbf{W}) \right\|_F \leq \mathcal{O}\left( \sqrt{m(1 + \|\xi\|_2^2)} \right)$$

for all $t \in [n]$ and $l \in [L-1]$ with probability at least $1 - \mathcal{O}(nL)e^{-\Omega\left( m\omega^{\frac{2}{3}} L \right)}$.

# E    Proofs

Within this section, we provide full proofs for all theoretical results derived within this work. We begin with an overview of further notation used within the analysis.

**Notation.** For some arbitrary set of weight matrices $\mathbf{W} = \{W_1, \ldots, W_L\}$, we define $h_{(t,0,\xi)} = x_t + \xi$ and $h_{(t,l,\xi)} = \sigma(W_l h_{(t,l-1,\xi)})$ for $l \in [L-1]$ and arbitrary perturbation vector $\xi$. Similarly, we define $g_{(t,0,\xi)} = x_t + \xi$ and $g_{(t,l,\xi)} = W_l h_{(t,l-1,\xi)}$ for $l \in [L-1]$. For the weight matrices at initialization $\mathbf{W}^{(0)} = \{W_1^{(0)}, \ldots W_L^{(0)}\}$, we would similarly define such vectors with the notation $h_{(t,l,\xi)}^{(0)}$ and $g_{(t,l,\xi)}^{(0)}$ (or $h'_{(t,l,\xi)}, g'_{(t,l,\xi)}$ for weight matrices $\mathbf{W}'$ and so on). We also define $D_{(t,l,\xi)} = \mathtt{diag}(\mathbb{1}\{(W_l h_{(t,l,\xi)})_1 > 0\}, \ldots, \mathbb{1}\{(W_l h_{(t,l,\xi)})_m > 0\})$ for all $t \in [n]$ and $l \in [L-1]$.

## E.1    Proof of Theorem 1

*Proof.* For $t \in [n]$ and associated perturbation vectors $\xi_t$ for $t \in [n]$, let $L_{(t,\xi_t)}^{0-1}(\mathbf{W}^{(t)}) = \mathbb{1}\{y_t \cdot f_{\mathbf{W}^{(t)}}(x_t + \xi_t) < 0\}$, where $L_{(t,\xi_t)}^{0-1}(\mathbf{W}^{(t)}) \leq 4L_{(t,\xi_t)}(\mathbf{W}^{(t)})$ due to properties of the cross entropy loss function (i.e., $\mathbb{1}\{z \leq 0\} \leq 4\ell(z)$). With this in mind, when $m > \mathcal{O}\left(nB\sqrt{1 + \Xi}L^6 \log^3(m)\right)$ and $\omega \leq \mathcal{O}\left(L^{-6} \log^{-3}(m)\right)$, we can invoke Lemma 1 to yield

$$\frac{1}{n} \sum_{t=1}^{n} L_{(t,\xi_t)}^{0-1}(\mathbf{W}^{(t)}) \leq \frac{4}{n} \sum_{t=1}^{n} L_{(t,\xi_t)}(\mathbf{W}^\star) + \mathcal{O}\left( \frac{LR^2}{nm^{\frac{1}{2}}} \right) + \mathcal{O}\left( m^{-\frac{1}{2}} BL(1 + \Xi) \right)$$

with probability at least $1 - \mathcal{O}(nBL)e^{-\Omega\left( m\omega^{\frac{2}{3}} L \right)}$. Then, because the model is trained in a streaming fashion as described in Section 5.1, the $t$-th new data example is seen for the first time only at iteration $t$ within the data stream. Thus, we have that weights $\mathbf{W}^{(t)}$ only depend on data examples $(x_1, y_1), \ldots, (x_{t-1}, y_{t-1})$. In other words, $\mathbf{W}^{(t)}$ is independent of the $t$-th new data example $(x_t, y_t)$. Thus, we can invoke an online-to-batch conversion argument (see Proposition 1 in (Cesa-Bianchi et al., 2004)) to yield

$$\frac{1}{n} \sum_{t=1}^{n} L_{\mathcal{D}}^{0-1}(\mathbf{W}^{(t)}) \leq \frac{1}{n} \sum_{t=1}^{n} L_{(t,\xi_t)}^{0-1}(\mathbf{W}^{(t)}) + \sqrt{\frac{2\log(1/\delta)}{n}}$$

with probability at least $1 - \delta$ for $\delta \in (0, 1]$. From here, we define $\hat{\mathbf{W}}$ as a random entry chosen uniformly from the set $\{\mathbf{W}^{(0)}, \ldots, \mathbf{W}^{(n)}\}$. Thus, we have the following by definition

$$\frac{1}{n} \sum_{t=1}^{n} L_{\mathcal{D}}^{0-1}(\mathbf{W}^{(t)}) = \mathbb{E}\left[L_{\mathcal{D}}^{0-1}(\hat{\mathbf{W}})\right]$$

Then, the following can be derived by combining the above expressions

$$\mathbb{E}\left[L_{\mathcal{D}}^{0-1}(\hat{\mathbf{W}})\right] \leq \frac{4}{n} \sum_{t=1}^{n} L_{(t, \xi_t)}(\mathbf{W}^\star) + \mathcal{O}\left(\frac{LR^2}{nm^{\frac{1}{2}}}\right)$$
$$+ \mathcal{O}\left(m^{-\frac{1}{2}} BL(1 + \Xi)\right) + \sqrt{\frac{2 \log(\frac{1}{\delta})}{n}}$$

(4)

with probability at least $1 - \delta - \mathcal{O}(nBL)e^{-\Omega\left(m\omega^{\frac{2}{3}} L\right)}$ for all $\mathbf{W}^\star \in \mathcal{B}(\mathbf{W}^{(0)}, R/m)$. Now, we can compare the neural network function $f_{\mathbf{W}^\star}$ with $F_{\mathbf{W}^{(0)}, \mathbf{W}^\star}$ as follows

$$L_{(t, \xi_t)}(\mathbf{W}^\star) \overset{(i)}{\leq} \ell(y_t \cdot F_{\mathbf{W}^{(0)}, \mathbf{W}^\star}(x_t + \xi_t)) + \mathcal{O}\left(\sqrt{m(1 + \Xi)} \omega^{\frac{4}{3}} L^3 \log(m)\right)$$

where $(i)$ holds from the 1-Lipschitz continuity of $\ell(\cdot)$ and Lemma 3 with probability at least $1 - \mathcal{O}(nL)e^{-\Omega\left(m\omega^{\frac{2}{3}} L\right)}$. Then, we can plug this inequality into equation 4 to yield

$$\mathbb{E}\left[L_{\mathcal{D}}^{0-1}(\hat{\mathbf{W}})\right] \leq \frac{4}{n} \sum_{t=1}^{n} \ell[y_t \cdot F_{\mathbf{W}^{(0)}, \mathbf{W}^\star}(x_t + \xi_t)] + \mathcal{O}\left(\sqrt{m(1 + \Xi)} \omega^{\frac{4}{3}} L^3 \log(m)\right)$$
$$+ \mathcal{O}\left(\frac{LR^2}{nm^{\frac{1}{2}}}\right) + \mathcal{O}\left(m^{-\frac{1}{2}} BL(1 + \Xi)\right) + \sqrt{\frac{2 \log(\frac{1}{\delta})}{n}}$$

Then, by taking an infimum over all $\mathbf{W}^\star \in \mathcal{B}(\mathbf{W}^{(0)}, R/m)$, we get the following

$$\mathbb{E}\left[L_{\mathcal{D}}^{0-1}\left(\hat{\mathbf{W}}\right)\right] \leq \inf_{f \in \mathcal{F}(\mathbf{W}^{(0)}, R/m)} \left(\frac{4}{n} \sum_{t=1}^{n} \ell(y_t \cdot f(x_t + \xi_t))\right)$$
$$+ \mathcal{O}\left(\sqrt{m(1 + \Xi)} \omega^{\frac{4}{3}} L^3 \log(m)\right) + \mathcal{O}\left(\frac{LR^2}{nm^{\frac{1}{2}}}\right)$$
$$+ \mathcal{O}\left(m^{-\frac{1}{2}} BL(1 + \Xi)\right) + \sqrt{\frac{2 \log(1/\delta)}{n}}$$

From here, by noting that $\omega \leq \mathcal{O}\left(L^{-6} \log^{-3}(m)\right)$ and $m \geq \mathcal{O}\left(nB\sqrt{1 + \Xi} L^6 \log^3(m)\right)$, the expression can be simplified as follows

$$\mathbb{E}\left[L_{\mathcal{D}}^{0-1}\left(\hat{\mathbf{W}}\right)\right] \leq \inf_{f \in \mathcal{F}(\mathbf{W}^{(0)}, R/m)} \left(\frac{4}{n} \sum_{t=1}^{n} \ell(y_t \cdot f(x_t + \xi_t))\right) + \sqrt{\frac{2 \log(1/\delta)}{n}}$$
$$+ \mathcal{O}\left(\frac{(1 + \Xi)^{\frac{3}{4}} \sqrt{nB}}{L^2 \log^{\frac{3}{2}}(m)} + \frac{R^2}{n^{\frac{3}{2}} B^{\frac{1}{2}} (1 + \Xi)^{\frac{1}{4}} L^2 \log^{\frac{3}{2}}(m)}\right.$$
$$\left. + \frac{B^{\frac{1}{2}} (1 + \Xi)^{\frac{3}{4}}}{n^{\frac{1}{2}} L^2 \log^{\frac{3}{2}}(m)}\right)$$

(5)

We then analyze the asymptotic portion of the expression above—highlighted in red—as follows.

$$\mathcal{O}\left(\frac{(1+\Xi)^{\frac{3}{4}}\sqrt{nB}}{L^2\log^{\frac{3}{2}}(m)} + \frac{R^2}{n^{\frac{3}{2}}B^{\frac{1}{2}}(1+\Xi)^{\frac{1}{4}}L^2\log^{\frac{3}{2}}(m)} + \frac{B^{\frac{1}{2}}(1+\Xi)^{\frac{3}{4}}}{n^{\frac{1}{2}}L^2\log^{\frac{3}{2}}(m)}\right)$$

$$= \mathcal{O}\left(\frac{(1+\Xi)B(n^2+1) + R^2}{L^2\log^{\frac{3}{2}}(m)nB^{\frac{1}{2}}(1+\Xi)^{\frac{1}{4}}}\right)$$

$$= \mathcal{O}\left(\frac{(1+\Xi)^{\frac{3}{4}}B^{\frac{1}{2}}n}{L^2\log^{\frac{3}{2}}(m)}\right) + \mathcal{O}\left(\frac{R^2}{L^2\log^{\frac{3}{2}}(m)nB^{\frac{1}{2}}(1+\Xi)^{\frac{1}{4}}}\right)$$

$$\overset{(i)}{\leq} \mathcal{O}\left(\frac{(1+\Xi)^{\frac{3}{4}}B^{\frac{1}{2}}n + R^2}{L^2\log^{\frac{3}{2}}(m)}\right)$$

where $(i)$ holds due to the fact that $nB^{\frac{1}{2}}(1+\Xi)^{\frac{1}{4}} > 1$. Then, by substituting this simplified asymptotic expression, we have

$$\mathbb{E}\left[L_{\mathcal{D}}^{0-1}\left(\hat{\mathbf{W}}\right)\right] \leq \inf_{f\in\mathcal{F}(\mathbf{W}^{(0)}, R/m)}\left(\frac{4}{n}\sum_{t=1}^{n}\ell(y_t\cdot f(x_t+\xi_t))\right) + \sqrt{\frac{2\log(1/\delta)}{n}} + \mathcal{O}\left(\frac{(1+\Xi)^{\frac{3}{4}}B^{\frac{1}{2}}n + R^2}{L^2\log^{\frac{3}{2}}(m)}\right)$$

where the asymptotic portion of the expression can be made arbitrarily small by increasing the value of $m$. Realizing that this result holds with probability $1 - \delta - \mathcal{O}(nBL)e^{-\Omega\left(m\omega^{\frac{2}{3}}L\right)} \approx 1 - \delta$ as the value of $m$ increases yields the desired result. $\qquad\square$

### E.2 Proof of Lemma 1

*Proof.* We assume $\mathbf{W}^{(0)}$ is initialized as described in Section 5.1, then updated according to Algorithm 1 with $B$ replay samples taken from buffer $\mathcal{R}$ at each iteration and distinct, arbitrary data perturbation vectors—representing data augmentation—applied to both new and replayed data throughout streaming. For $R > 0$, we have $\mathbf{W}^{\star} \in \mathcal{B}(\mathbf{W}^{(0)}, R/m)$, where $\mathbf{W}^{\star} \in \mathcal{B}(\mathbf{W}^{(0)}, \omega)$ whenever $m \geq \mathcal{O}(L^6\log^3(m))$ (i.e., $R$ is a constant that does not appear in the asymptotic bound), which is looser than the overparameterization requirement within Lemma 2. Similarly, by Lemma 2, we have $\mathbf{W}^{(0)}, \dots, \mathbf{W}^{(n)} \in \mathcal{B}(\mathbf{W}^{(0)}, \omega)$ with probability at least $1 - \mathcal{O}(nLB)e^{-\Omega\left(m\omega^{\frac{2}{3}}L\right)}$ whenever $m \geq \mathcal{O}\left(nB\sqrt{1+\Xi}L^6\log^3(m)\right)$ and $\eta = \mathcal{O}(m^{-\frac{3}{2}})$.

Thus, for $\mathbf{W}^{(t)}, \mathbf{W}^{\star} \in \mathcal{B}(\mathbf{W}^{(0)}, \omega)$, by Corollary 1 we have with probability at least $1 - \mathcal{O}(nL)e^{-\Omega\left(m\omega^{\frac{2}{3}}L\right)}$ that

$$\begin{aligned} L_{(t,\xi_t)}(\mathbf{W}^{(t)}) - L_{(t,\xi_t)}(\mathbf{W}^{\star}) &\leq \left\langle\nabla_{\mathbf{W}^{(t)}}L_{(t,\xi_t)}(\mathbf{W}^{(t)}), \mathbf{W}^{(t)} - \mathbf{W}^{\star}\right\rangle \\ &\quad + \mathcal{O}\left(\sqrt{m(1+\Xi)}\omega^{\frac{4}{3}}L^3\log(m)\right) \\ &= \sum_{l=1}^{L}\left\langle\nabla_{W_l^{(t)}}L_{(t,\xi_t)}(\mathbf{W}^{(t)}), W_l^{(t)} - W_l^{\star}\right\rangle \\ &\quad + \mathcal{O}\left(\sqrt{m(1+\Xi)}\omega^{\frac{4}{3}}L^3\log(m)\right) \end{aligned} \qquad (6)$$

We can focus on bounding the red term within the expression above as follows

$$\sum_{l=1}^{L} \left\langle \nabla_{W_l^{(t)}} L_{(t,\xi_t)}(\mathbf{W}^{(t)}), W_l^{(t)} - W_l^\star \right\rangle$$

$$\overset{i}{=} \sum_{l=1}^{L} \frac{1}{\eta} \left\langle W_l^{(t)} - W_l^{(t+1)} - \eta \sum_{(x_{\text{rep}}, y_{\text{rep}}) \sim \mathcal{S}_t} \nabla_{W_l^{(t)}} L_{(x_{\text{rep}}, y_{\text{rep}}, \xi_{\text{rep}})}(\mathbf{W}^{(t)}), W_l^{(t)} - W_l^\star \right\rangle$$

$$\overset{ii}{\leq} \sum_{l=1}^{L} \frac{1}{2\eta} \left( \left\| W_l^{(t)} - W_l^{(t+1)} - \eta \sum_{(x_{\text{rep}}, y_{\text{rep}}) \sim \mathcal{S}_t} \nabla_{W_l^{(t)}} L_{(x_{\text{rep}}, y_{\text{rep}}, \xi_{\text{rep}})}(\mathbf{W}^{(t)}) \right\|_F^2 + \left\| W_l^{(t)} - W_l^\star \right\|_F^2 \right.$$

$$\left. - \left\| W_l^{(t+1)} - W_l^\star + \eta \sum_{(x_{\text{rep}}, y_{\text{rep}}) \sim \mathcal{S}_t} \nabla_{W_l^{(t)}} L_{(x_{\text{rep}}, y_{\text{rep}}, \xi_{\text{rep}})}(\mathbf{W}^{(t)}) \right\|_F^2 \right)$$

$$= \sum_{l=1}^{L} \frac{1}{2\eta} \left( \left\| \eta \nabla_{W_l^{(t)}} L_{(t,\xi_t)}(\mathbf{W}^{(t)}) \right\|_F^2 + \left\| W_l^{(t)} - W_l^\star \right\|_F^2 \right.$$

$$\left. - \left\| W_l^{(t+1)} - W_l^\star + \eta \sum_{(x_{\text{rep}}, y_{\text{rep}}) \sim \mathcal{S}_t} \nabla_{W_l^{(t)}} L_{(x_{\text{rep}}, y_{\text{rep}}, \xi_{\text{rep}})}(\mathbf{W}^{(t)}) \right\|_F^2 \right)$$

$$\overset{iii}{\leq} \sum_{l=1}^{L} \frac{1}{2\eta} \left( \left\| \eta \nabla_{W_l^{(t)}} L_{(t,\xi_t)}(\mathbf{W}^{(t)}) \right\|_F^2 + \left\| W_l^{(t)} - W_l^\star \right\|_F^2 \right.$$

$$\left. - \left\| W_l^{(t+1)} - W_l^\star \right\|_F^2 + \left\| \eta \sum_{(x_{\text{rep}}, y_{\text{rep}}) \sim \mathcal{S}_t} \nabla_{W_l^{(t)}} L_{(x_{\text{rep}}, y_{\text{rep}}, \xi_{\text{rep}})}(\mathbf{W}^{(t)}) \right\|_F^2 \right)$$

$$\overset{iv}{\leq} \sum_{l=1}^{L} \frac{1}{2\eta} \left( \left\| W_l^{(t)} - W_l^\star \right\|_F^2 - \left\| W_l^{(t+1)} - W_l^\star \right\|_F^2 \right)$$
$$+ \mathcal{O}\left( \eta B L m (1 + \Xi) \right)$$

where $i$ follows from equation 3, $ii$ holds from the identity that $2\langle A, B\rangle \leq \|A\|_F^2 + \|B\|_F^2 - \|A - B\|_F^2$, and $iii$ holds due to the lower triangle inequality, and $iv$ holds due to the upper triangle inequality and Lemma 7 with probability at least $1 - \mathcal{O}(nBL)e^{-\Omega\left( m\omega^{\frac{2}{3}} L \right)}$ by taking union bound across all data, replay examples, and network layers. Plugging this bound into equation 6, we get

$$L_{(t,\xi_t)}(\mathbf{W}^{(t)}) - L_{(t,\xi_t)}(\mathbf{W}^\star) \leq \sum_{l=1}^{L} \frac{1}{2\eta} \left( \left\| W_l^{(t)} - W_l^\star \right\|_F^2 - \left\| W_l^{(t+1)} - W_l^\star \right\|_F^2 \right)$$
$$+ \mathcal{O}\left( \eta B L m (1 + \Xi) + \sqrt{m(1 + \Xi)} \omega^{\frac{4}{3}} L^3 \log(m) \right)$$

Then, telescoping this expression over $t \in [n]$ yields

$$
\begin{aligned}
\sum_{t=1}^{n} L_{(t,\xi_t)}(\mathbf{W}^{(t)}) &\leq \sum_{t=1}^{n} L_{(t,\xi_t)}(\mathbf{W}^{\star}) + \sum_{l=1}^{L} \frac{\left\| W_l^{(0)} - W_l^{\star} \right\|_F^2}{2\eta} \\
&\quad + \mathcal{O}\left( n\eta BLm(1+\Xi) + n\sqrt{m(1+\Xi)}\omega^{\frac{4}{3}}L^3 \log(m) \right) \\
&\overset{i}{\leq} \sum_{t=1}^{n} L_{(t,\xi_t)}(\mathbf{W}^{\star}) + \frac{LR^2}{2\eta m^2} \\
&\quad + \mathcal{O}\left( n\eta BLm(1+\Xi) + n\sqrt{m(1+\Xi)}\omega^{\frac{4}{3}}L^3 \log(m) \right) \\
&\overset{ii}{\leq} \sum_{t=1}^{n} L_{(t,\xi_t)}(\mathbf{W}^{\star}) + \mathcal{O}\left( LR^2 m^{-\frac{1}{2}} \right) + \mathcal{O}\left( nm^{-\frac{1}{2}}BL(1+\Xi) \right)
\end{aligned}
$$

where $i$ follows from the fact that $\mathbf{W}^{\star} \in \mathcal{B}(\mathbf{W}^{(0)}, R/m)$ and $ii$ holds for sufficiently small $\omega$ with $\eta = \frac{\kappa}{m^{\frac{3}{2}}}$, where $\kappa$ is a small, positive constant. Thus, the desired result holds with probability at least $1 - \mathcal{O}(nBL)e^{-\Omega\left( m\omega^{\frac{2}{3}}L \right)}$ with overparameterization given by the expression below.

$$
m \geq \mathcal{O}\left( nB\sqrt{1+\Xi}L^6 \log^3(m) \right)
$$

$\square$

### E.3  Proof of Lemma 2

*Proof.* We assume $\mathbf{W}^{(0)}$ is initialized as described in Section 5.1 and updated according to Algorithm 1 with $B$ replay examples sampled uniformly from the replay buffer $\mathcal{R}$ at each update and arbitrary perturbation vectors—representing data augmentation—applied separately to both new and replayed data throughout streaming. We take $\omega = C_1 L^{-6} \log^{-3}(m)$ such that $C_1$ is a small enough constant to satisfy assumptions on $\omega$ in Lemma 7. From here, we can show that $\mathbf{W}^{(0)}, \ldots, \mathbf{W}^{(n)} \in \mathcal{B}(\mathbf{W}^{(0)}, \omega)$ through induction.

**Base Case.** The base case $\mathbf{W}^{(0)} \in \mathcal{B}(\mathbf{W}^{(0)}, \omega)$ holds trivially.

**Inductive Case.** Assume that $\mathbf{W}^{(t)} \in \mathcal{B}(\mathbf{W}^{(0)}, \omega)$. By Lemma 7, we have

$$
\left\| \nabla_{W_l} L_{(t,\xi_t)}(\mathbf{W}) \right\|_F \leq \mathcal{O}\left( \sqrt{m(1+\|\xi_t\|_2^2)} \right) \tag{7}
$$

with probability at least $1 - e^{-\Omega\left( m\omega^{\frac{2}{3}}L \right)}$. Denoting the indices of data elements within our replay buffer as $\mathcal{R}$, we can then show the following over iterations of Algorithm 1.

$$
\begin{aligned}
&\left\| W_l^{(t+1)} - W_l^{(0)} \right\|_F \\
&\overset{i}{\leq} \sum_{j=1}^{t} \left\| W_l^{(j+1)} - W_l^{(j)} \right\|_F \\
&\overset{ii}{=} \sum_{j=1}^{t} \left\| -\eta \left( \sum_{(x_{\text{rep}}, y_{\text{rep}}) \sim \mathcal{S}_t} \nabla_{W_l} L_{(x_{\text{rep}}, y_{\text{rep}}, \xi_{\text{rep}})}\left( \mathbf{W}^{(j)} \right) + \nabla_{W_l} L_{(j,\xi_j)}\left( \mathbf{W}^{(j)} \right) \right) \right\|_F \\
&\overset{iii}{\leq} \eta \sum_{j=1}^{t} \left( \sum (x_{\text{rep}}, y_{\text{rep}}) \sim \mathcal{S}_t \left\| \nabla_{W_l} L_{(x_{\text{rep}}, y_{\text{rep}}, \xi_{\text{rep}})}\left( \mathbf{W}^{(j)} \right) \right\| + \left\| \nabla_{W_l} L_{(j,\xi_j)}\left( \mathbf{W}^{(j)} \right) \right\| \right) \\
&\overset{iv}{\leq} \mathcal{O}\left( \eta n B \left( \sqrt{m(1+\Xi)} \right) \right)
\end{aligned}
$$

where $i$ and $iii$ hold due to the upper triangle inequality, $ii$ holds from equation 3, and $iv$ holds due to equation 7 and the definition of $\Xi$. Thus, for $\omega = C_1 L^{-6} \log^{-3}(m)$, if we set $\eta = \frac{\kappa}{m^{\frac{3}{2}}}$ we have

$$\left\| W_l^{(t+1)} - W_l^{(0)} \right\|_F \le \mathcal{O}\left( \eta n B \sqrt{m(1+\Xi)} \right)$$
$$= \frac{\kappa n B \sqrt{1+\Xi}}{\sqrt{m}} \overset{i}{\le} \omega$$

for some small enough absolute constant $\kappa$ and $m \ge \mathcal{O}\left( nB\sqrt{1+\Xi}L^6 \log^3(m) \right)$. Thus, the inductive step is complete and we have $\mathbf{W}^{(0)}, \ldots, \mathbf{W}^{(n)} \in \mathcal{B}(\mathbf{W}^{(0)}, \omega)$ with probability at least $1 - \mathcal{O}(nLB)e^{-\Omega\left( m\omega^{\frac{2}{3}} L \right)}$ by taking a union bound over all data examples, replay examples, and network layers. $\square$

### E.4 Proof of Lemma 3

*Proof.* We consider some fixed $t \in [n]$ with perturbation vector $\xi$ and two weight matrices $\mathbf{W}, \mathbf{W}' \in \mathcal{B}(\mathbf{W}^{(0)}, \omega)$, where $\mathbf{W}^{(0)}$ is initialized as described in Section 3.1. It should be noted that $\xi$ is an arbitrary perturbation vector that can be different for any $t \in [n]$. We omit the subscript $\xi_t$ to emphasize that the result can hold with different perturbations for any $t \in [n]$ and both when data is encountered newly or used for replay. In particular, the result only depends on the value of $\|\xi\|_2^2$, thus allowing arbitrary settings of $\xi$ to be used for new or replayed data. From the formulation of the forward pass in equation 1, it can be shown that $f_{\mathbf{W}'}(x_t + \xi) = \sqrt{m} \cdot W_L' h'_{(t,L-1,\xi)}$ and $f_{\mathbf{W}}(x_t + \xi) = \sqrt{m} \cdot W_L h_{(t,L-1,\xi)}$. These identities allow us to derive the following via direct calculation

$$
\begin{aligned}
&f_{\mathbf{W}'}(x_t + \xi) - F_{\mathbf{W},\mathbf{W}'}(x_t + \xi) \\
&= -\sqrt{m} \cdot \sum_{l=1}^{L-1} W_L \left( \prod_{r=l+1}^{L-1} D_{(t,r,\xi)} W_r \right) D_{(t,l,\xi)} \left( W_l' - W_l \right) h_{(t,l-1,\xi)} \\
&\quad + \sqrt{m} \cdot W_L' \left( \textcolor{red}{h'_{(t,L-1,\xi)} - h_{(t,L-1,\xi)}} \right)
\end{aligned}
\tag{8}
$$

Then, from Claim in 11.2 in (Allen-Zhu et al., 2019), it is known that, for all $t \in [n]$, $h_{(t,L-1,\xi)} - h'_{(t,L-1,\xi)}$ (i.e., the red term within the above expression) can be re-written as

$$
h_{(t,L-1,\xi)} - h'_{(t,L-1,\xi)} = \sum_{l=1}^{L-1} \left( \prod_{r=l+1}^{L-1} \left( D'_{(t,r,\xi)} + D''_{(t,r)} \right) W_r' \right) \left( D'_{(t,l,\xi)} + D''_{(t,l)} \right) \left( W_l - W_l' \right) h_{(t,l-1,\xi)}
$$

where $D''_{(t,l)} \in \mathbb{R}^{m \times m}$ for $l \in [L-1]$ is any random diagonal matrix with at most $\mathcal{O}\left( m\omega^{\frac{2}{3}} L \right)$ non-zero entries in the range $[-1, 1]$. With this in mind, we can then rewrite equation 8 as follows.

$$
\begin{aligned}
&f_{\mathbf{W}'}(x_t + \xi) - F_{\mathbf{W},\mathbf{W}'}(x_t + \xi) \\
&= \sqrt{m} \cdot \sum_{l=1}^{L-1} W_L' \left( \prod_{r=l+1}^{L-1} \left( D'_{(t,r,\xi)} + D''_{(t,r)} \right) W_r' \right) \left( D'_{(t,l,\xi)} + D''_{(t,l)} \right) \left( W_l - W_l' \right) h_{(t,l-1,\xi)} \\
&\quad - \sqrt{m} \cdot \sum_{l=1}^{L-1} W_L \left( \prod_{r=l+1}^{L-1} D_{(t,r,\xi)} W_r \right) D_{(t,l,\xi)} \left( W_l' - W_l \right) h_{(t,l-1,\xi)}
\end{aligned}
$$

Now, given that $\omega \leq \mathcal{O}\left(L^{-6} \log^{-3}(m)\right)$, we can unroll this expression to arrive at the final result as follows

$$
\begin{aligned}
&\left|f_{\mathbf{W}'}(x_t + \xi) - F_{\mathbf{W},\mathbf{W}'}(x_t + \xi)\right| \\
&= \sqrt{m} \cdot \left| \sum_{l=1}^{L-1} \left( W_L' \left( \prod_{r=l+1}^{L-1} \left( D'_{(t,r,\xi)} + D''_{(t,r)} \right) W_r' \right) \left( D'_{(t,l,\xi)} + D''_{(t,l)} \right) (W_l - W_l') h_{(t,l-1,\xi)} \right) \right. \\
&\qquad\qquad \left. - \left( W_L \left( \prod_{r=l+1}^{L-1} D_{(t,r,\xi)} W_r \right) D_{(t,l,\xi)} (W_l' - W_l) h_{(t,l-1,\xi)} \right) \right| \\
&\overset{i}{\leq} \sqrt{m} \cdot \sum_{l=1}^{L-1} \left\| \left( W_L' \left( \prod_{r=l+1}^{L-1} \left( D'_{(t,r,\xi)} + D''_{(t,r)} \right) W_r' \right) \left( D'_{(t,l,\xi)} + D''_{(t,l)} \right) \right. \right. \\
&\qquad\qquad \left. \left. + W_L \left( \prod_{r=l+1}^{L-1} D_{(t,r,\xi)} W_r \right) D_{(t,l,\xi)} \right\|_2 \cdot \|W_l - W_l'\|_2 \cdot \|h_{(t,l-1,\xi)}\|_2 \right. \\
&\overset{ii}{\leq} \mathcal{O}\left( \sqrt{m} \omega^{\frac{1}{3}} L^2 \log(m) \right) \sum_{l=1}^{L-1} \|W_l - W_l'\|_2 \cdot \|h_{(t,l-1,\xi)}\|_2 \\
&\overset{iii}{\leq} \mathcal{O}\left( \sqrt{m\left(1 + \|\xi\|_2^2\right)} \omega^{\frac{1}{3}} L^2 \log(m) \left( \omega L^{\frac{5}{2}} \sqrt{\log(m)} + 1 \right) \right) \sum_{l=1}^{L-1} \|W_l - W_l'\|_2 \\
&\overset{iv}{\leq} \mathcal{O}\left( \sqrt{m\left(1 + \|\xi\|_2^2\right)} \omega^{\frac{1}{3}} L^2 \log(m) \left( \omega L^{\frac{5}{2}} \sqrt{\log(m)} + 1 \right) (\omega L) \right) \\
&\overset{v}{\leq} \mathcal{O}\left( \sqrt{m\left(1 + \|\xi\|_2^2\right)} \omega^{\frac{4}{3}} L^3 \log(m) \right)
\end{aligned}
$$

where $i$ holds due to triangle inequality, $ii$ holds due to Lemma 5 with probability at least $1 - \mathcal{O}(nL)e^{-\Omega\left(m\omega^{\frac{2}{3}}L\right)}$, $iii$ holds due to Corollary 2 with probability at least $1 - \mathcal{O}(nL)e^{-\Omega\left(m\omega^{\frac{2}{3}}L\right)}$, $iv$ holds due to the definition of the $\omega$ neighborhood, and $v$ holds for sufficiently small $\omega$. $\qquad\square$

### E.5  Proof of Corollary 1

*Proof.* Consider some fixed $t \in [n]$ and arbitrary perturbation vector $\xi$, where we again omit the subscript $\xi_t$ to emphasize that the result holds with different perturbations for any $t \in [n]$ and both when data is encountered newly or sampled for replay. We consider $\mathbf{W}, \mathbf{W}' \in \mathcal{B}(\mathbf{W}^{(0)}, \omega)$, where $\mathbf{W}^{(0)}$ is initialized as described within Section 5.1. Recall that we utilize a standard cross-entropy loss function $\ell(z) = \log(1 + e^{-z})$. We denote the derivative of this function as $\ell'(z)$, where $\ell'(z) = \frac{d}{dz} \log(1 + e^{-z}) = \frac{-1}{e^z + 1}$. For $\mathbf{W}, \mathbf{W}' \in \mathcal{B}(\mathbf{W}^{(0)}, \omega)$, we can derive the following

$$
\begin{aligned}
L_{(t,\xi)}(\mathbf{W}') - L_{(t,\xi)}(\mathbf{W}) &= \ell\left(y_t f_{\mathbf{W}'}(x_t + \xi)\right) - \ell\left(y_t f_{\mathbf{W}}(x_t + \xi)\right) \\
&\overset{i}{\geq} \ell'\left(y_t f_{\mathbf{W}}(x_t + \xi)\right) \cdot y_t \cdot \left(f_{\mathbf{W}'}(x_t + \xi) - f_{\mathbf{W}}(x_t + \xi)\right)
\end{aligned}
$$

where $i$ holds due to the convexity of $\ell(\cdot)$. From here, we have

$$
\begin{aligned}
&\ell'\left(y_t f_{\mathbf{W}}(x_t + \xi)\right) \cdot y_t \cdot \left(f_{\mathbf{W}'}(x_t + \xi) - f_{\mathbf{W}}(x_t + \xi)\right) \\
&= \ell'(y_t f_{\mathbf{W}}(x_t + \xi)) \cdot y_t \cdot \Big(f_{\mathbf{W}'}(x_t + \xi) - f_{\mathbf{W}}(x_t + \xi) \\
&\qquad \pm \langle \nabla_{\mathbf{W}} f_{\mathbf{W}}(x_t + \xi), \mathbf{W}' - \mathbf{W}\rangle \Big) \\
&= \ell'\left(y_t f_{\mathbf{W}}(x_t + \xi)\right) \cdot y_t \cdot \langle \nabla_{\mathbf{W}} f_{\mathbf{W}}(x_t + \xi), \mathbf{W}' - \mathbf{W}\rangle \\
&\qquad + \ell'\left(y_t f_{\mathbf{W}}(x_t + \xi)\right) \cdot y_t \cdot \Big(f_{\mathbf{W}'}(x_t + \xi) - f_{\mathbf{W}}(x_t + \xi) \\
&\qquad - \langle \nabla_{\mathbf{W}} f_{\mathbf{W}}(x_t + \xi), \mathbf{W}' - \mathbf{W}\rangle \Big) \\
&\geq \ell'\left(y_t f_{\mathbf{W}}(x_t + \xi)\right) \cdot y_t \cdot \langle \nabla_{\mathbf{W}} f_{\mathbf{W}}(x_t + \xi), \mathbf{W}' - \mathbf{W}\rangle \\
&\qquad - \Big| \ell'\left(y_t f_{\mathbf{W}}(x_t + \xi)\right) \cdot y_t \cdot \Big(f_{\mathbf{W}'}(x_t + \xi) - f_{\mathbf{W}}(x_t + \xi) \\
&\qquad - \langle \nabla_{\mathbf{W}} f_{\mathbf{W}}(x_t + \xi), \mathbf{W}' - \mathbf{W}\rangle \Big)\Big| \\
&= \sum_{l=1}^{L} \langle \nabla_{W_l} L_{(t,\xi)}(\mathbf{W}), W_l' - W_l \rangle - \Big| \ell'\left(y_t f_{\mathbf{W}}(x_t + \xi)\right) \cdot y_t \\
&\qquad \cdot \Big(f_{\mathbf{W}'}(x_t + \xi) - f_{\mathbf{W}}(x_t + \xi) - \langle \nabla_{\mathbf{W}} f_{\mathbf{W}}(x_t + \xi), \mathbf{W}' - \mathbf{W}\rangle \Big)\Big|
\end{aligned}
$$

Then, by noticing that $|\ell'(y_t f_{\mathbf{W}}(x_t + \xi)) \cdot y_t)| \leq 1$ and invoking Lemma 3, we can derive the following with probability at least $1 - \mathcal{O}(nL)e^{-\Omega\left(m\omega^{\frac{2}{3}}L\right)}$

$$
L_{(t,\xi)}(\mathbf{W}') - L_{(t,\xi)}(\mathbf{W}) \geq \sum_{l=1}^{L} \langle \nabla_{W_l} L_{(t,\xi)}(\mathbf{W}), W_l' - W_l \rangle - \mathcal{O}\left(\sqrt{m(1 + \|\xi\|_2^2)} \omega^{\frac{4}{3}} L^3 \log(m)\right)
$$

whenever $\omega \leq \mathcal{O}\left(L^{-6} \log^{-3}(m)\right)$.

$\square$

### E.6 Proof of Lemma 4

*Proof.* Consider some $t \in [n]$ and arbitrary perturbation vector $\xi$, where we omit the subscript $\xi_t$ to emphasize that the result holds with different perturbations for any $t \in [n]$ and both when data is newly encountered or sampled for replay. Consider random weight matrices $\mathbf{W}^{(0)}$ initialized as in Section 5.1, and $\mathbf{W}$ such that $\mathbf{W} \in \mathcal{B}(\mathbf{W}^{(0)}, \omega)$.

From Lemma 6, we have with probability at least $1 - \mathcal{O}(nL)e^{-\Omega(m/L)}$ that $\|h_{(t,j,\xi)}^{(0)}\|_2^2, \|g_{(t,j,\xi)}^{(0)}\|_2^2 \in \left[1 - \|\xi\|_2^2, 1 + \|\xi\|_2^2\right]$. Furthermore, if $m \geq \Omega(nL \log(nL))$, we have that for all $i \in [n]$ and all $1 \leq a \leq b \leq L$

$$
\left\| \prod_{l=a}^{b} W_l^{(0)} D_{(t,l-1,\xi)}^{(0)} \right\|_2 \leq \mathcal{O}(\sqrt{L}) \tag{9}
$$

with probability at least $1 - e^{-\Omega(m/L)}$ by Lemma 7.3 in (Allen-Zhu et al., 2019).[17] Now, we can prove the desired result with an induction argument over layers of the network.

**Base Case.** $\|g_{(t,0,\xi)} - g_{(t,0,\xi)}^{(0)}\|_2 = \|x_t + \xi - (x_t + \xi)\|_2 = 0$, so the base case trivially holds. The same is true for $\|h_{(t,0,\xi)} - h_{(t,0,\xi)}^{(0)}\|_2$.

---

[17] equation 9 has one extra factor of $D_{(t,l-1,\xi)}^{(0)}$ within the expression in comparison to Lemma 7.3 in (Allen-Zhu et al., 2019), but this does not impact the norm of the overall expression because $\left\| D_{l-1}^{(0)} \right\|_2 \leq 1$.

**Inductive Case.** Assume the inductive hypothesis holds for $l - 1$. We can derive the following

$$
\begin{aligned}
g_{(t,l,\xi)} - g^{(0)}_{(t,l,\xi)} &= \left(W_l^{(0)} + W_l - W_l^{(0)}\right)\left(D^{(0)}_{(t,l-1,\xi)} + D_{(t,l-1,\xi)} - D^{(0)}_{(t,l-1,\xi)}\right)\left(g^{(0)}_{(t,l-1,\xi)} + g_{(t,l-1,\xi)} - g^{(0)}_{(t,l-1,\xi)}\right) \\
&\quad - W_l^{(0)} D^{(0)}_{(t,l-1,\xi)} g^{(0)}_{(t,l-1,\xi)} \\
&= \left(W_l - W_l^{(0)}\right)\left(D^{(0)}_{(t,l-1,\xi)} + D_{(t,l-1,\xi)} - D^{(0)}_{(t,l-1,\xi)}\right)\left(g^{(0)}_{(t,l-1,\xi)} + g_{(t,l-1,\xi)} - g^{(0)}_{(t,l-1,\xi)}\right) \\
&\quad + W_l^{(0)}\left(D_{(t,l-1,\xi)} - D^{(0)}_{(t,l-1,\xi)}\right)\left(g^{(0)}_{(t,l-1,\xi)} + g_{(t,l-1,\xi)} - g^{(0)}_{(t,l-1,\xi)}\right) \\
&\quad + W_l^{(0)} D^{(0)}_{(t,l-1,\xi)}\left(g_{(t,l-1,\xi)} - g^{(0)}_{(t,l-1,\xi)}\right)
\end{aligned}
$$

From here, we can telescope over the $g_{(t,l,\xi)} - g^{(0)}_{(t,l,\xi)}$ terms to obtain the expression below, where distinct terms of interest are highlighted with separate colors

$$
\begin{aligned}
g_{(t,l,\xi)} - g^{(0)}_{(t,l,\xi)} = \sum_{a=1}^{l}\left(\prod_{b=l}^{a+1} W_b^{(0)} D^{(0)}_{(t,b-1,\xi)}\right)\Bigg( & \\
& \textcolor{green}{\left(W_a - W_a^{(0)}\right)\left(D^{(0)}_{(t,a-1,\xi)} + D_{(t,a-1,\xi)} - D^{(0)}_{(t,a-1,\xi)}\right)\left(g^{(0)}_{(t,a-1,\xi)} + g_{(t,a-1,\xi)} - g^{(0)}_{(t,a-1,\xi)}\right)} \\
& + \textcolor{red}{W_a^{(0)}\left(D_{(t,a-1,\xi)} - D^{(0)}_{(t,a-1,\xi)}\right)\left(g^{(0)}_{(t,a-1,\xi)} + g_{(t,a-1,\xi)} - g^{(0)}_{(t,a-1,\xi)}\right)}\Bigg)
\end{aligned}
$$

$$(10)$$

Now, we focus on a single summation element of the green term in equation 10 and provide an upper bound on the norm of this expression.

$$
\begin{aligned}
& \left\|\left(\prod_{b=l}^{a+1} W_b^{(0)} D^{(0)}_{(t,b-1,\xi)}\right)\left(W_a - W_a^{(0)}\right)\left(D^{(0)}_{(t,a-1,\xi)} + D_{(t,a-1,\xi)} - D^{(0)}_{(t,a-1,\xi)}\right)\right. \\
& \left.\left(g^{(0)}_{(t,a-1,\xi)} + g_{(t,a-1,\xi)} - g^{(0)}_{(t,a-1,\xi)}\right)\right\|_2 \\
& \leq \left\|\prod_{b=l}^{a+1} W_b^{(0)} D^{(0)}_{(t,b-1,\xi)}\right\|_2 \cdot \left\|W_a - W_a^{(0)}\right\|_2 \cdot \left\|D_{(t,a-1,\xi)}\right\|_2 \cdot \left\|g^{(0)}_{(t,a-1,\xi)} + g_{(t,a-1,\xi)} - g^{(0)}_{(t,a-1,\xi)}\right\|_2 \\
& \overset{i}{\leq} \mathcal{O}\left(\sqrt{L}\right) \cdot \left\|W_a - W_a^{(0)}\right\|_2 \cdot \left\|D_{(t,a-1,\xi)}\right\|_2 \cdot \left\|g^{(0)}_{(t,a-1,\xi)} + g_{(t,a-1,\xi)} - g^{(0)}_{(t,a-1,\xi)}\right\|_2 \\
& \overset{ii}{\leq} \mathcal{O}\left(\sqrt{L}\omega\right) \cdot \left\|D_{(t,a-1,\xi)}\right\|_2 \cdot \left\|g^{(0)}_{(t,a-1,\xi)} + g_{(t,a-1,\xi)} - g^{(0)}_{(t,a-1,\xi)}\right\|_2 \\
& \overset{iii}{\leq} \mathcal{O}\left(\sqrt{L}\omega\right) \cdot \left\|g^{(0)}_{(t,a-1,\xi)} + g_{(t,a-1,\xi)} - g^{(0)}_{(t,a-1,\xi)}\right\|_2 \\
& \overset{iv}{\leq} \mathcal{O}\left(\sqrt{L}\omega\right)\left[\|g^{(0)}_{(t,a-1,\xi)}\|_2 + \|g_{(t,a-1,\xi)} - g^{(0)}_{(t,a-1,\xi)}\|_2\right] \\
& \overset{v}{\leq} \mathcal{O}\left(\sqrt{L}\omega\right)\left[\sqrt{1 + \|\xi\|_2^2} + \mathcal{O}\left(\omega L^{\frac{5}{2}}\sqrt{(1 + \|\xi\|_2^2)\log(m)}\right)\right]
\end{aligned}
$$

where $i$ holds due to equation 9, $ii$ holds because $\mathbf{W} \in \mathcal{B}(\mathbf{W}^{(0)}, \omega)$, $iii$ holds because $\|D_{i,a-1}\|_2 \leq 1$ by construction, $iv$ holds by upper triangle inequality, and $v$ holds by Lemma 6 and the inductive hypothesis.

Now that we have bounded the green term, we can focus on a single summation element of the red term in equation 10. First, we make the following definition

$$
\begin{aligned}
\zeta &\triangleq \left(D_{(t,a-1,\xi)} - D^{(0)}_{(t,a-1,\xi)}\right)\left(g^{(0)}_{(t,a-1,\xi)} + g_{(t,a-1,\xi)} - g^{(0)}_{(t,a-1,\xi)}\right) \\
&= \left(D_{(t,a-1,\xi)} - D^{(0)}_{(t,a-1,\xi)}\right)\left(W_{a-1}^{(0)} h^{(0)}_{(t,a-2,\xi)} + g_{(t,a-1,\xi)} - g^{(0)}_{(t,a-1,\xi)}\right)
\end{aligned}
$$

Then, by Claim 8.3 and Corollary 8.4 in (Allen-Zhu et al., 2019), we have that with probability at least $1 - e^{-\Omega(m\omega^{\frac{2}{3}}L)}$

$$\left\|\frac{1}{c}\zeta\right\|_0 \leq \mathcal{O}\left(m\omega^{\frac{2}{3}}L\right) \quad \text{and} \quad \left\|\frac{1}{c}\zeta\right\|_2 \leq \mathcal{O}\left(\omega L^{\frac{3}{2}}\right) \tag{11}$$

where $c = \|h_{(t,a-2,\xi)}^{(0)}\|_2 \in [\sqrt{1 - \|\xi\|_2^2}, \sqrt{1 + \|\xi\|_2^2}]$. Additionally, we define

$$\gamma \triangleq \left(\prod_{b=l}^{a+1} W_b^{(0)} D_{(t,b-1,\xi)}\right) \cdot W_a^{(0)} \cdot \left[\left(D_{(t,a-1,\xi)} - D_{(t,a-1,\xi)}^{(0)}\right)\left(g_{(t,a-1,\xi)}^{(0)} + g_{(t,a-1,\xi)} - g_{(t,a-1,\xi)}^{(0)}\right)\right]$$

$$= c\left(\prod_{b=l}^{a+1} W_b^{(0)} D_{(t,b-1,\xi)}\right) \cdot W_a^{(0)} \cdot \left[\frac{1}{c}\zeta\right]$$

By invoking Claim 8.5 from (Allen-Zhu et al., 2019), we can reformulate $\gamma$ as

$$\gamma = \left(c\left\|\frac{1}{c}\zeta\right\|_2\right)(\gamma_1 + \gamma_2)$$

where with probability at least $1 - e^{-\Omega(m\omega^{\frac{2}{3}}L\log(m))}$ the $\gamma_1$ and $\gamma_2$ terms can be bounded as

$$\|\gamma_1\|_2 \leq \mathcal{O}\left(L^{\frac{1}{2}}\omega^{\frac{1}{3}}\log(m)\right) \quad \text{and} \quad \|\gamma_2\|_\infty \leq \mathcal{O}\left(\sqrt{\frac{\log(m)}{m}}\right) \tag{12}$$

Combining all of this together, we get

$$
\begin{aligned}
\|g_{(t,l,\xi)} - g_{(t,l,\xi)}^{(0)}\|_2 &= \left\| \sum^{l} \left( \mathcal{O}\left(\sqrt{L}\omega\right) \left[ \sqrt{1+\|\xi\|_2^2} + \mathcal{O}\left(\omega L^{\frac{5}{2}}\sqrt{(1+\|\xi\|_2^2)\log(m)}\right) \right] \right. \right. \\
&\qquad \left. \left. + \left( c\left\|\frac{1}{c}\zeta\right\|_2 \right)(\gamma_1 + \gamma_2) \right) \right\|_2 \\
&= \left\| L\left( \mathcal{O}\left(\sqrt{L}\omega\right) \left[ \sqrt{1+\|\xi\|_2^2} + \mathcal{O}\left(\omega L^{\frac{5}{2}}\sqrt{(1+\|\xi\|_2^2)\log(m)}\right) \right] \right. \right. \\
&\qquad \left. \left. + \left( c\left\|\frac{1}{c}\zeta\right\|_2 \right)(\gamma_1 + \gamma_2) \right) \right\|_2 \\
&\overset{i}{\leq} \mathcal{O}\left(L^{\frac{3}{2}}\omega\right) \left[ \sqrt{1+\|\xi\|_2^2} + \mathcal{O}\left(\omega L^{\frac{5}{2}}\sqrt{(1+\|\xi\|_2^2)\log(m)}\right) \right] \\
&\qquad + L\left( c\left\|\frac{1}{c}\zeta\right\|_2 \right)\|\gamma_1 + \gamma_2\|_2 \\
&\overset{ii}{\leq} \mathcal{O}\left(L^{\frac{3}{2}}\omega\right) \left[ \sqrt{1+\|\xi\|_2^2} + \mathcal{O}\left(\omega L^{\frac{5}{2}}\sqrt{(1+\|\xi\|_2^2)\log(m)}\right) \right] \\
&\qquad + \left( \sqrt{1+\|\xi\|_2^2}\ \mathcal{O}\left(\omega L^{\frac{5}{2}}\right) \right)\|\gamma_1 + \gamma_2\|_2 \\
&\overset{iii}{\leq} \mathcal{O}\left(L^{\frac{3}{2}}\omega\right) \left[ \sqrt{1+\|\xi\|_2^2} + \mathcal{O}\left(\omega L^{\frac{5}{2}}\sqrt{(1+\|\xi\|_2^2)\log(m)}\right) \right] \\
&\qquad + \left( \sqrt{1+\|\xi\|_2^2}\ \mathcal{O}\left(\omega L^{\frac{5}{2}}\right) \right)(\|\gamma_1\|_2 + \|\gamma_2\|_2) \\
&\overset{iv}{\leq} \mathcal{O}\left(L^{\frac{3}{2}}\omega\right) \left[ \sqrt{1+\|\xi\|_2^2} + \mathcal{O}\left(\omega L^{\frac{5}{2}}\sqrt{(1+\|\xi\|_2^2)\log(m)}\right) \right] \\
&\qquad + \left( \sqrt{1+\|\xi\|_2^2}\ \mathcal{O}\left(\omega L^{\frac{5}{2}}\right) \right)\left( \mathcal{O}\left(L^{\frac{1}{2}}\omega^{\frac{1}{3}}\log(m)\right) + \mathcal{O}\left(\sqrt{\log(m)}\right) \right) \\
&= \mathcal{O}\left(L^{\frac{3}{2}}\omega\sqrt{1+\|\xi\|_2^2}\right) + \mathcal{O}\left(\omega^2 L^4\sqrt{(1+\|\xi\|_2^2)\log(m)}\right) \\
&\qquad \mathcal{O}\left(\omega^{\frac{4}{3}}L^3\sqrt{1+\|\xi\|_2^2}\log(m)\right) + \mathcal{O}\left(\omega L^{\frac{5}{2}}\sqrt{(1+\|\xi\|_2^2)\log(m)}\right)
\end{aligned}
$$

where $i$ holds by the upper triangle inequality and properties of norms, $ii$ holds by Lemma 6 and equation 11, $iii$ holds by triangle inequality, and $iv$ holds by equation 12 and invoking the upper bound $\|\gamma_2\|_2 \leq \sqrt{m}\|\gamma_2\|_\infty$. Then, when $\omega$ is sufficiently small, we get

$$
\begin{aligned}
\|g_{(t,l,\xi)} - g_{(t,l,\xi)}^{(0)}\|_2 &\leq \mathcal{O}\left( L^{\frac{3}{2}}\omega\sqrt{1+\|\xi\|_2^2} + \omega L^{\frac{5}{2}}\sqrt{1+\|\xi\|_2^2}\sqrt{\log(m)} \right) \\
&\leq \mathcal{O}\left( \omega L^{\frac{5}{2}}\sqrt{(1+\|\xi\|_2^2)\log(m)} \right) \quad\quad (13)
\end{aligned}
$$

thus completing the inductive portion of the proof for $\|g_{(t,l,\xi)} - g^{(0)}_{(t,l,\xi)}\|_2$. Then, to finish the inductive portion of the proof for $\|h_{(t,l,\xi)} - h^{(0)}_{(t,l,\xi)}\|_2$, we note that

$$
\begin{aligned}
\left\|h_{(t,l,\xi)} - h^{(0)}_{(t,l,\xi)}\right\|_2 &= \|D_{(t,l,\xi)}\left(g_{(t,l,\xi)} - g^{(0)}_{(t,l,\xi)}\right) + \left(D_{(t,l,\xi)} - D^{(0)}_{(t,l,\xi)}\right)g_{(t,l,\xi)}\|_2 \\
&\overset{i}{\leq} \|D_{(t,l,\xi)}\|_2 \cdot \left\|g_{(t,l,\xi)} - g^{(0)}_{(t,l,\xi)}\right\|_2 + \left\|D_{(t,l,\xi)} - D^{(0)}_{(t,l,\xi)}\right\|_2 \cdot \|g_{(t,l,\xi)}\|_2 \\
&\overset{ii}{\leq} 1 \cdot \mathcal{O}\left(\omega L^{\frac{5}{2}}\sqrt{1 + \|\xi\|_2^2}\sqrt{\log(m)}\right) + 1 \cdot \left(\omega L^{\frac{3}{2}}\sqrt{1 + \|\xi\|_2^2}\right) \\
&\leq \mathcal{O}\left(\omega L^{\frac{5}{2}}\sqrt{1 + \|\xi\|_2^2}\sqrt{\log(m)}\right)
\end{aligned}
$$

where $i$ holds from the upper triangle inequality and properties of the spectral norm and $ii$ holds from equation 11 and equation 13. Thus, we have completed the inductive case for $\|h_{(t,l,\xi)} - h^{(0)}_{(t,l,\xi)}\|_2$. From here, a union bound can be taken over all $t \in [n]$ and $l \in [L-1]$ to yield the desired result with probability $1 - \mathcal{O}(nL)e^{-\Omega(m/L)} - \mathcal{O}(nL)e^{-\Omega\left(m\omega^{\frac{2}{3}}L\right)} + \mathcal{O}(nL)e^{-\Omega\left(m\omega^{\frac{2}{3}}L\log(m)\right)} = 1 - \mathcal{O}(nL)e^{-\Omega\left(m\omega^{\frac{2}{3}}L\right)}$, where the last equality again holds when $\omega$ is sufficiently small. $\qquad\square$

### E.7 Proof of Lemma 5

*Proof.* We consider some fixed $t \in [n]$ and arbitrary perturbation vector $\xi$, where we omit the subscript $\xi_t$ to emphasize that the result holds with different perturbations for any $t \in [n]$ and both when data is newly encountered or sampled for replay. We define random diagonal matrices $D''_{(t,1)}, \ldots, D''_{(t,L-1)}$ as any diagonal matrix with at most $\mathcal{O}\left(m\omega^{\frac{2}{3}}L\right)$ entries in the range $\in [-1, 1]$. Consider two sets of model parameters $\mathbf{W}, \mathbf{W}' \in \mathcal{B}(\mathbf{W}^{(0)}, \omega)$, where $\mathbf{W}^{(0)}$ is initialized as described in Section 5.1. From equation 11, we know that

$$
\left\|\frac{1}{\|h^{(0)}_{(t,l-1,\xi)}\|_2}\left(D_{(t,l,\xi)} - D^{(0)}_{(t,l,\xi)}\right)g_{(t,l,\xi)}\right\|_0 \leq \mathcal{O}\left(m\omega^{\frac{2}{3}}L\right)
$$

with probability at least $1 - e^{-\Omega\left(m\omega^{\frac{2}{3}}L\right)}$. This bound holds for both $\mathbf{W}$ and $\mathbf{W}'$. Then, noticing that right multiplication by $g_{(t,l,\xi)}$ and division by a constant factor cannot increase the number of non-zero entries within the matrix $D_{(t,l,\xi)} - D^{(0)}_{(t,l,\xi)}$ (i.e., recall that this matrix is diagonal), we have

$$
\left\|D_{(t,l,\xi)} - D^{(0)}_{(t,l,\xi)}\right\|_0 \leq \mathcal{O}\left(m\omega^{\frac{2}{3}}L\right) \tag{14}
$$

From here, we can apply the upper triangle inequality to yield

$$
\begin{aligned}
\left\|D_{(t,l,\xi)} - D'_{(t,l,\xi)}\right\|_0 &= \left\|\left(D_{(t,l,\xi)} - D^{(0)}_{(t,l,\xi)}\right) - \left(D'_{(t,l,\xi)} - D^{(0)}_{(t,l,\xi)}\right)\right\|_0 \\
&\overset{i}{\leq} \left\|D_{(t,l,\xi)} - D^{(0)}_{(t,l,\xi)}\right\|_0 + \left\|D'_{(t,l,\xi)} - D^{(0)}_{(t,l,\xi)}\right\|_0 \\
&\overset{ii}{=} \mathcal{O}\left(m\omega^{\frac{2}{3}}L\right)
\end{aligned}
$$

where $i$ holds by the upper triangle inequality and $ii$ is due to equation 14. From here, we apply a union bound over all $t \in [n]$ and $l \in [L-1]$ to yield

$$
\left\|D_{(t,l,\xi)} - D'_{(t,l,\xi)}\right\|_0 \leq \mathcal{O}\left(m\omega^{\frac{2}{3}}L\right) \tag{15}
$$

for all $t \in [n]$ and $l \in [L-1]$ with probability at least $1 - \mathcal{O}(nL)e^{-\Omega\left(m\omega^{\frac{2}{3}}L\right)}$. equation 15 can also be extended to show the following properties with identical probability

$$
\begin{aligned}
\left\| D_{(t,l,\xi)} + D''_{(t,l)} - D^{(0)}_{(t,l,\xi)} \right\|_0 &\leq \mathcal{O}\left(m\omega^{\frac{2}{3}}L\right) \\
\left\| D'_{(t,l,\xi)} + D''_{(t,l)} - D^{(0)}_{(t,l,\xi)} \right\|_0 &\leq \mathcal{O}\left(m\omega^{\frac{2}{3}}L\right)
\end{aligned}
\tag{16}
$$

which holds due to the number of non-zero entries assumed to be within each random diagonal matrix $D''_{(t,l)}$ by construction. From here, we first note that with probability at least $1 - e^{-\Omega\left(m\omega^{\frac{2}{3}}L\log(m)\right)}$ we have

$$
\left\| \prod_{r=l}^{L-1} \left( D_{(t,l,\xi)} - D''_{(t,l)} \right) W_l \right\|_2 \leq \mathcal{O}\left(\sqrt{L}\right)
\tag{17}
$$

due to Lemma 8.6 in (Allen-Zhu et al., 2019). Then, we consider bounding the following expression

$$
\left\| W'_L \prod_{r=l}^{L-1} \left( D'_{(t,r,\xi)} + D''_{(t,r)} \right) W'_r - W_L \prod_{r=l}^{L-1} D_{(t,r,\xi)} W_r \right\|_2
$$

$$
= \left\| \left( W'_L + W^{(0)}_L - W^{(0)}_L \right) \prod_{r=l}^{L-1} \left( D'_{(t,r,\xi)} + D''_{(t,r)} \right) W'_r \right.
$$
$$
\left. - \left( W_L - W^{(0)}_L + W^{(0)}_L \right) \prod_{r=l}^{L-1} D_{(t,r,\xi)} W_r \right\|_2
$$

$$
= \left\| \left( W'_L - W^{(0)}_L \right) \prod_{r=l}^{L-1} \left( D'_{(t,r,\xi)} + D''_{(t,r)} \right) W'_r + W^{(0)}_L \prod_{r=l}^{L-1} \left( D'_{(t,r,\xi)} + D''_{(t,r)} \right) W'_r \right.
$$
$$
\left. - \left( W_L - W^{(0)}_L \right) \prod_{r=l}^{L-1} D_{(t,r,\xi)} W_r - W^{(0)}_L \prod_{r=l}^{L-1} D_{(t,r,\xi)} W_r \right\|_2
$$

$$
\overset{i}{\le} \left\| \left( W'_L - W^{(0)}_L \right) \prod_{r=l}^{L-1} \left( D'_{(t,r,\xi)} + D''_{(t,r)} \right) W'_r \right\|_2 + \left\| \left( W_L - W^{(0)}_L \right) \prod_{r=l}^{L-1} D_{(t,r,\xi)} W_r \right\|_2
$$
$$
+ \left\| W^{(0)}_L \prod_{r=l}^{L-1} \left( D'_{(t,r,\xi)} + D''_{(t,r)} \right) W'_r - W^{(0)}_L \prod_{r=l}^{L-1} D_{(t,r,\xi)} W_r \right\|_2
$$

$$
\overset{ii}{\le} \mathcal{O}\left( \sqrt{L}\omega \right) + \left\| W^{(0)}_L \prod_{r=l}^{L-1} \left( D'_{(t,r,\xi)} + D''_{(t,r)} \right) W'_r - W^{(0)}_L \prod_{r=l}^{L-1} D_{(t,r,\xi)} W_r \right\|_2
$$

$$
= \mathcal{O}\left( \sqrt{L}\omega \right) + \left\| W^{(0)}_L \prod_{r=l}^{L-1} \left( D'_{(t,r,\xi)} + D''_{(t,r)} \right) W'_r - W^{(0)}_L \prod_{r=l}^{L-1} D_{(t,r,\xi)} W_r \right.
$$
$$
\left. \pm W^{(0)}_L \prod_{r=l}^{L-1} D^{(0)}_{(t,r,\xi)} W^{(0)}_r \right\|_2
$$

$$
\overset{iii}{\le} \mathcal{O}\left( \sqrt{L}\omega \right) + \left\| W^{(0)}_L \prod_{r=l}^{L-1} \left( D'_{(t,r,\xi)} + D''_{(t,r)} \right) W'_r - W^{(0)}_L \prod_{r=l}^{L-1} D^{(0)}_{(t,r,\xi)} W^{(0)}_r \right\|_2
$$
$$
+ \left\| W^{(0)}_L \prod_{r=l}^{L-1} D_{(t,r,\xi)} W_r - W^{(0)}_L \prod_{r=l}^{L-1} D^{(0)}_{(t,r,\xi)} W^{(0)}_r \right\|_2
$$

$$
\overset{iv}{\le} \mathcal{O}\left( \sqrt{L}\omega \right) + \mathcal{O}\left( \omega^{\frac{1}{3}} L^2 \sqrt{\log(m)} \right)
$$

$$
\overset{v}{\le} \mathcal{O}\left( \omega^{\frac{1}{3}} L^2 \sqrt{\log(m)} \right)
$$

where $i$ and $iii$ hold due to the upper triangle inequality, $ii$ holds due to equation 17, $iv$ holds by Lemma 8.7 in (Allen-Zhu et al., 2019) with probability at least $1 - e^{-\Omega\left( m\omega^{\frac{2}{3}} L \log(m) \right)}$, and $v$ holds for $\omega \le \mathcal{O}\left( L^{-6} \log^{-3}(m) \right)$. Then, the desired result follows with probability at least $1 - \mathcal{O}(nL) e^{-\Omega\left( m\omega^{\frac{2}{3}} L \right)} - \mathcal{O}(nL) e^{-\Omega\left( m\omega^{\frac{2}{3}} L \log(m) \right)} = 1 - \mathcal{O}(nL) e^{-\Omega\left( m\omega^{\frac{2}{3}} L \right)}$ by taking a union bound across all $t \in [n]$ and $l \in [L-1]$.

$\square$

## E.8 Proof of Lemma 6

*Proof.* Consider some fixed $t \in [n]$, $j \in [L-1]$, and arbitrary perturbation vector $\xi$, where we omit the subscript $\xi_t$ to emphasize that the result holds with different perturbations for any $t \in [n]$ and both when

data is newly encountered or sampled for replay. Assume the neural network weights at initialization $\mathbf{W}^{(0)}$ follow the initialization scheme describe in Section 5.1. For $l \in [L]$, we can define $\Delta_l^{(0)} \triangleq \frac{\|h_{(t,l,\xi)}^{(0)}\|_2^2}{\|h_{(t,l-1,\xi)}^{(0)}\|_2^2}$. Applying a logarithm (with arbitrary base $a > 1$) to this definition, we have

$$\log\left(\left\|h_{(t,j,\xi)}^{(0)}\right\|_2^2\right) = \log(\|x_t + \xi\|_2^2) + \sum_{l=0}^{j} \log\left(\Delta_l^{(0)}\right)$$

Given some $\epsilon \in (0, 1]$, we can invoke Fact 7.2 and the proof of Lemma 7.1 from (Allen-Zhu et al., 2019) to show that

$$\left|\sum_{l=0}^{j} \log\left(\Delta_l^{(0)}\right)\right| \leq \epsilon$$

with probability at least $1 - \mathcal{O}(e^{-\Omega(\epsilon^2 m/L)})$. Thus, we have

$$\log(\|x_t + \xi\|_2^2) - \epsilon \leq \log\left(\left\|h_{(t,j,\xi)}^{(0)}\right\|_2^2\right) \leq \log(\|x_t + \xi\|_2^2) + \epsilon \tag{18}$$

We first expand the upper bound to derive a bound on $\|h_{(t,j,\xi)}^{(0)}\|_2^2$. We begin by exponentiating both sides of the inequality in equation 18 to obtain the following

$$\left\|h_{(t,j,\xi)}^{(0)}\right\|_2^2 \leq a^{\log(\|x_t + \xi\|_2^2) + \epsilon}$$
$$= (\|x_t + \xi\|_2^2) \cdot a^\epsilon$$
$$\overset{i}{\leq} (\|x_t\|_2^2 + \|\xi\|_2^2) \cdot a^\epsilon$$
$$\overset{ii}{\leq} (\|x_t\|_2^2 + \|\xi\|_2^2) \cdot a$$
$$\overset{iii}{=} (1 + \|\xi\|_2^2) \cdot a$$

where $i$ follows from the upper triangle inequality, $ii$ is implied by the fact that $a > 1$ and $\epsilon \in (0, 1]$, and $iii$ follows from the unit norm assumption on input data. From here, we note that the base $a$ chosen for the logarithm is arbitrary, and that any base greater than one can be chosen. With this in mind, we note that $\lim_{a \to 1^+}(1 + \|\xi\|_2^2) \cdot a = 1 + \|\xi\|_2^2$, which yields the upper bound $\|h_{(t,j,\xi)}^{(0)}\|_2^2 \leq 1 + \|\xi\|_2^2$.

We can similarly derive a lower bound on $\|h_{(t,j,\xi)}^{(0)}\|_2^2$ as follows, where we begin by exponentiating both sides of equation 18

$$\|h_{(t,j,\xi)}^{(0)}\|_2^2 \geq a^{\log(\|x_t + \xi\|_2^2) - \epsilon}$$
$$= (\|x_t + \xi\|_2^2) \cdot a^{-\epsilon}$$
$$\overset{i}{\geq} \left(\left|\|x_t\|_2^2 - \|\xi\|_2^2\right|\right) \cdot a^{-\epsilon}$$
$$\overset{ii}{\geq} (1 - \|\xi\|_2^2) \cdot a^{-\epsilon}$$
$$\overset{iii}{\geq} (1 - \|\xi\|_2^2) \cdot \frac{1}{a}$$

where $i$ follows from the lower triangle inequality, $ii$ follows from the norm assumption on the data, and $iii$ follows from the fact that $\epsilon \in (0, 1]$ and $a > 1$. Noting that the base $a$ chosen for the logarithm is arbitrary, we have $\lim_{a \to 1^+}(1 - \|\xi\|_2^2) \cdot \frac{1}{a} = 1 - \|\xi\|_2^2$.

By invoking both the upper and lower bounds derived above, we end up with $\|h_{(t,j,\xi)}^{(0)}\|_2^2 \in \left[1 - \|\xi\|_2^2, 1 + \|\xi\|_2^2\right]$ with probability at least $1 - \mathcal{O}\left(e^{-\Omega(m/L)}\right)$ due to the fact that $\epsilon \in (0, 1]$. From here, we can take a union bound over all all $t \in [n]$ and $j \in [L-1]$ to yield the final result with probability $1 - \mathcal{O}(nL)e^{-\Omega(m/L)}$. $\quad\square$

### E.9 Proof of Corollary 2

*Proof.* Consider some fixed $t \in [n]$, $j \in [L-1]$, and arbitrary perturbation vector $\xi$, where we omit the subscript $\xi_t$ to emphasize that the result holds with different perturbations for any $t \in [n]$ and both when data is newly encountered or sampled for replay. Assume the neural network weights at initialization $\mathbf{W}^{(0)}$ follow the initialization scheme described in Section 5.1. From here, we take $\mathbf{W} \in \mathcal{B}(\mathbf{W}^{(0)}, \omega)$. If we then assume $\omega \leq \mathcal{O}\left(L^{-\frac{9}{2}} \log^{-3}(m)\right)$, then we have from Lemma 4 that

$$\left\| h_{(t,j,\xi)} - h_{(t,j,\xi)}^{(0)} \right\|_2 \leq \mathcal{O}\left(\omega L^{\frac{5}{2}} \sqrt{(1 + \|\xi\|_2^2) \log(m)}\right) \tag{19}$$

with probability at least $1 - e^{-\Omega\left(m\omega^{\frac{2}{3}}L\right)}$. Similarly, from Lemma 6, we have the following

$$\left\| h_{(t,j,\xi)}^{(0)} \right\|_2 \leq \sqrt{1 + \|\xi\|_2^2}$$

with probability at least $1 - e^{-\Omega(m/L)}$. Then, we can use these expressions to derive a bound on $\|h_{(t,j,\xi)}\|_2$ as follows

$$\left\| h_{(t,j,\xi)} - h_{(t,j,\xi)}^{(0)} \right\|_2 \overset{i}{\geq} \left| \|h_{(t,j,\xi)}\|_2 - \left\| h_{(t,j,\xi)}^{(0)} \right\|_2 \right|$$
$$\overset{ii}{\geq} \left| \|h_{(t,j,\xi)}\|_2 - \sqrt{1 + \|\xi\|_2^2} \right|$$
$$\geq \|h_{(t,j,\xi)}\|_2 - \sqrt{1 + \|\xi\|_2^2}$$

where $i$ follows from the lower triangle inequality and $ii$ follows from Lemma 6. Then, combining the expression above with equation 19, we derive

$$\|h_{(t,j,\xi)}\|_2 \leq \mathcal{O}\left(\omega L^{\frac{5}{2}} \sqrt{(1 + \|\xi\|_2^2) \log(m)}\right) + \sqrt{1 + \|\xi\|_2^2}$$
$$= \mathcal{O}\left(\sqrt{1 + \|\xi\|_2^2} \left(\omega L^{\frac{5}{2}} \sqrt{\log(m)} + 1\right)\right)$$

Then, by taking a union bound over all $t \in [n]$ and $j \in [L-1]$, we have the desired result with probability at least $1 - \mathcal{O}(nL)e^{-\Omega\left(m\omega^{\frac{2}{3}}L\right)}$. □

### E.10 Proof of Lemma 7

*Proof.* Consider some fixed $t \in [n]$, $l \in [L-1]$, and arbitrary perturbation vector $\xi$, where we omit the subscript $\xi_t$ to emphasize that the result holds with different perturbations for any $t \in [n]$ and both when data is newly encountered or sampled for replay. Given $\mathbf{W} \in \mathcal{B}(\mathbf{W}^{(0)}, \omega)$ with $\mathbf{W}^{(0)}$ initialized as described in Section 5.1, we have

$$\|\nabla_{W_L} f_{\mathbf{W}}(x_t + \xi)\|_2 \overset{i}{=} \left\| \sqrt{m} \cdot h_{(t,L-1,\xi)} \right\|_2$$
$$\overset{ii}{\leq} \mathcal{O}\left(\sqrt{m(1 + \|\xi\|_2^2)} \left(\omega L^{\frac{5}{2}} \sqrt{\log(m)} + 1\right)\right)$$

where $i$ holds because $\nabla_{W_L} f_{\mathbf{W}}(x_t + \xi) = \sqrt{m} \cdot h_{(t,L-1,\xi)}^{\top}$ and $ii$ holds due to Corollary 2 with probability at least $1 - e^{-\Omega\left(m\omega^{\frac{2}{3}}L\right)}$. For layers $l \in [L-1]$, we have

$$\nabla_{W_l} f_{\mathbf{W}}(x_t + \xi) = \sqrt{m} \cdot \left( h_{(t,l-1,\xi)} W_L \left( \prod_{r=l+1}^{L-1} D_{(t,r,\xi)} W_r \right) D_{(t,l,\xi)} \right)^{\top}$$

which allows us to show

$$
\begin{aligned}
\|\nabla_{W_l} f_{\mathbf{W}}(x_t + \xi)\|_F &= \sqrt{m} \cdot \left\| h_{(t,l-1,\xi)} W_L \left( \prod_{r=l+1}^{L-1} D_{(t,r,\xi)} W_r \right) D_{(t,l,\xi)} \right\|_F \\
&\overset{i}{=} \sqrt{m} \cdot \|h_{(t,l-1,\xi)}\|_2 \cdot \left\| W_L \left( \prod_{r=l+1}^{L-1} D_{(t,r,\xi)} W_r \right) \right\|_2 \\
&\overset{ii}{\leq} \mathcal{O} \left( \sqrt{m(1 + \|\xi\|_2^2)} \left( \omega L^{\frac{5}{2}} \sqrt{\log(m)} + 1 \right) \right) \\
&\quad \cdot \left\| W_L \left( \prod_{r=l+1}^{L-1} D_{(t,r,\xi)} W_r \right) \right\|_2
\end{aligned}
\tag{20}
$$

where $i$ holds due to properties of norms and $ii$ holds due to Corollary 2 with probability at least $1 - e^{-\Omega\left( m\omega^{\frac{2}{3}} L \right)}$. Now, we must bound the red term within the expression above to complete the proof

$$
\begin{aligned}
\left\| W_L \left( \prod_{r=l+1}^{L-1} D_{(t,r,\xi)} W_r \right) \right\|_2 &\overset{i}{\leq} \left\| W_L \left( \prod_{r=l+1}^{L-1} \left( D_{(t,r,\xi)} + D''_{(t,r)} \right) W_r \right) \right\|_2 \\
&= \left\| W_L \left( \prod_{r=l+1}^{L-1} \left( D_{(t,r,\xi)} + D''_{(t,r)} \right) W_r \right) \right. \\
&\quad \left. \pm W_L^{(0)} \prod_{r=l+1}^{L-1} D^{(0)}_{(t,r,\xi)} W_r^{(0)} \right\|_2 \\
&\overset{ii}{\leq} \left\| W_L \left( \prod_{r=l+1}^{L-1} \left( D_{(t,r,\xi)} + D''_{(t,r)} \right) W_r \right) \right. \\
&\quad \left. - W_L^{(0)} \prod_{r=l+1}^{L-1} D^{(0)}_{(t,r,\xi)} W_r^{(0)} \right\|_2 \\
&\quad + \left\| W_L^{(0)} \prod_{r=l+1}^{L-1} D^{(0)}_{(t,r,\xi)} W_r^{(0)} \right\|_2 \\
&\overset{iii}{\leq} \mathcal{O} \left( \omega^{\frac{1}{3}} L^2 \sqrt{\log(m)} \right) + \mathcal{O}(1) \\
&= \mathcal{O}(1)
\end{aligned}
\tag{21}
$$

where $i$ holds for some random diagonal matrix $D''_{(t,l)} \in [-1,1]^{m \times m}$ with at most $\mathcal{O}\left( m\omega^{\frac{2}{3}} L \right)$ non-zero entries and $ii$ is due to the upper triangle inequality. $iii$ holds due to Lemma 5 when $\omega \leq \mathcal{O}\left( L^{-6} \log^{-3}(m) \right)$ and Lemma 7.4 in (Allen-Zhu et al., 2019) with probabilities at least $1 - e^{-\Omega\left( m\omega^{\frac{2}{3}} L \right)}$ and $1 - e^{-\Omega(m/L)}$, respectively. Thus, we have from equation 20 and equation 21

$$
\begin{aligned}
\|\nabla_{W_l} f_{\mathbf{W}}(x_t + \xi)\|_F &\leq \mathcal{O} \left( \sqrt{m(1 + \|\xi\|_2^2)} \left( \omega L^{\frac{5}{2}} \sqrt{\log(m)} + 1 \right) \right) \\
&\leq \mathcal{O} \left( \sqrt{m(1 + \|\xi\|_2^2)} \right)
\end{aligned}
$$

where the final inequality holds given suffiently small $\omega$. Then, a union bound can be taken over all $t \in [n]$ and $l \in [L-1]$ to yield the desired bound on $\|\nabla_{W_l} f_{\mathbf{W}}(x_t + \xi)\|_F$ with probability at least

$1 - \mathcal{O}(nL)e^{-\Omega(m/L)} - \mathcal{O}(nL)e^{-\Omega\left(m\omega^{\frac{2}{3}}L\right)} = 1 - \mathcal{O}(nL)e^{-\Omega\left(m\omega^{\frac{2}{3}}L\right)}$. From here, we translate this result to a bound on $\|\nabla_{W_l} L_{(t,\xi)}(\mathbf{W})\|_F$ as follows

$$
\begin{aligned}
\left\|\nabla_{W_l} L_{(t,\xi)}(\mathbf{W})\right\|_F &= \left\|\ell'\left(y_t \cdot f_{\mathbf{W}}(x_t + \xi)\right) \cdot y_t \cdot \nabla_{W_l} f_{\mathbf{W}}(x_t + \xi)\right\|_F \\
&= \left|\ell'\left(y_t \cdot f_{\mathbf{W}}(x_t + \xi)\right) \cdot y_t\right| \cdot \left\|\nabla_{W_l} f_{\mathbf{W}}(x_t + \xi)\right\|_F \\
&\leq \mathcal{O}\left(\sqrt{m(1 + \|\xi\|_2^2)}\right)
\end{aligned}
$$

where the final inequality is derived by noticing that $|\ell'\left(y_t \cdot f_{\mathbf{W}}(x_t + \xi)\right) \cdot y_t| \leq 1$ and invoking the bound on $\|\nabla_{W_l} f_{\mathbf{W}}(x_t + \xi)\|_F$ derived above. Thus, we have arrived at the desired result. $\qquad\square$

