# OpenReview forum: "Cold Start Streaming Learning for Deep Networks"
_TMLR — Withdrawn by Authors_

### Review · Reviewer_Xs73 · 2022-12-27

**Summary Of Contributions:**

This paper proposes a streaming learning approach claiming to make all model parameters trainable, and is not dependent to a beforehand pre-training process. The authors prove the convergence of the proposed approach under certain conditions, and empirically demonstrate its superiority over previous approaches.

**Audience:**

Yes

**Broader Impact Concerns:**

I don't see any potential ethical concern in this submission.

**Claims And Evidence:**

Yes

**Requested Changes:**

Please refer to the weaknesses part.

**Strengths And Weaknesses:**

## Strengths
- The empirical results look pretty strong to me although I'm not familiar with this field and all compared baselines. The proposed CSSL even outperforms previous approaches without pre-training (Table 4).

## Weaknesses
- I would hope that the authors can provide more insights of their improvement. The main components of CSSL, like replay buffer, data compression are not novel. Besides, the "REMIND + Extra Params" baseline that has the same amount of trainable parameters still perform worse than CSSL, even without pre-training. So I cannot get the intuition why it's so much better. I would recommend more control experiments.
- The claim that CSSL does not rely on base initialization is too strong, as it also gains a lot from the pre-training process.
- Some analysis looks pretty straightforward, so maybe not worth it to put them in the main text. For example, Table 1 that analyses the initialization data for REMIND and Figure 2 that analyses the effects of frozen parameters. It could better if the authors can ablate their own proposed method.
- The experimental results only include the image classification settings, are there any more realistic online / stream learning scenarios like recommender system? It would be better to have support from multiple domains.

---

> ### Author Response · Authors · 2023-01-11
> **Response to Reviewer Xs73 (Part One)**
>
> We thank the reviewer for their thoughtful response and look forward to the subsequent discussion. We have responded to each of the itemized concerns below. First, we would like to make a general point about our proposal.
>
> The main benefits of CSSL are:
> - The method is simple (easy to implement/deploy and can be analyzed theoretically)
> - The method performs really well (even without pre-training)
>
> Fundamentally, our contribution is a simple approach that works well practically. The finding that such a simple approach can perform so well is (in our opinion) non-trivial. CSSL surpasses the performance of all baseline models, making it incredibly useful for practical applications (especially given that it is so easy to implement!). We discuss such applications where CSSL may be useful in the paper; see the end of Section 4.
>
> $\texttt{“I would hope …  intuition why it's so much better”}$: As emphasized within the paper, the primary differences between CSSL and REMIND are (i) using an extensive data augmentation pipeline and (ii) training the model end-to-end. If we train the model end-to-end but use only simple augmentations, CSSL performs quite poorly. These results are shown in our data augmentation analysis in Appendix B.1 (Table 10). Once we add data augmentations such as Mixup and AutoAugment, the performance of CSSL improves drastically, indicating that the end-to-end model is overfitting without these augmentations and needs more regularization to perform well.
>
> With this in mind, we claim that the reason CSSL performs well is because of the two reasons outline above (i.e., data augmentation and end-to-end training). With just end-to-end training, CSSL overfits and performs quite poorly. When data augmentation is added this problem goes away and CSSL performs quite well. We try to emphasize this point clearly in the paper but will gladly restructure writing and include the results from Appendix B.1 into the main text if it is deemed necessary!
>
> Beyond our intuitions and analysis of CSSL, we agree that the impressive performance of such a simple approach is somewhat shocking. For this reason, we perform extensive theoretical analysis as explained in Section 5. We emphasize that this analysis is extensive and non-trivial. It holds for multi-layer networks trained in a streaming fashion with a replay buffer and data augmentation. The final convergence rate that is derived provides intuition for our results by demonstrating that the data augmentation procedure does indeed play a role in the final quality/performance of the model.
>
> $\texttt{“I would recommend more control experiments.”}$: Given the discussion above, we claim that there are not extra control experiments that are necessary. We emphasize that end-to-end training is completely necessary for our proposal. A core benefit of CSSL is mitigating the hard requirement upon base initialization that is present in prior methods, which is impossible without end-to-end learning. From here, the only extra component of CSSL is the data augmentation pipeline. We perform extensive ablation experiments that use no augmentation and different combinations of augmentation in Appendix B.1, showing that CSSL performance improves drastically as a product of more sophisticated data augmentation. Thus, we claim that the improved performance of CSSL is made possible via extensive data augmentation.
>
> Notably, training end-to-end also has various performance benefits. This can be seen in the multi-task streaming learning experiments, where CSSL can better adapt to more drastic shifts in the underlying data stream. We see this again in Core50 experiments, where CSSL can recover from poor base initializations over small subsets of data. To understand the improved performance of CSSL in these scenarios, we point to the results illustrated in Figure 5. Within this Figure, we see a clear trend that CSSL reaches a stable plateau in performance over time, while baselines deteriorate significantly after base initialization and steadily decline in performance throughout streaming. Put simply, we see these results as an indication that end-to-end learning prevents CSSL from being permanently biased towards data observed during base initialization.
>
> $\texttt{“The claim that CSSL ... pre-training process.”}$: We agree that this claim is too strong and will modify the writing accordingly. We claim instead that CSSL (i) reduces requirements for base initialization but (ii) can still benefit from pre-training if the application permits. The main differentiation between CSSL and baseline methodologies is that baseline performance completely collapses without a sufficient subset of in-domain data to use for pre-training (e.g., see Core50 or multi-task streaming learning results). CSSL has no such collapse but can still benefit significantly from pre-training.

---

> > ### Author Response · Authors · 2023-01-11
> > **Response to Reviewer Xs73 (Part Two)**
> >
> > $\texttt{“Some analysis … frozen parameters.”}$: We agree that this analysis is straightforward and will gladly move it to the appendix. The main purpose of these experiments is to provide quantitative evidence that (i) freezing parameters during training negatively impacts performance and (ii) using less base initialization data is a problem for REMIND. We include these experiments simply to ensure that these statements are not made without experimental proof. Thus, they allow us to better motivate our proposal by demonstrating some major shortcomings of prior work.
> >
> > $\texttt{“It could better if the authors can ablate their own proposed method.”}$: See above response. For the frozen parameter experiments, we do not adopt a streaming approach. Namely, we just fine-tune an ImageNet pre-trained model on CIFAR-10/100 with different ratios of parameters frozen. We do this specifically to show that a model that is fine-tuned with frozen layers cannot reach the same accuracy as a network that is fine-tuned end-to-end. We claim that such an experiment accomplishes our desired goal, which (as described above) is to demonstrate that freezing network parameters during training has a negative influence on model performance.
> >
> > For experiments that utilize different amounts of data for base initialization, we include these results solely to demonstrate the dependence of REMIND upon high quality base initialization. Although we could repeat these experiments with the proposed method, we emphasize that similar experiments are already included in our experimental section. For example, on Core50 experiments (which have much smaller subsets of data for base initialization), we see the performance of baselines suffer significantly, while CSSL continues to achieve reasonable/better performance. These experiments demonstrate that CSSL can better recover from low quality base initializations in comparison to baseline methods, which is the same finding that would be studied by repeating the experiments of Table 1 with CSSL.
> >
> > We claim that we analyze CSSL (and demonstrate its effectiveness) sufficiently within later experimental analysis. These preliminary experiments are solely meant to provide a bit of motivation for why existing methods may not be desirable in certain situations. We will gladly restructure these experiments to make this more clear, or move any of the less useful results into the appendix.
> >
> > $\texttt{“The experimental results … multiple domains.”}$: We agree that it would be useful to explore the extension of CSSL into different experimental domains. However, we claim that this is a substantially separate publication that should be explored in future work. Our reasoning behind this boils down to the fact that our data augmentation pipeline, which is a key component of our methodology, is domain specific. In different domains (e.g., recommendation systems), the implementation of data augmentation and replay techniques will be substantially different from the computer vision domain, thus requiring separate methodology and analysis.
> >
> > We note that our experimental analysis is already quite extensive. We perform experiments over both small (CIFAR) and large (ImageNet) scale image classification datasets, as well as a dataset for classification in video (Core50). Beyond this, we formulate a new benchmark task (multi-task streaming learning) that studies four different image classification datasets and is never considered in prior work. Considering all of these core experiments (and the more specific analysis performed in subsequent sections), we spent several months of GPU time to complete the experimental section.
> >
> > Additionally, we wish to emphasize that all related works in this area focus heavily (if not solely) upon image classification as well. See for example the following papers that perform experiments only on image classification:
> > - https://arxiv.org/abs/1909.01520
> > - https://arxiv.org/abs/1809.05922
> > - https://arxiv.org/abs/1611.07725
> > - https://arxiv.org/abs/1807.09536

---

### Review · Reviewer_UXW1 · 2023-01-16

**Summary Of Contributions:**

The paper presents a simple online continual learning (or streaming learning) method, Cold Start Streaming Learning (CSSL), which combines replay and data augmentation to avoid forgetting. Theoretical convergence guarantees are derived via Neural Tangent Random Feature (NTRF). Experimental results on baseline methods demonstrates the effectiveness of the proposed method to some extent. Finally, the authors propose a new multi-task streaming learning setting, where CSSL performs favorably well.

**Audience:**

Yes

**Claims And Evidence:**

Yes

**Requested Changes:**

Major points that are all critical:
- Justify the novelty of the proposed method by including and comparing with all recent similar work (representative papers published in top venues such as ICML, NeurIPS, ICLR, CVPR, etc. in the last two years).
- Justify the uniqueness of the benefits of the proposed method.
- Include the most up-to-date state-of-the-art methods in the experiments.

Please see the weaknesses part for more details and add corresponding discussions involving [1-10].

**Strengths And Weaknesses:**

Strengths:

- The paper is well written and easy to understand.
- The theoretical guarantees derived via NTRF is quite interesting.

Weaknesses:

- The method itself seems incremental, because both replay [1-2], data augmentation [3-4], data compression [5-6] are widely adopted in the continual learning community. Although that not all the aforementioned methods are proposed for online continual learning, lots of them could be modified trivially for this setting. Please see [7] for a detailed review on this topic.
- The major benefits of the proposed method are not unique! From a scientific perspective, comparing with only three methods in Section C cannot lead to such conclusion. In fact, the benefits are quite common in existing methods: for example, almost all replay-based methods assumes full plasticity with optional pre-training. Please refer [7] and more recent papers on online continual learning for more details.
- The claim that "Fixing network parameters is detrimental to the learning process." is not necessarily true: [8] assumes a frozen pre-trained transformer backbone, and achieves state-of-the-art performance.
- How is the theory related to *catastrophic forgetting*, the key challenge in continual learning?
- For the experiment part, a *lot* classic work and recent work are missing for comparison. Take the following two recent papers on online continual learning (streaming learning) for example:
  1. [9] is a NeurIPS 2022 paper, which discusses a very similar topic as this paper. Therefore, all comparing methods in Table 1 [9] should be take into consideration.
  2. [10] is a ICML 2022 paper, which also focuses on online continual learning, though from a different perspective. Similarly, all comparing methods in Table 1 [10] should be take into consideration.
Given this, the current comparison results cannot justify the effectiveness of the proposed method properly.
- The so-called multi-task streaming learning setting is not novel, I would rather call it class-incremental setting with larger domain gap.

[1] Buzzega, Pietro, et al. "Dark experience for general continual learning: a strong, simple baseline." NeurIPS 2020.

[2] Fini, Enrico, et al. "Online continual learning under extreme memory constraints." ECCV 2020.

[3] Zhu, Fei, et al. "Class-Incremental Learning via Dual Augmentation." NeurIPS 2021.

[4] Qin, Chengwei, and Shafiq Joty. "Continual Few-shot Relation Learning via Embedding Space Regularization and Data Augmentation." arXiv preprint arXiv:2203.02135 (2022).

[5] Caccia, Lucas, et al. "Online learned continual compression with adaptive quantization modules." ICML 2020.

[6] Wang, Liyuan, et al. "Memory Replay with Data Compression for Continual Learning." ICLR 2022.

[7] Mai, Zheda, et al. "Online continual learning in image classification: An empirical survey." Neurocomputing 469 (2022): 28-51.

[8] Wang, Zifeng, et al. "Learning to prompt for continual learning." CVPR 2022.

[9] Zhang, Yaqian, et al. "A simple but strong baseline for online continual learning: Repeated Augmented Rehearsal." NeurIPS 2022.

[10] Guo, Yiduo, Bing Liu, and Dongyan Zhao. "Online continual learning through mutual information maximization." ICML 2022.

---

> ### Author Response · Authors · 2023-02-17
> **Response to Reviewer UXW1 (Part One)**
>
> We thank the reviewer for their response and for the many useful references that have been provided. We respond to itemized points from the review below.
>
> "$\texttt{The method itself … trivially for this setting.}$": We agree that our technique is simple and a combination of existing techniques. However, we emphasize that the impressive performance of the resulting methodology is highly non-trivial. The performance of CSSL significantly surpasses that of the main streaming learning baseline that precedes our work [1]. Additionally, we emphasize that generalizing batch-incremental learning techniques to the streaming learning setup oftentimes causes issues in terms of performance (e.g., see Fig. 3 and Table 1 in [1]). Nearly all of these techniques have a reliance upon digesting an entire batch of data at a time (e.g., by performing multiple epochs, distillation over all data, etc.).
>
> To make a more general point, we emphasize the following statement on the TMLR website “TMLR emphasizes technical correctness over subjective significance”. Although one may argue that our methodology is not significantly novel, we emphasize that our method perform well and that we perform extensive analysis to supplement this performance. We claim that our approach is technically correct and practically useful (especially due to the ease of implementation).
>
> "$\texttt{The major benefits of … continual learning for more details.}$": Please see the itemized discussion of each of the provided papers below.
>
> "$\texttt{The claim that … is not necessarily true}$": Indeed, freezing network parameters may not impact network performance given a high-quality initialization and a downstream task that is aligned well with the pre-training task. However, we show in Figure 2 that fine-tuning a network (in an offline fashion) with frozen parameters is consistently worse than training the network end-to-end (pre-trained on ImageNet and fine-tuned on CIFAR-10/100). Going further, we emphasize that freezing parameters leads to a massive performance deterioration when the downstream task is poorly aligned with the pre-training task (e.g., see all multi-task streaming learning experiments). Overall, we agree that freezing parameters is not always an issue, but we provide extensive empirical evidence that training the network end-to-end during streaming is likely to be beneficial in certain, notable cases. In many ways, this is the crux of our comparison to CSSL in each of the experimental domains, where we see that CSSL significantly surpasses baseline performance.
>
> "$\texttt{How is the theory related to catastrophic forgetting, the key challenge in continual learning?}$": The theory explicitly shows that the resulting network does not suffer from catastrophic forgetting. There is no assumption made on the underlying data stream (i.e., it could be non-i.i.d.), yet the network still converges given a strategy of streaming, replay sampling, and data augmentation.
>
> "$\texttt{The so-called multi-task streaming … domain gap.}$": We agree. Possibly “novel” is not the correct term. But, this experimental setting is not directly explored in prior work to the best of our knowledge.
>
> Thank you for the extensive references to prior work on online learning. We emphasize that we compare to the same techniques presented by the prior SOTA methodology for streaming learning [1]. A majority of the works referenced by the reviewer are batch-incremental learning techniques. CSSL studies the streaming learning setup. We emphasize that many batch-incremental techniques generalize poorly to the streaming learning domain (e.g., see Fig. 3 and Table 1 in [1]). However, we address each of these papers below (ordered the same as in the review):
>
> $\textbf{Paper 1}$: This method performs multiple training epochs on each incoming data batch (i.e., it does not perform streaming learning). Although we can adapt their method to perform streaming, we claim that this is an unreasonable ask given that it would require modifying their approach to a new setting. Additionally, this technique is only applied to small scale datasets, whereas we test on large scale datasets like ImageNet.
>
> $\textbf{Paper 2}$: It seems that this method performs a modified, batch-level version of streaming. But, we emphasize that this method is also only applied to small scale datasets (largest dataset is CIFAR-10).
>
> $\textbf{Paper 3}$: Again, this paper adopts batch incremental settings that are quite different from streaming learning.
>
> $\textbf{Paper 4}$: This paper studies relation learning problems that are completely unrelated to image classification. Here, it seems non-trivial to generalize their approach to settings we study in our work (i.e., streaming learning for image classification).

---

> ### Author Response · Authors · 2023-02-17
> **Response to Reviewer UXW1 (Part Two)**
>
> $\textbf{Paper 5}$: This paper performs streaming but is focused upon compression rather than image classification. Their approach is only applied to online learning problems for smaller datasets, while their compression scheme is used for iid offline training on larger datasets like ImageNet.
>
> $\textbf{Paper 6}$: Again, this work considers a batch-incremental learning paradigm that receives and processes large chunks of data at a time. Generalizing this methodology to the streaming setting does not seem trivial.
>
> $\textbf{Paper 7}$: Thank you for the reference to this survey!
>
> $\textbf{Paper 8}$: Again this work considers batch-incremental learning, not streaming learning. Additionally, this method is fundamentally based upon leveraging pre-trained models to perform incremental learning, which goes against the main benefits of CSSL.
>
> $\textbf{Paper 9}$: We emphasize that this paper was made publicly available within weeks of our submission to TMLR.
>
> $\textbf{Paper 10}$: Again, this paper studies batch-incremental learning.
>
> Given our discussions of the papers above, it seems like papers 2 and 5 may be the most valid comparisons. However, we emphasize that we already compare to widely established streaming learning techniques (i.e., same as in [1]). In our opinion, these baselines that we choose are the most important comparisons, while papers 2 and 5 seem only tangentially related to streaming learning (e.g., paper 5 especially seems to mostly focus on compression). We are glad to add these as baselines to our work if the reviewer thinks it will significantly improve the quality of our experimental analysis, though we emphasize that we provide extensive empirical analysis (i.e., well over a year of GPU time) of our technique relative to the prior streaming SOTA and several prior techniques.
>
> $\textbf{Citations:}$
>
> [1] Hayes, Tyler L., et al. "Remind your neural network to prevent catastrophic forgetting." Computer Vision–ECCV 2020: 16th European Conference, Glasgow, UK, August 23–28, 2020, Proceedings, Part VIII 16. Springer International Publishing, 2020.

---

### Review · Reviewer_urYR · 2023-01-30

**Summary Of Contributions:**

The submission presents a novel approach for the continuous training of DNN models in a streaming setting (each data is seen only once in a sequential stream of data). Extant models heavily rely on a multistage strategy: (a) initialization using pretraining, following by (b) learning in a streaming setting.  Pretraining may induce inductive biases reducing the capability of the model to adapt to new data. To retain the benefits of pretraining, extant models also freeze a majority of network parameters during streaming, reducing its learning capacity. The presented approach proposes a single phase strategy removing the above constraints, allowing the model to start cold (i.e. without the necessity of pretraining) and for all network weights to change during streaming. It relies on the well-known techniques of memory replay and data augmentation, in a simple training setup allowing novel theoretical convergence guarantees. The approach is also tested in a multitask learning setup. The simple approach is shown to be surprisingly effective, demonstrating highly competitive performance against the state of the art on several image classification benchmarks.


**Audience:**

Yes

**Claims And Evidence:**

Yes

**Requested Changes:**

## Recommended changes for acceptance
My primary recommendations are to make the experimental evaluation more focussed along the following lines, to more systematically communicate and support the gains in the system performance.
### Comparison with a single, CSSL model (or a proposed set)
It is recommended that a single CSSL model, or an appropriately-tagged set is identified and a performance graph (with appropriately selected axes) demonstrating the tradeoffs between these and comparison with other SOTA models is added.
### Data Augmentation
It seems that the primary gains in the proposed approach comes from the utilization of sophisticated data augmentation. It’ll be instructive to apply the same data augmentation on the main baseline SOTA approaches along with a comparison. Also, kindly add how much extra time is needed to compute the augmentations for each training batch.
### Batch vs streaming
While “streaming learning methodologies … cannot be derived via simple modifications to existing techniques” (B.6), CSSL can be trained in the batch-incremental mode. It will be illustrative to compare the performance of CSSL with iCarl, and E2EIL in the batch-incremental mode. (Figure 6)
### Replay Buffer Size
(Section 6.1, Figure 3) Equalizing for replay buffer size allows for different amounts of samples in the memory buffer for different strategies. This confounds the comparative impact of the number of historical samples used in training. Kindly either modify the experiments accordingly or explain if it is difficult to do so.
### Metrics Used
- Some experimental results are reported using Top-1 $\Omega_{all}$ while others using Top-5 $\Omega_{all}$. Kindly incorporate both or use the same one or explain logistical difficulties.
- $\Omega_{all}$ measures the streaming to offline performance ratio. It seems that offline performance should be an upper bound to the achievable streaming performance. If so, how do you explain numbers > 1 in Table 4, even when the buffer size doesn’t admit the entire data to be stored? Smaller than the entire data can indeed be sufficient. However, the metric and the experimental setup that produces numbers > 1 seems to indicate inadequacy of both? Kindly explain.
### Nonstationary settings/ System performance in time
Performance curves as a function of time should be added as they will illustrate the system behavior in adapting to the non-stationary data distribution.
### Multi-task setting
- (a) The task setup is not clear - more clarity, and even a system architecture are needed. It would seem that some part of the representation space would be shared between tasks using some backbone architecture. Several different task heads are then attached to the backbone. Is this correct?
- (b) While the data seems to be ordered similar to the class-incremental setting and hence following that logic (different y-spaces), different memory buffers are used - why?

## Suggestions for quality improvement
Suggestions below will improve the clarity of the writeup.
- **Data stream settings**: The ordering of that data, for the single and multi-task settings, should be made more clear by using a figure to describe the data schema.
- **(Section 6.1, Figure 3)**: Since the discussion reports performance to the third decimal place (0.974) which is difficult to ascertain from the figure, it is encouraged to add a table to the Appendix.
- **“REMIND + Extra Params”** - kindly point to/ add details as to how exactly the architecture is modified to equalize the number of parameters.
- **Typos and Grammar**
    - **(Appendix B.3)** “entrties” → entities
    - **(Appendix B.4)** rephrase paragraph 1, last line.
    - **(Section B.5)** Table B.5 → Table 14.
- **(Appendix B.3)** What is interpolation based eviction strategies a mutually exclusive choice with quantization and resizing strategies in Section 6.4? Why can’t the two be done together?
- **(Section 3.2)** Kindly clearly identify where lossy and lossless compression is used and discuss impact on training time and on computational statistical performance, as applicable.


**Strengths And Weaknesses:**

## Strengths

### Approach & Impact
- The approach at its core is simple and follows the standard methodology similar to offline batch training with the following differences: (a) each data point is seen only once except when in the replay buffer which is dynamically and simply updated, and (b) each training batch is constructed using a sophisticated data augmentation strategy on the sample comprising of the new data and random sampling over the replay buffer. This formulation, while being closer to offline training in spirit, eschews the necessity of pretraining, and freezing of network parameters to retain the gains of pretraining. It bridges the gap between offline and online training, and may also be useful in continuous/ never-ending learning setups.
- Incorporation of sophisticated data augmentation techniques is claimed to be the primary driver, and shown to be effective in obtaining good performance in the proposed, otherwise simple, formulation. Data augmentation incorporates necessary inductive biases (invariances and regularization) to reduce the need of training data. It can be easily performed in both the offline and online settings.
### Theoretical soundness
Theoretical results are presented for asymptotic bounds on generalization error under standard assumptions. This follows standard methodologies available in the literature, but contains several steps and needing to incorporate different results in the literature.
### Novelty
The novelty is moderate and sufficient for publication. Multitask learning in the streaming setting is introduced and the approach is evaluated on this task and compared against the SOTA approaches. Significant performance gains are shown.
### Performance
The proposed approach shows competitive performance against the state of the art on the benchmarks. The performance gains appear robust to various experimental settings, and data orderings.
### Experimental Evaluation
There is an extensive experimental evaluation of the proposed approach on benchmarks against the SOTA. Several performance settings are considered and empirical evaluation of design choices is presented.

## Weaknesses:
### Scope of the submission
The manuscript is too large - 16 pages of main paper + 32 pages of Appendix with experiments, and proofs. In addition, code is provided. The paper contains (a) a new approach (with somewhat limited novelty which is not a concern), (b) an additional, novel task, (c) large set of experiments, and evaluation, and (d) convergence theorems for the new approach backed by 22 pages of technical proofs. This makes not only reviewing hard, but dilutes the primary takeaways and conclusions.
### Lack of clarity
The section on experiments (Section 6) is confusing because of the large variety of experimental setups and associated variations in hyperparameter choices (data augmentation, memory buffer sizes, buffer stored in memory/ on the disk …), and reporting metrics (top-1, top-5) leading to difficulty in drawing comparative conclusions between benefits of different system choices as well as comparative gains over SOTA.
### Proof and Practice
The practical import of the proof on the implemented learning system should be discussed.  Even if there is a gap between ML theory and practice, I recommend doing simulations that provide controlled, empirical validation of the theoretical results.
### Overall
While there seems to be merit to the work: (a) a simple streaming model shown to be effective with powerful data augmentation schemes, and, (b) a corresponding proof of convergence, it seems to me that the paper tries to take on and present too much – leading to lack of clarity, careful analysis and discussion, and a convincing and clear articulation of merit.

---

> ### Author Response · Authors · 2023-02-17
> **Response to Reviewer urYR (Part One)**
>
> We thank the reviewer for their thoughtful and useful response. We respond to itemized points from the review below.
>
> "$\texttt{The manuscript is too large}$": We are glad to make the manuscript more succinct. We believe that this can be accomplished by eliminating Section 4, making the description of the methodology more compact, and trimming down the theoretical results section to focus on practical takeaways. We agree that the paper--especially the main text—can be modified to focus more clearly on key takeaways and contributions.
>
> "$\texttt{The section on … over SOTA.}$": We explore a variety of experimental domains and even propose a new domain to highlight the benefit of CSSL. All experimental domains and metrics (excluding multi-task streaming learning) are directly adopted from prior work [1]. For multi-task streaming learning, we choose to report performance in terms of top-1 accuracy for simplicity. To match other metrics, we could compute an $\Omega_{\text{all}}$ score. However, this would require training multiple baseline models across each dataset to normalize performance, which we thought would be more complicated and less insightful compared to top-1 accuracy.
>
> "$\texttt{The practical import … theoretical results.}$": We agree that numerical results related to the proof would be useful. However, the style of proof that we have chosen is not apt for this type of analysis. For example, the paper upon which our theoretical analysis is based [2] performs no numerical simulations. Given that our theoretical argument is based upon the neural tangent random feature (NTRF) construct, it is not clear how we could perform a numerical simulation to support this bound.
>
> "$\texttt{It is recommended that a single CSSL model … SOTA models is added.}$": Please see Figure 5. We tried to provide exactly this visualization—a plot of CSSL performance (in terms of top-1 accuracy) compared to all baselines throughout the duration of the streaming process.
>
> "$\texttt{It’ll be instructive … SOTA approaches along with a comparison.}$": Applying data augmentation to the baseline methods used in this work is not possible, as the baseline methods operate on feature vectors and not images. The baseline methods pass each image that arrives in the data stream through a fixed feature extractor, then store feature vectors within the replay buffer (either directly or using a quantized hash table) instead of actual images. The data augmentation techniques that we use for CSSL are not directly applicable to these feature vectors. However, we do apply mixup and random crops to these feature vectors as is recommended in [1].
>
> "$\texttt{Also, kindly … augmentations for each training batch.}$": Currently, we have added timing analysis that studies the difference in latency between the forward and backward pass of REMIND and CSSL; see Table 15. These timing metrics implicitly provide information about the added time due to data augmentation, though we are glad to add more specific information about data augmentation in particular to this table (e.g., as an extra method of CSSL without augmentations).
>
> "$\texttt{It will … iCarl, and E2EIL in the batch-incremental mode.}$": We will gladly add CSSL results to this plot. Streaming learning can be easily adapted to batch incremental settings by just performing streaming over each batch of data. If we take this naive approach (as is done for REMIND in Figure 6), CSSL actually still outperforms both E2EIL and iCARL; see Figure 5 as an example of this performance.
>
> "$\texttt{(Section 6.1, Figure 3) Equalizing … difficult to do so.}$": The problem with fulfilling this request is that all baseline approaches can store the full dataset within the replay buffer using minimal memory (less than the lowest level in Figure 3). This is because these methods use a fixed feature extractor that converts images into low-dimensional vectors. These vectors can then be quantized to save memory, making the replay buffer incredibly memory efficient. CSSL is admittedly less memory efficient than these baseline techniques, but we claim that CSSL can offer far improved performance at the cost of using more memory for replay. To emphasize this point, we present CSSL performance at multiple different memory capacities for the replay buffer, as is done in Figure 3. We understand that using different numbers of samples between CSSL and baselines is somewhat confusing, but this is unavoidable given that baselines compress each sample so significantly.
>
> "$\texttt{Some experimental results are reported using Top-1 while others using Top-5}$": The choice of Top-5 vs. Top-1 $\Omega_{\text{all}}$ is made to match exactly the settings of prior work [1]. The main reason for doing this is to ensure that the performance of our baselines matches prior work closely. Although we could add both Top-1 and Top-5 metrics, we found that this provides no added information (i.e., patterns in performance are the same).

---

> ### Author Response · Authors · 2023-02-17
> **Response to Reviewer urYR (Part Two)**
>
> "$\texttt{Offline performance should be an upper bound to the achievable streaming performance}$": $\Omega_{\text{all}}$ scores $>1$ are explained by the fact that our offline models are trained with normal data augmentation (crops and flips) to match experimental settings of prior work [1]. Because of this, CSSL may outperform offline training baselines in certain cases (especially small datasets prone to overfitting) simply due to the use of better data augmentation. We are glad to clarify this in the paper.
>
> "$\texttt{Performance curves as a function of time … non-stationary data distribution.}$": Reporting performance in terms of time is difficult due to implementation differences between CSSL and baseline techniques. To provide meaningful results, we would have to implement all techniques using the same framework/methods to ensure timing is accurate and not due to implementation differences. To provide some idea of timing differences, we report timing of different streaming operations using a standardized implementation of REMIND/CSSL in Table 15.
>
> "$\texttt{It would seem … attached to the backbone.}$": Exactly correct. All layers are shared except for the final classification layer. Each dataset has its own classification layer. We can modify the text to make this more clear.
>
> "$\texttt{Data stream settings...}$": We are glad to add such a figure.
>
> "$\texttt{While the data … different memory buffers are used - why?}$": Different memory buffers are used because each task/dataset has its own output space (i.e., different numbers of classes). Mixing replay examples from different datasets in a single update is difficult. We cannot compute cross entropy loss over multiple target vectors of different size (though maybe this could be custom implemented using a padding approach).  For this reason, we adopt the simpler approach of storing replay examples in a separate buffer for each dataset. Then, we can easily perform separate replay sampling/updates on each dataset.
>
> "$\texttt{Since the discussion … add a table to the Appendix}$": We will gladly add this table.
>
> "$\texttt{Remind + Extra Params}$": Please see Appendix A.1 under “Baseline details”.
>
> "$\texttt{What is interpolation … done together?}$": Interpolation-based eviction, instead of getting rid of an image, combines it with another via an average or using a technique like Mixup (i.e., we interpolate two images instead of evicting an image). Thus, replay samples may contain multiple images that are interpolated into a single image. This strategy can be combined with quantization/resizing. However, interpolation eviction strategies were not found to work well even without added quantization/resizing. As such, we decided to not further explore this technique.
>
> "$\texttt{Kindly clearly identify where lossy and lossless … as applicable.}$": Resizing is performed via simple image resizing, which is lossy. Using differently-sized images does not impact the speed of the forward/backward pass much on a GPU. Our quantization scheme is also lossy, as it just “rounds off” information beyond the specified number of bits. This quantization is simulated in software via rounding, so it is difficult to measure the timing impact of this approach (i.e., we would have to implement this on an FPGA or something similar that can support arbitrary precision storage/arithmetic). The statistical impact on performance is measured by comparing techniques with different levels of quantization/resizing; see Table 11 for these results.
>
> We will gladly fix all typos and minor errors within the writing and thank the reviewer for pointing these out.
>
> $\textbf{Citations:}$
>
> [1] Hayes, Tyler L., et al. "Remind your neural network to prevent catastrophic forgetting." Computer Vision–ECCV 2020: 16th European Conference, Glasgow, UK, August 23–28, 2020, Proceedings, Part VIII 16. Springer International Publishing, 2020.
>
> [2] Cao, Yuan, and Quanquan Gu. "Generalization bounds of stochastic gradient descent for wide and deep neural networks." Advances in neural information processing systems 32 (2019).

---

### Note · Authors · 2023-03-11

**Comment:**

We thank the reviewers for their valuable insight. The latest date for reviewers to make a formal decision was Feb. 27. Given that this decision date has passed several weeks ago, we choose to withdraw the paper and resubmit to another venue with the suggested changes.

**Withdrawal Confirmation:**

I have read and agree with the venue's withdrawal policy on behalf of myself and my co-authors.